# Memorize or Generalize? Evaluating LLM Code Generation with Code Rewriting

## Abstract

Large language models (LLMs) have recently demonstrated exceptional code generation capabilities. However, there is a growing debate whether LLMs are mostly doing memorization (i.e., replicating or reusing large parts of their training data) versus generalization (i.e., beyond training data). Existing evaluations largely proxy memorization with surface/structural similarity, thereby conflating benign reuse of repeated code with harmful recall and neglecting task correctness under semantic variation. We define harmful memorization behaviorally as *failure at high similarity* and introduce a semantic perturbation *code rewriting*, which rewrites a semantically different answer at a similar difficulty level for a given coding task, then reverse-engineers a novel coding task. We further propose *Memorization Risk Index (MRI)*, a normalized score that combines two signals: (i) how similar the model's answer for the rewritten task is to the original ground-truth solution, and (ii) how much performance drops from the original task to its rewritten counterpart. MRI is high only when both conditions hold—when the model outputs similar code but fails the perturbed task—thereby capturing harmful memorization rather than benign reuse of repeated code. Empirical evaluations on code generation benchmarks MBPP+ and BigCodeBench reveal that (1) memorization does not increase with larger models and in many cases alleviates as they scale; (2) supervised fine-tuning (SFT) improves accuracy while introduces memorization; (3) reinforcement learning with proximal policy optimization (PPO) achieves a more balanced trade-off between memorization and generalization.

## 1 Introduction

Large language models (LLMs) have made incredible advances in automated code generation, and are rapidly becoming essential tools in software development (Sourcegraph, 2024; Tabnine, 2024; Team et al., 2023; Anthropic, 2025; Chen et al., 2021). Modern code-focused LLMs can achieve state-of-the-art performance on programming benchmarks (Rozière et al., 2024). For example, specialized models like Qwen-2.5 Coder (Hui et al., 2024) and Code Llama (Rozière et al., 2024) have pushed the boundaries of translating natural language into code. These advancements raise an important question: when do LLMs truly generalize to new programming tasks, and when are they merely reproducing memorized training examples?

Understanding memorization in code generation is critical. Existing evaluations largely measure memorization via surface or structural overlap (e.g., regurgitation audits, contamination filters, and entropy-based detectors) (Yang et al., 2024; Riddell et al., 2024; Dong et al., 2024), treating high similarity as evidence of memorization. This conflates benign reuse of repeated code (i.e. idioms, APIs) with harmful recall and, crucially, does not test whether the model solves the task under semantic variation.

To systematically study harmful memorization, we build on the intuition that performance gaps under semantic perturbations contribute to reveal whether a model is generalizing or harmful memorizing. If a model simply recalls solutions, even small semantic changes could cause large accuracy drops, often accompanied by high overlap with training-like code (Bayat et al., 2024). Concretely, we propose *code-rewriting*, which introduces semantic shifts to prompts while maintaining similar syntax, to investigate whether the success of a model comes from genuine reasoning or harmful memorization. To quantify these behaviors, we introduce Memorization Risk Index (MRI), a nor-

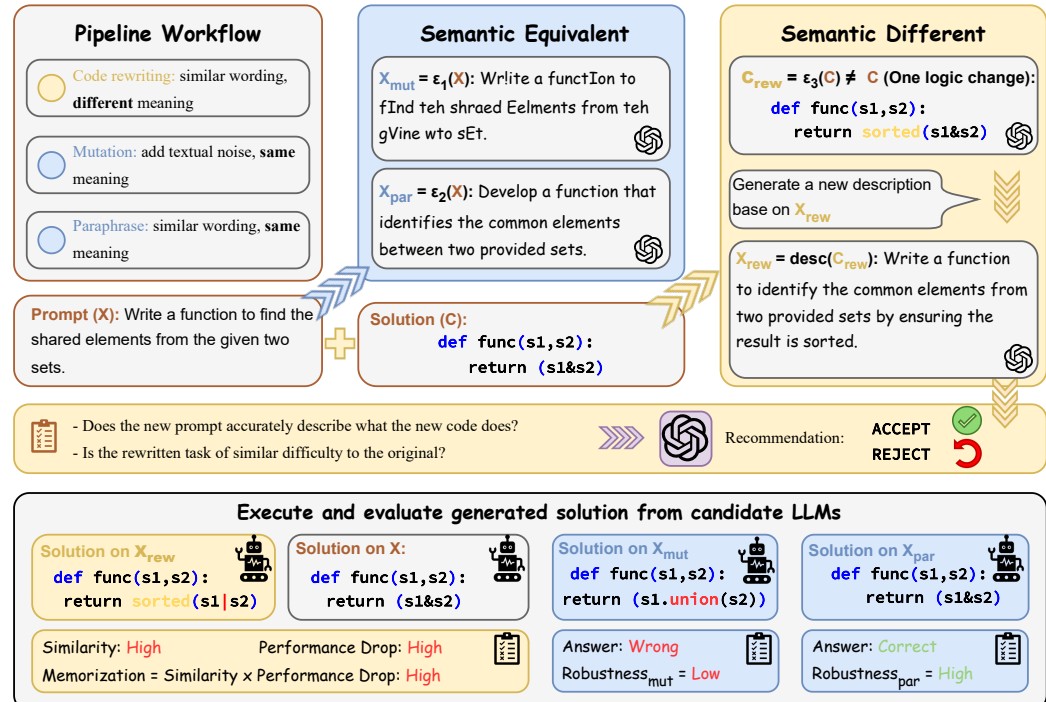

Figure 1: Our proposed Code Rewriting vs. Popular semantic equivalent perturbations. $X$ denotes text and $C$ denotes code. *Code rewriting* that creates semantically different tasks, first rewrite a new code solution $C_{\text{rew}}$ from the origin solution $C$, then generating a new description $X_{\text{rew}}$ based on $C_{\text{rew}}$. A judge agent will then choose to accept or reject the *code rewriting* task for quality assurance. *Mutation* and *paraphrase* that create semantically equivalent tasks, are included for robustness evaluation as a comparison to memorization. All perturbations are performed by GPT-5, shown as the ChatGPT logo. Generation prompts are in Appendix B.

malized score that combines two signals: (i) how similar the model's answer for the *code rewriting* task is to the original ground-truth solution (combining both semantic and syntax level similarity), and (ii) how much performance drops from the original task to its *code rewriting* counterpart. MRI captures harmful memorization as *failure under high similarity* on code-rewriting tasks.

**Terminology.** In this paper, we use the term *memorization* to specifically denote *harmful memorization*: which we define as cases that (1) exhibit high similarity to the original solution and (2) lead to performance drops under semantically altered *code rewriting*. Unless otherwise stated, all subsequent uses of "memorization" follow this definition.

To differentiate our method from existing work in evaluating robustness (Chen et al., 2024; 2023; Mastropaolo et al., 2023; Wang et al., 2022), we also include two semantic-preserving perturbations, *mutation* and *paraphrase*, as reference baselines. We report Relative Accuracy Drop (RAD) to measure LLMs performance consistency under semantic-preserving perturbations. Our primary analysis still targets harmful memorization via the semantics-altering code-rewriting perturbation and MRI.

Our evaluation include coding benchmarks across different difficulty levels, from simpler problems in MBPP+ (Liu et al., 2023) to more difficult tasks in BIGCODEBENCH (Zhuo et al., 2024). Furthermore, we investigate the effect of post-training strategies by comparing Supervised Fine-Tuning (SFT) and Proximal Policy Optimization (PPO). Our results reveal the following trends: **(1)** harmful memorization does not increase with larger models and in many cases improves as they scale; **(2)** harmful memorization alleviates rapidly on simpler tasks but persists on more difficult ones; **(3)** SFT improves raw accuracy but substantially amplifies harmful memorization; **(4)** PPO achieves a more balanced trade-off, mitigating harmful memorization while maintaining competitive accuracy.

In summary, our work makes the following key contributions:

• We propose a novel **automated pipeline** for *code rewriting*, which rewrites a semantically different answer at a similar difficulty level for a given coding question, then reverse-engineers a novel coding question.

- We introduce MRI, a metric that captures harmful memorization as **failure under high similarity** on *code rewriting* tasks, rather than treating similarity alone as memorization.
- We conduct a comprehensive empirical study across benchmarks and training strategies, providing insights into when LLMs memorize in code generation.

## 2 RELATED WORK

### 2.1 CODE GENERATION WITH LLMS

Large Language Models (LLMs) have shown remarkable ability in automated code generation. Models such as ChatGPT (OpenAI et al., 2024), Qwen-Coder (Hui et al., 2024), and DeepSeek-Coder (Guo et al., 2024) have pushed the boundaries in the coding domain, notably with ChatGPT achieving state-of-the-art performance on challenging benchmarks such as BIGCODEBENCH (Zhuo et al., 2024), LiveCodeBench (Jain et al., 2024) and EvalPlus leaderboard (Liu et al., 2023). While LLM-based code generation models have made significant strides in translating natural language to executable code, most evaluations focus on static benchmark performance, overlooking memorization behaviors to prompt variations.

### 2.2 MEMORIZATION IN CODE GENERATION

A model that memorizes may output correct-looking solutions simply because it has seen near-identical problems during pre-training, rather than reasoning about program semantics (Pappu et al., 2024; Duan et al., 2024; Kassem et al., 2024; Carlini et al., 2019; Bayat et al., 2024; Xie et al., 2024). Such behavior can mislead evaluation benchmarks, inflate metrics, and compromise trustworthiness when models are deployed in real-world development environments (Hartmann et al., 2023; Lu et al., 2024; Staab et al., 2023; Zanella-Béguelin et al., 2020).

In code generation, prior work operationalizes general memorization as *regurgitation*—via prefix to suffix extraction, mass sampling with clone detection against the training corpus, and contamination-aware splits of HumanEval/MBPP (Chen et al., 2021)—and under these measurements reports that the measured general memorization rate *increases with model size* (Yang et al., 2024; Al-Kaswan et al., 2024; Wang et al., 2024). However, general memorization is not inherently harmful: if a training-like solution still satisfies the a semantic different question, re-use does not constitute risk (it is correct and passes tests) (Bayat et al., 2024). To distinguish harmful memorization from genuine generalization, we introduce *code-rewriting*, which deliberately shifts task semantics while preserving surface syntax, and we quantify it with a *Memorization Risk Index (MRI)* that multiplies similarity to the original solution by the relative accuracy drop under the semantic shift (high only when the answer copied surface forms but fail on the task with new semantics). Lai et al. (2022) uses semantic perturbations—changing the reference solution's semantics without increasing difficulty—to probe general memorization; unlike their manually authored data-science tasks, we perform automated code rewriting and introduce MRI.

## 3 METHODOLOGY

### 3.1 CODE REWRITING

*Code rewriting* is used to evaluate a model's memorization via solving **semantically different** problems that are superficially similar to original tasks. The automated pipeline to generate *code rewriting* tasks is shown in Figure 1. Specifically, we first modify **one logic** in ground truth solution while preserving the original function signature—including the function name, input, and output format. We then generate a new task description that reflects the altered code while similar to origin tasks in syntax. Formally, let $x \in T$ be the original prompt in text space $T$ and $c \in C$ its ground truth code solution in code space $C$. We apply a rewriting function $\epsilon_3$ that produces a new code $c_{rew} = \epsilon_3(c)$ where $c_{rew} \neq c$ functionally but both $c$ and $c_{rew}$ share the same signature. The new prompt $x_{rew}$ is then generated from $c_{rew}$, resulting in a semantically different task:

$$x_{rew} = \text{desc}(c_{rew}) \tag{1}$$

$$\text{where} \quad \text{sig}(c_{rew}) = \text{sig}(c), \quad c_{rew} \neq c \tag{2}$$

where desc($\cdot$) denotes generating a description from code, and sig($\cdot$) extracts the function signature. This process enables us to assess whether LLMs can recognize and solve tasks that share format but differ in semantic content.

**Data Validation.** To ensure the reliability of *code rewriting* datasets, we conducted both LLM-as-a-judge and manual quality assurance. For LLM-as-a-judge (shown in Figure 1), we forward *code rewriting* tasks to GPT-5 (OpenAI, 2025) to check (i) if the rewritten code match the rewritten prompt and (ii) if the rewritten task align with the original task in difficulty. For manual validation, two experienced python programmers randomly reviewed 10% of generated evolution problems for all three evolution types to ensure their quality. We also provide 5 regressed tasks for each dataset (PASSED in original but FAILED in *code rewriting*) in Appendix H to show difficulty alignment. Additionally, we validate each rewritten solution by making it pass its corresponding rewritten unit test. Details about this process can be found in Appendix B.5.

### 3.1.1 METRIC-MEMORIZATION RISK INDEX

MRI consists of two signals: (i) how similar the model's answer for the rewritten task is to the original ground-truth solution, and (ii) how much performance drops from the original task to its rewritten counterpart.

**(i) Similarity.** For every rewritten task $i \in \mathcal{T}_{\text{rew}}$, where $\mathcal{T}$ refers to a task set, we measure two similarities between the model-generated solution for the rewritten version of task $i$ and the ground-truth solution of its original version:

- Semantic level AST similarity: normalized tree-edit overlap between abstract-syntax trees
- Syntax level edit similarity: $(1 - (\text{Levenshtein distance}/\text{max-len})$, capturing token-level overlap.

Formally, let $\text{AST}_i \in [0, 1]$ denote AST similarity, and let $\text{Edit}_i \in [0, 1]$ denote edit similarity. We combine these scores into a unified similarity score:

$$S_i = \frac{\text{AST}_i + \text{Edit}_i}{2} \tag{3}$$

Because our analysis is corpus-level, we define the mean similarity over all rewritten tasks as:

$$\text{Sim}(\mathcal{T}_{\text{rew}}) = \frac{1}{|\mathcal{T}_{\text{rew}}|} \sum_{i \in \mathcal{T}_{\text{rew}}} S_i, \quad \text{Sim} \in [0, 1]. \tag{4}$$

**(ii) Relative Accuracy Drop for Rewriting.** For a task set $\mathcal{T}$, Pass@1 is reported as $\text{Acc}(\mathcal{T})$. To capture the performance loss induced by semantic rewriting, we define

$$\text{RAD}_{\text{rew}} = \max\left(0, \frac{\text{Acc}(\mathcal{T}_{\text{ori}}) - \text{Acc}(\mathcal{T}_{\text{rew}})}{\text{Acc}(\mathcal{T}_{\text{ori}})}\right), \quad \text{RAD}_{\text{rew}} \in [0, 1]. \tag{5}$$

$\text{RAD}_{\text{rew}} = 0$ when rewriting does not hurt accuracy and increases when it does; the $\max(0, \cdot)$ prevents negative values when the performance on rewritten tasks happen to be better.

**MRI.** Finally, we introduce the MRI, defined as the product of solution-similarity and relative accuracy drop:

$$\text{MRI} = \text{Sim}(\mathcal{T}_{\text{rew}}) \times \text{RAD}_{\text{rew}}, \quad \text{MRI} \in [0, 1]. \tag{6}$$

This multiplicative design let MRI high only when both conditions for harmful memorization hold: (i) the model copies the original solution's surface form (high $\text{Sim}(\mathcal{T}_{\text{rew}})$) and (ii) that copied solution now fails (high $\text{RAD}_{\text{rew}}$), therefore distinguishes harmful memorization from generalization.

**Accounting for low baseline accuracy.** While $\text{RAD}_{\text{rew}}$ is defined as a relative drop in accuracy, it can become numerically large when the baseline Pass@1 on the original tasks is very low. To avoid over-interpreting such cases, we only compute and compare $\text{RAD}_{\text{rew}}$ and MRI for model–benchmark pairs that satisfy a simple competence threshold $\text{Acc}(\mathcal{T}_{\text{ori}}) >= 10\%$ and at least 50 number of correctly solved original tasks. For settings below this threshold, we still report

$\text{Acc}(\mathcal{T}_{\text{ori}})$ and $\text{Acc}(\mathcal{T}_{\text{rew}})$, but mark $\text{RAD}_{\text{rew}}$ and MRI as "N/A" and exclude them from cross-model MRI comparisons. The concrete statistical justification are provided in Appendix D.

We additionally report an instance-level variant of MRI (iMRI), which aggregates similarity and failure on a per-example basis. Detailed definition and results are in Appendix E, iMRI yields very similar trends and model rankings to our original MRI, showing that our findings are robust to both low-baseline effects and the choice of aggregation scheme.

### 3.2 MUTATION AND PARAPHRASE

To differentiate *code rewriting* from **semantic-preserving** perturbation techniques in work evaluating robustness (Chen et al., 2024; 2023; Mastropaolo et al., 2023; Wang et al., 2022), we include *mutation* and *paraphrase*, as reference baselines. These two perturbations reveal if LLM could generate consistent and correct responses under minor surface level changes (Li et al., 2022). *Mutation* and *paraphrase* are adapted in spirit from ReCode's robustness benchmark (Wang et al., 2022).

**Mutation.** To assess whether LLMs are robust to superficial textual noise, mutation evolution applies small perturbations—such as word-scrambling, random-capitalization, and character-noising—that preserve the underlying problem semantics. Formally, let $x \in T$ denote the original problem prompt in the text space $T$. Mutation evolution applies a perturbation function $\epsilon_1 : T \to T$ such that the mutated prompt $x_{mut} = \epsilon_1(x)$ preserves the original semantics:

$$x_{mut} = \epsilon_1(x), \quad x, x_{mut} \in T \tag{7}$$

where $\epsilon_1$ injects textual noise without altering the problem's underlying meaning.

**Paraphrase.** Paraphrase evolution aims to evaluate whether LLMs can generalize to diverse surface realizations of the same problem. In this setting, prompts are reworded textual expression but preserve semantics. Formally, let $x \in T$ be the original prompt. We define a paraphrasing function $\epsilon_2 : T \to T$ such that:

$$x_{par} = \epsilon_2(x), \quad x, x_{par} \in T \tag{8}$$

where $x_{par}$ is a semantically equivalent but textually different paraphrase of $x$.

#### 3.2.1 METRIC—ROBUSTNESS RELATIVE ACCURACY DROP

Once we perturb a prompt without changing its semantics, what fraction of previously-solved tasks remain solved? To answer this question and differentiate robustness with memorization, for each semantic-preserving transformation $p \in \{\text{mut}, \text{par}\}$ (mutation/paraphrase), we define the **Robustness Relative Accuracy Drop**:

$$\text{RAD}_p = \max\left(0, \ \frac{\text{Acc}(\mathcal{T}_{\text{ori}}) - \text{Acc}(\mathcal{T}_p)}{\text{Acc}(\mathcal{T}_{\text{ori}})}\right), \quad \text{RAD}_p \in [0, 1]. \tag{9}$$

Here, $\text{Acc}(\cdot)$ denotes Pass@1 on the indicated task set section 3.1.1. $\text{RAD}_p = 0$ (high robustness) when semantic-preserving changes do not hurt accuracy and increases toward 1 as performance degrades (low robustness).

### 3.3 FINE-TUNING METHODS

To investigate the memorization phenomenon, we use the original tasks in MBPP+ and BIG-CODEBENCH for fine-tuning[1] and only evaluated on their rewriting counterparts to decouple the rewriting pipeline from post-training signals. More training details can be found at Appendix J.

---

[1] For clarity, both SFT and PPO are initialized from the same base model and trained independently; PPO is not performed on top of an SFT checkpoint.

### 3.3.1 SUPERVISED FINE-TUNING

Supervised Fine-tuning adapts a pre-trained model to a specific task by training it on a labeled dataset, teaching it to predict the correct label for each input. In our setup, coding problems serve as the inputs, while code solutions act as the corresponding labels. However, overfitting occurs when the model fits the training data too closely, reducing its ability to generalize to unseen tasks. This is typically indicated by a rise in validation loss where model begin to memorize training examples. Therefore, we distinguish between early-stage and late-stage memorization by the checkpoint where the loss on the validation set begins to increase. **We select such checkpoint for evaluation to distinguish memorization from overfitting**.

### 3.3.2 REINFORCEMENT LEARNING

Reinforcement Learning enhances fine-tuning efficiency. A leading method is Proximal Policy Optimization (PPO)(Schulman et al., 2017), which alternates between sampling data through interaction with the environment, and optimizing a "surrogate" objective function using stochastic gradient ascent. We utilize the same model architecture for the actor, critic, and reference models for simplicity, and define the reward function based on the correctness of the generated code. Compared to other reinforcement learning methods like DPO (Rafailov et al., 2024), we suggest that using accuracy as the reward function offers a more direct and efficient optimization path. **We evaluate using the checkpoint that achieves the highest validation reward**.

## 4 EXPERIMENT SETUP

### 4.1 DATASETS

We conduct our evaluation on two widely-adopted code generation benchmarks: MBPP+ (Liu et al., 2023) and BIGCODEBENCH (Zhuo et al., 2024).

**Dataset Statistics.** MBPP+ contains 378 tasks, and BIGCODEBENCH comprises 1140 tasks. We use 4:1 train/test split for fine-tuning. For models without fine-tuning, we use the **complete set** of benchmark tasks for evaluation. For models that undergo SFT and PPO, we train on the **training split** and evaluate on the **test split**. Due to the small size of MBPP+ test split ($n = 78$), estimation on this split may be imprecise and directional, we use BIGCODEBENCH to explore the impact of fine-tuning strategies on memorization.

**Task Generation.** For each original task, we generate one perturbed variant for each of *code rewriting, mutation* and *paraphrase*. More about the generation process is given in Appendix K.

### 4.2 MODELS

In this paper, we conduct the scale-up experiments on Qwen-2.5 series (Hui et al., 2024), Qwen-2.5-Coder series (Qwen et al., 2025), Llama-3.1 series (Dubey et al., 2024) and Llama-4 series (AI, 2024). For fine-tuning, we choose Qwen-2.5-7B, and Qwen-2.5-Coder-7B. All training and inference were conducted on a server equipped with 4 NVIDIA A100 GPUs (80GB), with a total computational budget of approximately 40 GPU hours, using PyTorch and HuggingFace Transformers.

## 5 RESULT ANALYSIS

### 5.1 MEMORIZATION ANALYSIS ON INSTRUCT MODELS

**Harmful memorization does not increase with larger models and in many cases decreases as they scale.** Across Qwen2.5 Instruct and its Coder Instruct families, scaling is *associated with* lower $\text{RAD}_{\text{rew}}$ and hence lower MRI (Table 1 and Figure 2). On MBPP+, Qwen-Instruct's MRI falls from 0.0722 at 0.5B to 0.0113 at 14B, reaching 0.0000 at 32B, driven by a decrease in $\text{RAD}_{\text{rew}}$ from $0.2697 \rightarrow 0.0414 \rightarrow 0.0000$. A similar pattern holds for Qwen-Coder (MRI $0.0615 \rightarrow 0.0313 \rightarrow 0.0354$ as $\text{RAD}_{\text{rew}}$ goes from $0.2663 \rightarrow 0.0896 \rightarrow 0.0993$). Notably, $\text{Sim}(\mathcal{T}_{\text{rew}})$ does not uniformly decline with scale (e.g., Qwen-Instruct: 0.2678 at 0.5B $\rightarrow$ 0.3369 at 32B), indicating

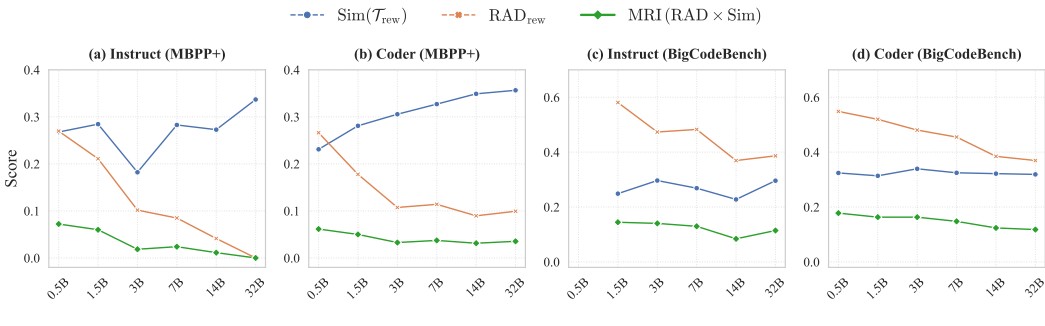

Figure 2: Scaling trends in MRI across Qwen-2.5 Instruct vs. Coder. **(a)-(b)** show results on MBPP+, while **(c)-(d)** show results on BIGCODEBENCH. The Qwen-2.5 Instruct-0.5B point on BIGCODEBENCH is omitted because its baseline Pass@1 falls below our competence threshold (see Appendix D).

that larger models may continue to reuse surface patterns; however, because their failures under semantic shifts largely vanish, such reuse is not harmful and thus produces much lower MRI.

| Model | MBPP+ | | | | BigCodeBench | | | |
|---|---|---|---|---|---|---|---|---|
| | $\text{Acc}(\mathcal{T}_{\text{ori}})$ (↑) | $\text{Sim}(\mathcal{T}_{\text{rew}})$ (↓) | $\text{RAD}_{\text{rew}}$ (↓) | **MRI** (↓) | $\text{Acc}(\mathcal{T}_{\text{ori}})$ (↑) | $\text{Sim}(\mathcal{T}_{\text{rew}})$ (↓) | $\text{RAD}_{\text{rew}}$ (↓) | **MRI** (↓) |
| Qwen-2.5 0.5B-Instruct | 0.4021 | 0.2678 | 0.2697 | 0.0722 | 0.0947 | 0.2648 | N/A | N/A |
| Qwen-2.5-1.5B-Instruct | 0.5767 | 0.2845 | 0.2110 | 0.0600 | 0.2281 | 0.2487 | 0.5808 | 0.1444 |
| Qwen-2.5-3B-Instruct | 0.6243 | 0.1822 | 0.1017 | 0.0185 | 0.3132 | 0.2965 | 0.4734 | 0.1403 |
| Qwen-2.5-7B-Instruct | 0.6852 | 0.2828 | 0.0849 | 0.0240 | 0.3798 | 0.2686 | 0.4827 | 0.1296 |
| Qwen-2.5-14B-Instruct | 0.7037 | 0.2728 | 0.0414 | 0.0113 | 0.3895 | 0.2277 | 0.3694 | 0.0841 |
| Qwen-2.5-32B-Instruct | 0.7513 | 0.3369 | 0.0000 | 0.0000 | 0.4404 | 0.2959 | 0.3865 | 0.1143 |
| Qwen-2.5-coder-0.5B-Instruct | 0.4471 | 0.2310 | 0.2663 | 0.0615 | 0.1088 | 0.3242 | 0.5484 | 0.1778 |
| Qwen-2.5-coder-1.5B-Instruct | 0.5952 | 0.2810 | 0.1778 | 0.0500 | 0.2465 | 0.3137 | 0.5196 | 0.1630 |
| Qwen-2.5-coder-3B-Instruct | 0.6402 | 0.3056 | 0.1074 | 0.0328 | 0.3579 | 0.3393 | 0.4804 | 0.1630 |
| Qwen-2.5-coder-7B-Instruct | 0.7196 | 0.3271 | 0.1140 | 0.0373 | 0.4088 | 0.3247 | 0.4549 | 0.1477 |
| Qwen-2.5-coder-14B-Instruct | 0.7381 | 0.3490 | 0.0896 | 0.0313 | 0.4675 | 0.3216 | 0.3846 | 0.1237 |
| Qwen-2.5-coder-32B-Instruct | 0.7725 | 0.3565 | 0.0993 | 0.0354 | 0.4772 | 0.3189 | 0.3695 | 0.1178 |
| Llama-3.1-8B-Instruct | 0.5529 | 0.1486 | 0.0133 | 0.0020 | 0.3079 | 0.2132 | 0.4444 | 0.0947 |
| Llama-3.1-70B-Instruct | 0.6984 | 0.1518 | 0.0000 | 0.0000 | 0.4175 | 0.2404 | 0.3676 | 0.0884 |
| Llama-4-Scout-17B-Instruct (16E) | 0.6614 | 0.1446 | 0.0160 | 0.0023 | 0.4061 | 0.2343 | 0.3909 | 0.0916 |
| Llama-4-Maverick-17B-Instruct (128E) | 0.7751 | 0.2669 | 0.0307 | 0.0082 | 0.4860 | 0.2357 | 0.3953 | 0.0932 |

Table 1: Memorization risk for Qwen-2.5, Llama-3.1 and Llama-4 instruct series. MRI persists in harder tasks (BIGCODEBENCH), as $\text{RAD}_{\text{rew}}$ stays high even as $\text{Sim}(\mathcal{T}_{\text{rew}})$ is comparable.

On BIGCODEBENCH, the effect from scaling up is milder and sometimes non-monotonic. Qwen-Instruct's MRI drops from 0.1444 (1.5B) to 0.0841 (14B) but increases to 0.1143 at 32B, with $\text{RAD}_{\text{rew}}$ trending from $0.5808 \to 0.3694 \to 0.3865$. On the other hand, Qwen-Coder shows a steadier decline ($0.1778 \to 0.1178$ from 0.5B→32B) with relatively flat $\text{Sim}(\mathcal{T}_{\text{rew}})$. Overall, scale reduces memorization primarily by improving resistance to semantic shifts ($\text{RAD}_{\text{rew}}$), while surface-form similarity can remain high. The gains are pronounced on simpler tasks (MBPP+) and partially eroded on harder ones (BIGCODEBENCH); on BIGCODEBENCH, the non-zero MRI is explained by persistently high $\text{RAD}_{\text{rew}}$ with roughly unchanged $\text{Sim}(\mathcal{T}_{\text{rew}})$.

We also evaluate on Llama families (Table 1). While Llama 3.1 exhibit similarly low MRI as scale increases, Llama 4 [2] shows comparable MRI on both dataset. On MBPP+, the MRI from Llama-3.1 (8B/70B) declined in small degree ($0.0020 \to 0.0000$), and Llama-4 models are near zero (0.0023 and 0.0082); on BIGCODEBENCH, Llama-3.1 shifts little ($0.0947 \to 0.0884$), and Llama-4 remains comparably low but non-zero (0.0932 and 0.0916). These results reveal a task-dependency on harmful memorization: on easier problems, larger Llama models effectively drive $\text{RAD}_{\text{rew}} \to 0$ (hence negligible MRI) even when $\text{Sim}(\mathcal{T}_{\text{rew}})$ is moderate, whereas on BIGCODEBENCH the non-zero risk is dominated by persistent $\text{RAD}_{\text{rew}} = 0.4060$ for Llama 3.1 series and 0.3931 for Llama 4 series at similar similarity levels.

---

[2] The two Llama-4 variants we evaluate are MoE models with similar per-token activated compute; their difference is mainly *capacity* (number of experts) rather than dense compute scaling.

**Ablation Studies.** We additionally verify in an extensive set of ablations (Appendix F) that our findings are stable under implementation choices. Varying the parser, AST/edit weightings, and similarity function gives MRI values that are highly correlated with our default definition and preserves the trends.

### 5.1.1 ADDITIONAL FINDINGS

**Harmful memorization declines rapidly on simpler tasks but persists on more difficult ones.** On the introductory-level tasks in MBPP+, memorization risk (MRI) decreases notably as models scale up. For instance, for Qwen-2.5-Instruct's MRI falls from 0.0722 at 0.5B parameters to effectively zero at 32B. Conversely, on the more challenging BIGCODEBENCH, MRI values remain significant even at large scales (0.1178 for Qwen-2.5-32B-Instruct). This discrepancy shows that while larger models better capture underlying semantics changes, they do not completely eliminate memorization, especially in scenarios of challenging tasks that demand deeper reasoning.

**Coder models encourages code reuse but does not substantially increase harmful memorization.** Coder models yield higher $\mathrm{Sim}(\mathcal{T}_{\mathrm{rew}})$ than their instruction-only counterparts. For instance, on BIGCODEBENCH, Qwen-2.5-Coder series scores $\mathbf{0.3237} \pm 0.0086$ vs. $\mathbf{0.2670} \pm 0.0268$ for the instruction-only variant (mean $\pm$ SD over 6 seeds). However, $\mathrm{RAD}_{\mathrm{rew}}$ remains comparable across these variants, translating to only a slight increase in MRI ($\mathbf{0.1367} \pm 0.0224$ vs. $\mathbf{0.1142} \pm 0.0247$, mean $\pm$ SD over 6 seeds). This pattern suggests code-focused pre-training promotes superficial reuse of training data without significantly increase harmful memorization.

**Harmful memorization is driven by specific logic-edit types.** To better understand *which* kinds of semantic changes are responsible for failures, we annotated each rewritten pair with a fine-grained logic change taxonomy, including Component Swap, Data Transformation, Constraint Change, Logic Reversal, Constant/Operator Swap, Workflow Modification, and others (see Appendix G for definitions and examples). Both MBPP+ and BIGCODEBENCH show diverse coverage over these types. On BIGCODEBENCH, Component Swap and Data Transformation account for roughly one third of all changes each, followed by Constraint Change; on MBPP+, the mass is more evenly spread across Constraint Change, Logic Reversal, Data Transformation, Component Swap, and Constant/Operator Swap. For all subsequent analyses we discard logic types with fewer than 50 instances per dataset to avoid unstable estimates.

**MRI varies strongly by logic-change type and is largest when rewrites change constraints or data flow.** We report per-type MRI and $\mathrm{RAD}_{\mathrm{rew}}$ for representative Qwen-2.5 Instruct and Coder variants in Appendix G. On MBPP+, Constraint Change and Data Transformation systematically achieve the highest MRI across both families, whereas Component Swap shows much smaller MRI and even approaches zero for larger models. This pattern is mirrored in $\mathrm{RAD}_{\mathrm{rew}}$: going from 0.5B to 14B reduces, but does not entirely remove, the accuracy drop under Constraint Change and Data Transformation, while Logic Reversal becomes essentially harmless as scale increases. On the more challenging BIGCODEBENCH, all major types exhibit non-trivial MRI, but Data Transformation, Constraint Change, Logic Reversal, and Workflow Modification consistently sit above the overall MRI for many model sizes, and maintain large per-type $\mathrm{RAD}_{\mathrm{rew}}$ even at 14B. Qualitatively, these categories correspond to changes that alter global data flow (e.g., applying transforms before/after key operations), tighten or relax constraints (e.g., loop bounds or regex/SQL conditions), or restructure the workflow rather than merely swapping local operators.

### 5.2 IMPACT OF FINE-TUNING STRATEGIES ON MEMORIZATION

Figure 3 shows notable differences in memorization across different fine-tuning strategies on Qwen-2.5-7B and Qwen-2.5-Coder-7B on BIGCODEBENCH.

**SFT improves accuracy but introduces high memorization risk.** Models fine-tuned via SFT consistently achieve accuracy gains on original tasks. For Qwen-2.5-7B-SFT, accuracy was boosted from $0.3158 \rightarrow 0.3772$ on BIGCODEBENCH and increasing from $0.3684 \rightarrow 0.4079$ on the coder counterpart. However, for both Qwen-2.5-7B-SFT and Qwen-2.5-Coder-7B-SFT, these improvements come with significant increases in memorization, as indicated by much higher MRI scores

(e.g. $0.0799 \rightarrow 0.1747$ for Qwen-2.5-7B-SFT and $0.1392 \rightarrow 0.1921$ on for Qwen-2.5-Coder-7B-SFT). These trends reveals that SFT enhances surface-level accuracy at the expense of genuine generalization.

**PPO balances accuracy improvements and memorization risk.** Across both variants, PPO preserves baseline-level or higher accuracy while sharply reducing memorization risk relative to SFT. On Qwen-2.5-7B, accuracy moves from $0.3158 \rightarrow 0.3509$ (PPO) vs $0.3772$ (SFT), with MRI $0.0799 \rightarrow 0.0795$ (PPO) vs $0.1747$ (SFT); Similar trend was revaled by Qwen-2.5-Coder-7B, where accuracy is $0.3684 \rightarrow 0.3728$ (PPO) vs $0.4079$ (SFT), with MRI $0.1392 \rightarrow 0.1336$ (PPO) vs $0.1921$ (SFT). Overall, PPO yields a better risk–accuracy trade-off by keeping MRI near or below base levels while offering milder accuracy gains, in contrast to SFT's larger accuracy improvements accompanied by substantially higher MRI.

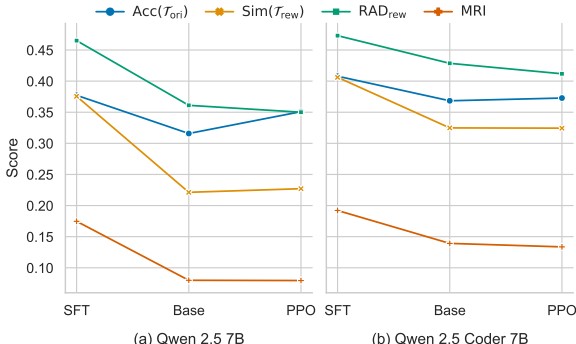

Figure 3: Effect of fine-tuning on Qwen-2.5-7B (base and Coder) on BIGCODEBENCH. SFT raises $\text{Acc}(\mathcal{T}_{\text{ori}})$ but also increases $\text{Sim}(\mathcal{T}_{\text{rew}})$ and $\text{RAD}_{\text{rew}}$, inflating MRI; PPO preserves or modestly improves accuracy while keeping $\text{RAD}_{\text{rew}}$ low, yielding a better risk–accuracy trade-off. Checkpoints selected for SFT and PPO follows rules in subsection 3.3. Dataset statistics can be found in subsection 4.1

**Implications for Fine-Tuning Decisions.** The choice between SFT and reinforcement-based approaches such as PPO is ultimately determined by how one prioritizes the trade-off between accuracy and memorization risk. If maximizing accuracy is the priority and the risks associated with memorization are acceptable, then SFT remains the optimal strategy. However, in settings where generalization and minimizing memorization risk are critical, PPO provides a better balance by offering modest accuracy improvements while considerably reducing memorization.

Beyond comparing training strategies, MRI also provides a practical diagnostic for *checkpoint selection*. As shown in Table 2, continuing SFT beyond the best-validation-loss checkpoint yields only small accuracy gains but can introduce a disproportionate increase in MRI (e.g., a 1.7% accuracy gain vs. a 19% MRI increase for Qwen-2.5-Coder-7B). This indicates a shift from genuine generalization toward harmful memorization. A simple and reproducible recipe thus is: among saved checkpoints with comparable validation loss or accuracy, prefer

| Model | Stage (loss) | $\text{Acc}(\mathcal{T}_{\text{ori}})$ ($\uparrow$) | MRI |
|---|---|---|---|
| Qwen-2.5-7B | Early (0.324) | 35.96% | 0.1540 |
| | Best-Val (0.302) | 37.72% | 0.1733 |
| | Late (0.326) | 39.47% | 0.1795 |
| Coder-7B | Early (0.325) | 35.96% | 0.1537 |
| | Best-Val (0.292) | 40.79% | 0.1864 |
| | Late (0.318) | 42.54% | 0.2216 |

Table 2: Changes in original-task accuracy and MRI during SFT training on BIGCODEBENCH. Accuracy increases steadily from Early to Best-Val to Late, but MRI grows more sharply

those lying on a better accuracy–MRI trade-off, and consider MRI to take precedence over marginal accuracy improvements when those improvements come at considerable memorization cost. We view this as a lightweight way for practitioners to incorporate MRI into training decisions without modifying the training objective itself.

## 5.3 ROBUSTNESS TO SEMANTIC-PRESERVING PERTURBATIONS

We differentiate from memorization by using two semantic-preserving perturbations—*mutation* and *paraphrase*—as reference baselines, and we quantify consistency under these baselines with RAD; our primary analysis remains memorization via semantics-altering rewriting and MRI.

**Mutation remains more challenging** Across BIGCODEBENCH, *mutation* induces a moderate RAD while *paraphrase* exhibits a milder influence on model accuracy: averaged over all models, $\text{RAD}_{\text{mut}} = 0.20 \pm 0.12$ and $\text{RAD}_{\text{par}} = 0.06 \pm 0.04$, compared to a much larger semantics-altering rewriting drop of $\text{RAD}_{\text{rew}} = 0.46 \pm 0.09$. On MBPP+, both mutation and rewriting are modest

$(\text{RAD}_{\text{mut}} = 0.10 \pm 0.08, \text{RAD}_{\text{rew}} = 0.10 \pm 0.09)$, and paraphrase is essentially invariant $(\text{RAD}_{\text{par}} = 0.01 \pm 0.01)$. These results confirm that our primary memorization analysis (via rewriting and MRI) targets a qualitatively different—and much stronger—source of variance than same-semantics perturbations.

**Scaling helps robustness to mutation; coder models show higher sensitivity to mutation on harder tasks.** Mutation accuracy drop decreases with model size on both benchmarks, while paraphrase remains near-zero with small fluctuations at the high end. On BIGCODEBENCH, coder models are the most mutation-sensitive (e.g., Qwen-2.5-coder avg. $\text{RAD}_{\text{mut}} = 0.25 \pm 0.12$) versus their instruction counterparts $(0.20 \pm 0.13)$, with Llama families lower still (Llama-3.1 $\text{RAD}_{\text{mut}} = 0.1050$, Llama-4 $\text{RAD}_{\text{mut}} = 0.1206$). On MBPP+, absolute drops are smaller for all families; Llama-4 shows the lowest RAD under mutation $(\text{RAD}_{\text{mut}} = 0.0578)$. Paraphrase occasionally yields zero or even negative drops (i.e., accuracy improves), consistent with minor wording changes sometimes helping the model parse constraints.

| Model | MBPP+ | | | | BigCodeBench | | | |
|---|---|---|---|---|---|---|---|---|
| | Acc($\mathcal{T}_{\text{ori}}$) ($\uparrow$) | RAD$_{\text{mut}}$ ($\downarrow$) | RAD$_{\text{par}}$ ($\downarrow$) | RAD$_{\text{rew}}$ ($\downarrow$) | Acc($\mathcal{T}_{\text{ori}}$) ($\uparrow$) | RAD$_{\text{mut}}$ ($\downarrow$) | RAD$_{\text{par}}$ ($\downarrow$) | RAD$_{\text{rew}}$ ($\downarrow$) |
| Qwen-2.5 0.5B-Instruct | 0.4021 | 0.2763 | 0.0000 | 0.2697 | 0.0947 | N/A | N/A | N/A |
| Qwen-2.5-1.5B-Instruct | 0.5767 | 0.2248 | 0.0000 | 0.2110 | 0.2281 | 0.2115 | 0.0000 | 0.5808 |
| Qwen-2.5-3B-Instruct | 0.6243 | 0.1144 | 0.0000 | 0.1017 | 0.3132 | 0.1989 | 0.0448 | 0.4734 |
| Qwen-2.5-7B-Instruct | 0.6852 | 0.0463 | 0.0000 | 0.0849 | 0.3798 | 0.1409 | 0.0878 | 0.4827 |
| Qwen-2.5-14B-Instruct | 0.7037 | 0.0338 | 0.0000 | 0.0414 | 0.3895 | 0.0631 | 0.0608 | 0.3694 |
| Qwen-2.5-32B-Instruct | 0.7513 | 0.0106 | 0.0000 | 0.0000 | 0.4404 | 0.1474 | 0.0757 | 0.3865 |
| *Qwen-2.5-Instruct (mean $\pm$ SD)* | 0.62 $\pm$ 0.12 | 0.12 $\pm$ 0.11 | 0.00 $\pm$ 0.00 | 0.12 $\pm$ 0.10 | 0.31 $\pm$ 0.13 | 0.20 $\pm$ 0.13 | 0.04 $\pm$ 0.04 | 0.49 $\pm$ 0.11 |
| Qwen-2.5-coder-0.5B-Instruct | 0.4471 | 0.2367 | 0.0000 | 0.2663 | 0.1088 | 0.4677 | 0.0000 | 0.5484 |
| Qwen-2.5-coder-1.5B-Instruct | 0.5952 | 0.1378 | 0.0000 | 0.1778 | 0.2465 | 0.2954 | 0.0391 | 0.5196 |
| Qwen-2.5-coder-3B-Instruct | 0.6402 | 0.0909 | 0.0000 | 0.1074 | 0.3579 | 0.2304 | 0.0686 | 0.4804 |
| Qwen-2.5-coder-7B-Instruct | 0.7196 | 0.0662 | 0.0294 | 0.1140 | 0.4088 | 0.1803 | 0.1073 | 0.4549 |
| Qwen-2.5-coder-14B-Instruct | 0.7381 | 0.0394 | 0.0143 | 0.0896 | 0.4675 | 0.1463 | 0.0938 | 0.3846 |
| Qwen-2.5-coder-32B-Instruct | 0.7725 | 0.0171 | 0.0000 | 0.0993 | 0.4772 | 0.1857 | 0.1085 | 0.3695 |
| *Qwen-2.5-Coder-Instruct (mean $\pm$ SD)* | 0.65 $\pm$ 0.12 | 0.10 $\pm$ 0.08 | 0.01 $\pm$ 0.01 | 0.14 $\pm$ 0.07 | 0.34 $\pm$ 0.14 | 0.25 $\pm$ 0.12 | 0.07 $\pm$ 0.04 | 0.46 $\pm$ 0.07 |
| Llama-3.1-8B-Instruct | 0.5529 | 0.1340 | 0.0000 | 0.0133 | 0.3079 | 0.1595 | 0.0513 | 0.4444 |
| Llama-3.1-70B-Instruct | 0.6984 | 0.0795 | 0.0189 | 0.0000 | 0.4175 | 0.0504 | 0.0399 | 0.3676 |
| *Llama-3.1-Instruct (mean)* | 0.6257 | 0.1068 | 0.0095 | 0.0067 | 0.3627 | 0.1050 | 0.0456 | 0.4060 |
| Llama-4-Scout-17B-Instruct (16E) | 0.6614 | 0.0200 | 0.0040 | 0.0160 | 0.4061 | 0.1058 | 0.0670 | 0.3909 |
| Llama-4-Maverick-17B-Instruct (128E) | 0.7751 | 0.0956 | 0.0375 | 0.0307 | 0.4860 | 0.1354 | 0.1119 | 0.3953 |
| *Llama-4-Instruct (mean)* | 0.7183 | 0.0578 | 0.0208 | 0.0234 | 0.4461 | 0.1206 | 0.0894 | 0.3931 |
| **All models (mean $\pm$ SD)** | 0.65 $\pm$ 0.11 | 0.10 $\pm$ 0.08 | 0.01 $\pm$ 0.01 | 0.10 $\pm$ 0.09 | 0.35 $\pm$ 0.12 | 0.20 $\pm$ 0.12 | 0.06 $\pm$ 0.04 | 0.46 $\pm$ 0.09 |

Table 3: Robustness under semantic-preserving *mutation* and *paraphrase* versus semantics-different rewriting. Mutation induces moderate drops; paraphrase is nearly invariant; rewrites are most disruptive—especially on BIGCODEBENCH—suggesting harmful memorization beyond surface-level robustness. The final row reports column-wise unweighted mean $\pm$ sample SD across 16 models.[3]

## 6 CONCLUSION AND FUTURE WORKS

In this paper, we reframed memorization in code generation as (1) exhibit high similarity to the golden solution of original tasks and (2) lead to performance drops under semantically modified variants. We measured such memorization with *code rewriting*—which preserves surface form while changing task semantics—and the Memorization Risk Index (MRI) that multiplies solution similarity with the relative accuracy drop (RAD) under rewriting. This design isolates harmful memorization from benign reuse. Our experiments on MBPP+ and BIGCODEBENCH show: (i) harmful memorization generally decreases with model scale on simpler tasks, (ii) persists more on harder tasks, and (iii) SFT raises accuracy but inflates MRI, while PPO delivers a better risk–accuracy trade-off. Taken together, these findings clarify when errors stem from harmful memorization rather than generalization and motivate the following next steps: (a) mitigation approach: further research is needed for reducing the impact of memorization. (b) evaluation transferability: while our current evaluation metrics are tailored for code generation, exploring their applicability to other domains, such as mathematical reasoning, could provide valuable insights.

---

[3]Mean is the unweighted arithmetic average computed *per column* across models; SD is the sample standard deviation (unbiased, $n-1$ denominator). Values are rounded to two decimals.

ETHICS STATEMENT

Our *code rewriting*, *mutation* and *paraphrase* pipeline is guided by ethical principles to ensure responsible outcomes.

(1) Data: Our dataset is constructed from MBPP+ and BIGCODEBENCH dataset, which guarantees ethical fairness. We actively work to eliminate any harmful or offensive content from the *code rewriting*, *mutation* and *paraphrase* variant datasets to mitigate potential risks.

(2) Responsible Usage and License: The use of the *code rewriting*, *mutation* and *paraphrase* variant datasets is intended solely for evaluating memorization in LLM code generation tasks. We encourage the responsible use of those datasets for educational and scientific purposes, while strongly discouraging any harmful or malicious activities.

REPRODUCIBILITY STATEMENT

To ensure the reproducibility of our work, we have illustrated the experiment details in the appendix, such as task generation prompts in Appendix B, training details in Appendix J and evolved-task generation configurations in Appendix K. For the dataset and code repository, all evolved tasks and the prompts used during generation will be released publicly upon publication, ensuring reproducibility and facilitating future research.

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

# APPENDIX

## A    USE OF LARGE LANGUAGE MODELS (LLMS)

We made limited use of a large language model (OpenAI's GPT-5) during the preparation of this work. Specifically:

- **Task Generation:** GPT-5 was employed to assist in generating tasks for *code rewriting*, *mutation*, and *paraphrase*. The role of the LLM in this context was restricted to providing task generation; all methodological design, filtering, and integration into our pipeline were carried out by the authors.
- **Writing Assistance:** GPT-5 was additionally used as a language aid for correcting grammar and improving clarity in the writing of the manuscript. The substantive content, research ideas, technical contributions, and overall narrative were conceived and written by the authors without reliance on the LLM.

Beyond these two use cases, no part of the research design, analysis, or interpretation depended on LLM assistance.

## B    PROMPTS FOR TASK GENERATION

We provide the full instruction prompts used to generate each evolution variant (mutation, paraphrasing, and code-rewriting) with GPT-5. For each evolution type, the system and user messages are shown as passed to the API.

### B.1    CODE-REWRITING EVOLUTION

---

**System Prompt**

**System:**   You are an experienced python programmer. Your goal is to transforms a given 'coding task prompt' into a new version. Follow the instructions carefully to transform the prompt.

---

**Code-Rewriting Evolution User Prompt**

**User:**

```
Given a coding task description (#The Given Prompt#) and its canonical solution (#Code#),
    perform the following steps:
1. Modify the canonical solution to create #New Code# by altering only ***ONE*** core
    logic or structure. Do not add additional 'if statements' to the code. Avoid
    superficial changes like variable renaming. Ensure the modified code has different
   semantics in a way that ***expected difficulty equivalent to the original problem***.
    Write a #New Entry Point# to the updated code. This function name must be very
    similar or the same as the old entry point, and reflect the modified code's logic
    changes if using #Old Entry Point# could mislead the programmer on the
    #Rewritten Prompt#.
2. Update #The Given Prompt# to create #Rewritten Prompt#. The new prompt must:"
    - Match the original's ***input signature*** exactly, but the output
        format could be different a little bit.
    - Reflect the modified code's logic changes explicitly.
        Retain the original phrasing structure and ***avoid unnecessary rephrasing***
        in a way that the #Rewritten Prompt# syntactically very similar
        to the #The Given Prompt#.
3. If any mismatch arises between new code and new prompt, revise either one
    (without adding more changes) so all constraints in Steps 1-2 are simultaneously
    satisfied.

Format your response exactly as:
```

```
New Code:
[code]

Explanation:
[logic changes]

Rewritten Prompt:
[updated description]

Old Entry Point:
[original function name]

New Entry Point:
[updated function name]
```

## B.2 CODE-REWRITING EVOLUTION LLM JUDGE

### System Prompt

**System:** You are an expert code reviewer. Your task is to evaluate whether an evolved coding task maintains appropriate quality standards in terms of prompt-code alignment and difficulty equivalence.

### Code-Rewriting Evolution LLM Judging Prompt

**User:** Please evaluate the quality of this evolved coding task by analyzing two key aspects:

```
**Original Task:**
Prompt: {original_prompt}
Code: {original_code}
**Evolved Task:**
Prompt: {rewritten_prompt}
Code: {rewritten_code}

**Evaluation Criteria:**
1. **Prompt-Code Alignment**: Does the new prompt accurately describe what the new code
     does?
   - Are the input/output specifications consistent?
   - Does the prompt clearly communicate the expected behavior?
   - Are there any ambiguities or mismatches?

2. **Difficulty Equivalence**: Is the evolved task of similar difficulty to the original?
   - Does it require similar algorithmic thinking?
   - Is the complexity level maintained (not significantly easier or harder)?
   - Does it test similar programming concepts and skills?

**Response Format:**
Provide your evaluation in the following format:

Alignment Score: [1-5, where 5 = perfect alignment, 1 = major misalignment]
Alignment Reasoning: [Brief explanation of why the prompt and code align or don't align]

Difficulty Score: [1-5, where 5 = equivalent difficulty, 3 = acceptable variation, 1 =
     significantly different]
Difficulty Reasoning: [Brief explanation of difficulty comparison]

Overall Recommendation: [ACCEPT/REJECT]
Overall Reasoning: [Brief summary of your decision]
Please be thorough but concise in your evaluation.
```

## B.3   MUTATION EVOLUTION

### System Prompt

**System:**   You are an experienced python programmer.  Your goal is to transforms a given 'coding task prompt' into a new version. Follow the instructions carefully to transform the prompt.

### Mutation Evolution User Prompt

**User:**  Given a coding task description "The Given Prompt" and its canonical solution "Code", perform the following steps:

- X word-scrambling operations
- Y random-capitalization operations
- Z character-noising operations

Definitions (one "operation" = one change):

- **Word scrambling**: choose a single word (alphabetic token) and randomly shuffle its internal letters.
- **Random capitalization**: flip the case of one letter (upper to lower or lower to upper) anywhere in the text.
- **Character noising**: insert, delete, **or** substitute one character (letter, digit, or punctuation).
  Please gives your answers to "Mutation Prompt" without any additional text or explanation.

**Response:**  Format your response as:

```
Mutation Prompt:
[Updated task description]
```

NOTE: The values X, Y, and Z — representing the number of word-scrambling, random-capitalization, and character-noising operations respectively — are automatically computed based on the length of the original prompt.  Specifically, we apply a total of $\approx 4$ noise operations per 5 words.  We first ensure at least one operation of each type is included (i.e., X, Y, Z $\geq$ 1), then randomly distribute the remaining operations among the three types. This strategy ensures a consistent noise budget proportional to the prompt's length while maintaining diversity in corruption types.

## B.4   PARAPHRASING EVOLUTION

### System Prompt

**System:**   You are an experienced python programmer.  Your goal is to transforms a given 'coding task prompt' into a new version. Follow the instructions carefully to transform the prompt.

### Paraphrasing Evolution User Prompt

**User:**  Given a coding-task description "The Given Prompt", produce a paraphrased version called "Paraphrased Prompt".

Guidelines:

1. Keep the task's meaning, requirements, and input/output specifications identical.
2. Refresh the wording: use synonyms, change sentence order, or rephrase clauses to add light linguistic "noise," but do **not** drop or add information.
3. Preserve any code-related tokens (e.g., variable names, file names, I/O examples) exactly as they appear unless the original prompt explicitly marks them as placeholders.
4. Retain the original structural cues—for example, if the prompt begins with 'Write a Python function. . . ', your rewrite should also begin with that instruction, albeit rephrased

Please gives your answers to "Paraphrased Prompt" without any additional text or explanation.
**Response:**  Format your response as:

```
Paraphrased Prompt:
[Updated task description]
```

## B.5 CODE REWRITING SOLUTION VALIDATION

Additionally, we ensured the validity of test cases for all rewritten tasks across both datasets, and validate each rewritten solution by making it pass its corresponding rewritten unit test. For MBPP+, we reuse the official test case inputs and generate the expected outputs using the rewritten ground-truth solutions, ensuring direct comparability. For BigCodeBench, we adopt the procedure outlined in Zhuo et al. (2024), constructing test cases for each rewritten task based on their guidelines to guarantee consistency and correctness. We installed all packages required by both dataset for assessing function correctness.

## C EXAMPLES OF CLEARER PARAPHRASED PROMPTS

### Mbpp/604

**Original Prompt:** Write a function to reverse words separated by spaces in a given string.
**Paraphrased Prompt:** Create a function that takes a string as input and returns the string with all words, which are divided by spaces, reversed in order.

### Mbpp/752

**Original Prompt:** Write a function to find the nth jacobsthal number. https://www.geeksforgeeks.org/jacobsthal-and-jacobsthal-lucas-numbers/ 0, 1, 1, 3, 5, 11, 21, 43, 85, 171, 341, 683, 1365, 2731, ...
**Paraphrased Prompt:** Create a function that computes the nth Jacobsthal number. Refer to https://www.geeksforgeeks.org/jacobsthal-and-jacobsthal-lucas-numbers/ for more information. The sequence begins as follows: 0, 1, 1, 3, 5, 11, 21, 43, 85, 171, 341, 683, 1365, 2731, ...

### Mbpp/753

**Original Prompt:** Write a function to find minimum k records from tuple list. https://www.geeksforgeeks.org/python-find-minimum-k-records-from-tuple-list/ - in this case a verbatim copy of test cases.
**Paraphrased Prompt:** Create a function that retrieves the smallest k elements from a list of tuples. Refer to https://www.geeksforgeeks.org/python-find-minimum-k-records-from-tuple-list/ and use the provided test cases exactly as they are.

## D STABILITY OF RAD AND MRI UNDER LOW BASELINE ACCURACY

The Relative Accuracy Drop (RAD) measures the *fractional* decrease in Pass@1 between the original and rewritten task sets. Although this captures the intended notion of rewriting-sensitivity, it becomes unstable when the baseline accuracy $\text{Acc}(\mathcal{T}_{\text{ori}})$ is extremely low: even a small absolute change can inflate the relative ratio and consequently MRI (defined as the product of similarity and RAD).

To ensure that RAD and MRI reflect meaningful performance changes rather than numerical instability, we compute these metrics only when two conditions hold.

**(1) Sufficient number of correctly solved original tasks.** For a benchmark with $N$ tasks and baseline accuracy $p = \text{Acc}(\mathcal{T}_{\text{ori}})$, the model solves $k = Np$ tasks. We require

$$k \geq M = 50,$$

which keeps the variance of the empirical Pass@1 sufficiently small under a binomial model. For instance, MBPP+ ($N = 378$) requires $p >= 13.2\%$, resulting in a 95% confidence interval of roughly $\pm 3$–4 percentage points; BigCodeBench ($N = 1{,}140$) requires $p >= 4.4\%$ with an even tighter interval. These uncertainties are far smaller than the 5–20 point drops typically observed in our memorization analyses.

**(2) A minimum competence threshold on baseline accuracy.** We additionally require

$$\text{Acc}(\mathcal{T}_{\text{ori}}) \geq 10\%,$$

which prevents the denominator in RAD from becoming so small that modest absolute changes (e.g., a 5-point drop) produce disproportionally large relative decreases (e.g., $> 50\%$).

**Application of the threshold.** When either condition is violated, we still report $\text{Acc}(\mathcal{T}_{\text{ori}})$ and $\text{Acc}(\mathcal{T}_{\text{rew}})$ for transparency, but we mark RAD and MRI as *"N/A (below competence threshold)"* and exclude them from cross-model comparisons. All analyses and figures in the main paper adhere to this stability rule.

These criteria ensure that large RAD values reflect genuine difficulty introduced by semantic rewriting rather than artifacts of low baseline accuracy, and that MRI remains a reliable indicator of harmful memorization behaviour.

## E  INSTANCE-LEVEL MRI (IMRI)

**Definition.** Our corpus-level MRI is defined as the product of the mean similarity between original and rewritten solutions and the overall accuracy drop under rewriting. This aggregate view is convenient for comparison across models and datasets, but it is natural to ask whether similar conclusions hold when similarity and failures are combined at the level of individual instances. To this end, we define an instance-level variant, iMRI, as

$$\text{iMRI} = \frac{1}{|\mathcal{T}_{\text{passed}}|} \sum_{k \in \mathcal{T}_{\text{passed}}} \left( \text{Sim}_k \cdot \text{Drop}_k \right), \tag{10}$$

where $\mathcal{T}_{\text{passed}}$ is the set of tasks that the model solves correctly on the original benchmark, $\text{Sim}_k$ is the similarity between the original and rewritten solution for example $k$, and $\text{Drop}_k \in \{0, 1\}$ indicates whether the model fails on the rewritten task ($\text{Drop}_k = 1$ if the rewrite fails, and 0 otherwise). Conditioning on $\mathcal{T}_{\text{passed}}$ reflects our focus on *harmful memorization as regression*: a failure on the rewrite is only meaningful when the original task was solved correctly in the first place.

**Alignment between MRI and iMRI.** Table 4 reports Pearson correlations between corpus-level MRI and instance-level iMRI across model sizes for Qwen-2.5 Instruct and Coder series on MBPP+ and BIGCODEBENCH. All correlations are very high ($\geq 0.92$), indicating that the two metrics induce essentially the same ordering over models:

- For the Qwen-2.5 Instruct series, the MRI–iMRI correlation is 0.9816 on MBPP+ and 0.9459 on BIGCODEBENCH.
- For the Qwen-2.5 Coder series, the correlation is 0.9211 on MBPP+ and 0.9731 on BIGCODEBENCH.

| Series | MBPP+ | BIGCODEBENCH |
|---|---|---|
| Qwen-2.5 Instruct | 0.9816 | 0.9459 |
| Qwen-2.5 Coder | 0.9211 | 0.9731 |

Table 4: Pearson correlation between corpus-level MRI and instance-level iMRI across model sizes for Qwen-2.5 Instruct and Coder series.

**Scaling trends under iMRI.** Tables 5 and 6 list MRI and iMRI side by side for all Qwen-2.5 Instruct and Coder models on MBPP+ and BIGCODEBENCH. Several observations reflect the main-text trends:

- On MBPP+ (Table 5), both MRI and iMRI *decrease* with scale in the Instruct and Coder series. iMRI values are numerically larger than MRI (as expected, since they average similarity over solved tasks only), but the relative ranking of models is preserved and the overall trend of reduced harmful memorization for larger models remains.

- On BIGCODEBENCH (Table 6), iMRI again tracks MRI closely: both metrics decline from small to medium scales and then plateau or mildly increase for the largest models, reflecting the same behavior discussed in the main text.

Taken together with the correlations in Table 4, these results indicate that corpus-level MRI is a stable summary of harmful memorization: instance-level aggregation yields qualitatively identical conclusions about scaling trends of Instruct vs. Coder models.

| Model | iMRI | MRI |
|---|---|---|
| Qwen-2.5-0.5B-Instruct | 0.1494 | 0.0722 |
| Qwen-2.5-1.5B-Instruct | 0.1078 | 0.0600 |
| Qwen-2.5-3B-Instruct | 0.0490 | 0.0185 |
| Qwen-2.5-7B-Instruct | 0.0650 | 0.0240 |
| Qwen-2.5-14B-Instruct | 0.0500 | 0.0113 |
| Qwen-2.5-32B-Instruct | 0.0325 | 0.0000 |
| Qwen-2.5-coder-0.5B-Instruct | 0.1107 | 0.0615 |
| Qwen-2.5-coder-1.5B-Instruct | 0.0870 | 0.0500 |
| Qwen-2.5-coder-3B-Instruct | 0.0828 | 0.0328 |
| Qwen-2.5-coder-7B-Instruct | 0.0771 | 0.0373 |
| Qwen-2.5-coder-14B-Instruct | 0.0722 | 0.0313 |
| Qwen-2.5-coder-32B-Instruct | 0.0730 | 0.0354 |

Table 5: Instance-level iMRI vs. corpus-level MRI for Qwen-2.5 Instruct and Coder series on MBPP+.

| Model | iMRI | MRI |
|---|---|---|
| Qwen-2.5-0.5B-Instruct | N/A | N/A |
| Qwen-2.5-1.5B-Instruct | 0.1870 | 0.1444 |
| Qwen-2.5-3B-Instruct | 0.1836 | 0.1403 |
| Qwen-2.5-7B-Instruct | 0.1727 | 0.1296 |
| Qwen-2.5-14B-Instruct | 0.1275 | 0.0841 |
| Qwen-2.5-32B-Instruct | 0.1538 | 0.1143 |
| Qwen-2.5-coder-0.5B-Instruct | 0.2609 | 0.1778 |
| Qwen-2.5-coder-1.5B-Instruct | 0.2379 | 0.1630 |
| Qwen-2.5-coder-3B-Instruct | 0.2108 | 0.1630 |
| Qwen-2.5-coder-7B-Instruct | 0.1959 | 0.1477 |
| Qwen-2.5-coder-14B-Instruct | 0.1619 | 0.1237 |
| Qwen-2.5-coder-32B-Instruct | 0.1510 | 0.1178 |

Table 6: Instance-level iMRI vs. corpus-level MRI for Qwen-2.5 Instruct and Coder series on BIGCODEBENCH.

## F    ABLATION STUDIES FOR SIMILARITY

### F.1    SETUP AND EVALUATION PROTOCOL

In this section we study how sensitive the Memorization Risk Index (MRI) is to implementation choices in its similarity component $S_i$. Unless otherwise stated, we use the same models and datasets as in the main experiments: all Qwen-2.5 Instruct and Coder variants evaluated on MBPP+ and BIGCODEBENCH.

For each model and benchmark, we compute MRI under our default definition (section 3.1.1) and under alternative variants that modify:

- the **parser and grammar** used to obtain AST trees,

- the **weighting** between AST-based and edit-based similarity,
- and the **similarity function** itself (token-overlap and embedding-based metrics).

## F.2 PARSER AND GRAMMAR

We performed a comprehensive robustness check over three definitions of similarity:

- **All-nodes AST** We replaced our standard "significant node" parsing with an All-nodes AST grammar. This treats every syntactic element (including non-functional nodes) as significant, testing robustness to parser strictness.
- **Raw Edit Similarity**: We removed the ast.unparse normalization step and computed edit similarity on raw code strings. This tests robustness to formatting and minor textual artifacts.
- **Combined**: We applied both All-nodes AST and Raw Edit similarity simultaneously.

As shown in Table 7 and Table 8, while stricter parsing (All-nodes) naturally yields higher raw similarity scores and shifts the absolute MRI values, the relative ranking of models and trends remain statistically invariant. We calculated the Pearson correlation ($r$) between the original MRI and the "Combined" variant (the most divergent definition). The correlation is consistently $r > 0.92$ on BigCodeBench for both Instruct and Coder model families, while on MBPP+ correlation is consistently $r > 0.99$ This confirms that MRI is not an artifact of the parser choice. The metric is robust to parsers/grammers.

| Model | MRI (original) | MRI (original edit sim & AST all nodes) | MRI (original AST & edit sim raw) | MRI (edit sim raw & AST all nodes) |
|---|---|---|---|---|
| Qwen-2.5 0.5B-Instruct | 0.0722 | 0.0948 | 0.0561 | 0.0787 |
| Qwen-2.5-1.5B-Instruct | 0.0600 | 0.0783 | 0.0455 | 0.0639 |
| Qwen-2.5-3B-Instruct | 0.0185 | 0.0289 | 0.0153 | 0.0256 |
| Qwen-2.5-7B-Instruct | 0.0240 | 0.0321 | 0.0173 | 0.0254 |
| Qwen-2.5-14B-Instruct | 0.0113 | 0.0151 | 0.0082 | 0.0120 |
| Qwen-2.5-32B-Instruct | 0.0000 | 0.0000 | 0.0000 | 0.0000 |
| Qwen-2.5-coder-0.5B-Instruct | 0.0615 | 0.0864 | 0.0453 | 0.0702 |
| Qwen-2.5-coder-1.5B-Instruct | 0.0500 | 0.0660 | 0.0377 | 0.0537 |
| Qwen-2.5-coder-3B-Instruct | 0.0328 | 0.0426 | 0.0232 | 0.0330 |
| Qwen-2.5-coder-7B-Instruct | 0.0373 | 0.0478 | 0.0257 | 0.0362 |
| Qwen-2.5-coder-14B-Instruct | 0.0313 | 0.0394 | 0.0223 | 0.0304 |
| Qwen-2.5-coder-32B-Instruct | 0.0354 | 0.0444 | 0.0249 | 0.0338 |

Table 7: MRI robustness to parser/grammar variants on MBPP+.

## F.3 WEIGHTING BETWEEN AST-BASED AND EDIT-BASED SIMILARITY

We use the edit/AST weights from "AST only" to "edit only" on both MBPP+ and BigCodeBench for Qwen 2.5 Instruct and its coder models (Table 10 and Table 11). Across all these settings, the MRI computed under our original 50/50 weighting has extremely high Pearson correlation with MRI computed under alternative weightings (Table 9). This indicates that reweighting would essentially rescale MRI rather than alter model rankings or our main findings.

## F.4 SIMILARITY FUNCTIONS (TOKEN-OVERLAP AND EMBEDDING-BASED METRICS

We added ablation evaluations across alternative similarity families, including 1–3-gram overlap, Jaccard, MinHash, and cosine code-embedding similarity on both BigCodeBench (Table 14) and MBPP+ (Table 13). We found that MRI is stable across similarity families. Jaccard, MinHash, and 1–3-grams produce MRI values that closely track our original MRI, and even cosine-embedding similarity (deviates most in scale) preserves the relative trend of models (Table 12).

| Model | MRI (original) | MRI (original edit sim & AST all nodes) | MRI (original AST & edit sim raw) | MRI (edit sim raw & AST all nodes) |
|---|---|---|---|---|
| Qwen-2.5 0.5B-Instruct | 0.1740 | 0.2828 | 0.1652 | 0.2739 |
| Qwen-2.5-1.5B-Instruct | 0.1444 | 0.2407 | 0.1570 | 0.2532 |
| Qwen-2.5-3B-Instruct | 0.1403 | 0.2219 | 0.1292 | 0.2108 |
| Qwen-2.5-7B-Instruct | 0.1296 | 0.2181 | 0.1480 | 0.2364 |
| Qwen-2.5-14B-Instruct | 0.0841 | 0.1519 | 0.0968 | 0.1646 |
| Qwen-2.5-32B-Instruct | 0.1143 | 0.1790 | 0.1047 | 0.1694 |
| Qwen-2.5-coder-0.5B-Instruct | 0.1778 | 0.2499 | 0.1410 | 0.2131 |
| Qwen-2.5-coder-1.5B-Instruct | 0.1630 | 0.2388 | 0.1254 | 0.2013 |
| Qwen-2.5-coder-3B-Instruct | 0.1630 | 0.2352 | 0.1323 | 0.2045 |
| Qwen-2.5-coder-7B-Instruct | 0.1477 | 0.2156 | 0.1187 | 0.1866 |
| Qwen-2.5-coder-14B-Instruct | 0.1237 | 0.1833 | 0.0894 | 0.1490 |
| Qwen-2.5-coder-32B-Instruct | 0.1178 | 0.1802 | 0.0992 | 0.1616 |

Table 8: MRI robustness to parser/grammar variants on BigCodeBench.

| Model Series | Avg. Pearson Corr. on MBPP+ | Avg. Pearson Corr. on BigCodeBench |
|---|---|---|
| Qwen-2.5 series | $0.9997 \pm 0.0004$ | $0.9921 \pm 0.0144$ |
| Coder series | $0.9974 \pm 0.0042$ | $0.9780 \pm 0.0367$ |

Table 9: Average Pearson correlation for AST/edit weightings.

## F.5 CONCLUSIONS

These results demonstrate that our main findings are not only suitable for the particular 5:5 AST+edit design. MRI behaves consistently across semantic metrics (AST) and surface-form metrics (edit, n-gram, Jaccard, MinHash and cosine code-embedding).

| Model | MRI (orig 5:5) | MRI (edit only) | MRI (AST only) | MRI (edit:ast=2:8) | MRI (edit:ast=4:6) | MRI (edit:ast=6:4) | MRI (edit:ast=8:2) |
|---|---|---|---|---|---|---|---|
| Qwen-2.5 0.5B-Instruct | 0.0722 | 0.1025 | 0.0420 | 0.0541 | 0.0662 | 0.0783 | 0.0904 |
| Qwen-2.5-1.5B-Instruct | 0.0600 | 0.0832 | 0.0368 | 0.0461 | 0.0554 | 0.0647 | 0.0739 |
| Qwen-2.5-3B-Instruct | 0.0185 | 0.0249 | 0.0122 | 0.0147 | 0.0172 | 0.0198 | 0.0223 |
| Qwen-2.5-7B-Instruct | 0.0240 | 0.0342 | 0.0139 | 0.0179 | 0.0220 | 0.0260 | 0.0301 |
| Qwen-2.5-14B-Instruct | 0.0113 | 0.0161 | 0.0065 | 0.0084 | 0.0103 | 0.0122 | 0.0142 |
| Qwen-2.5-32B-Instruct | 0.0000 | 0.0000 | 0.0000 | 0.0000 | 0.0000 | 0.0000 | 0.0000 |
| Qwen-2.5-coder-0.5B-Instruct | 0.0615 | 0.0905 | 0.0325 | 0.0441 | 0.0557 | 0.0673 | 0.0789 |
| Qwen-2.5-coder-1.5B-Instruct | 0.0500 | 0.0713 | 0.0286 | 0.0372 | 0.0457 | 0.0542 | 0.0627 |
| Qwen-2.5-coder-3B-Instruct | 0.0328 | 0.0472 | 0.0184 | 0.0242 | 0.0300 | 0.0357 | 0.0415 |
| Qwen-2.5-coder-7B-Instruct | 0.0373 | 0.0526 | 0.0219 | 0.0281 | 0.0342 | 0.0403 | 0.0465 |
| Qwen-2.5-coder-14B-Instruct | 0.0313 | 0.0433 | 0.0193 | 0.0241 | 0.0289 | 0.0337 | 0.0385 |
| Qwen-2.5-coder-32B-Instruct | 0.0354 | 0.0490 | 0.0218 | 0.0272 | 0.0327 | 0.0381 | 0.0436 |

Table 10: Ablation study for different AST/edit weightings on MBPP+.

| Model | MRI (orig 5:5) | MRI (edit sim) | MRI (AST sim) | MRI (edit:ast=2:8) | MRI (edit:ast=4:6) | MRI (edit:ast=6:4) | MRI (edit:ast=8:2) |
|---|---|---|---|---|---|---|---|
| Qwen-2.5 0.5B-Instruct | 0.1740 | 0.2745 | 0.0736 | 0.1138 | 0.1540 | 0.1941 | 0.2343 |
| Qwen-2.5-1.5B-Instruct | 0.1444 | 0.2166 | 0.0722 | 0.1011 | 0.1300 | 0.1589 | 0.1878 |
| Qwen-2.5-3B-Instruct | 0.1403 | 0.2158 | 0.0649 | 0.0951 | 0.1252 | 0.1554 | 0.1856 |
| Qwen-2.5-7B-Instruct | 0.1296 | 0.1958 | 0.0635 | 0.0899 | 0.1164 | 0.1429 | 0.1693 |
| Qwen-2.5-14B-Instruct | 0.0841 | 0.1227 | 0.0455 | 0.0609 | 0.0764 | 0.0918 | 0.1073 |
| Qwen-2.5-32B-Instruct | 0.1143 | 0.1711 | 0.0576 | 0.0803 | 0.1030 | 0.1257 | 0.1484 |
| Qwen-2.5-coder-0.5B-Instruct | 0.1778 | 0.2802 | 0.0753 | 0.1163 | 0.1573 | 0.1982 | 0.2392 |
| Qwen-2.5-coder-1.5B-Instruct | 0.1630 | 0.2487 | 0.0772 | 0.1115 | 0.1458 | 0.1801 | 0.2144 |
| Qwen-2.5-coder-3B-Instruct | 0.1630 | 0.2463 | 0.0796 | 0.1130 | 0.1463 | 0.1797 | 0.2130 |
| Qwen-2.5-coder-7B-Instruct | 0.1477 | 0.2192 | 0.0762 | 0.1048 | 0.1334 | 0.1620 | 0.1906 |
| Qwen-2.5-coder-14B-Instruct | 0.1237 | 0.1881 | 0.0593 | 0.0850 | 0.1108 | 0.1366 | 0.1623 |
| Qwen-2.5-coder-32B-Instruct | 0.1178 | 0.1804 | 0.0552 | 0.0803 | 0.1053 | 0.1303 | 0.1554 |

Table 11: Ablation study for different AST/edit weightings on BigCodeBench.

| Model Series | Avg. Pearson Corr. on MBPP+ | Avg. Pearson Corr. on BigCodeBench |
|---|---|---|
| Qwen-2.5 series | $0.9966 \pm 0.0029$ | $0.9671 \pm 0.0163$ |
| Coder series | $0.9752 \pm 0.0420$ | $0.9921 \pm 0.0144$ |

Table 12: Average Pearson correlation across similarity families.

| Model | MRI 1-gram | MRI 2-gram | MRI 3-gram | MRI Jaccard sim | MRI Minhash sim | MRI Cosine embed sim |
|---|---|---|---|---|---|---|
| Qwen-2.5 0.5B-Instruct | 0.0873 | 0.0404 | 0.0206 | 0.0712 | 0.0694 | 0.2126 |
| Qwen-2.5-1.5B-Instruct | 0.0704 | 0.0337 | 0.0177 | 0.0568 | 0.0552 | 0.1682 |
| Qwen-2.5-3B-Instruct | 0.0265 | 0.0123 | 0.0064 | 0.0224 | 0.0221 | 0.0800 |
| Qwen-2.5-7B-Instruct | 0.0282 | 0.0132 | 0.0068 | 0.0224 | 0.0219 | 0.0678 |
| Qwen-2.5-14B-Instruct | 0.0135 | 0.0065 | 0.0033 | 0.0107 | 0.0106 | 0.0330 |
| Qwen-2.5-32B-Instruct | 0.0000 | 0.0000 | 0.0000 | 0.0000 | 0.0000 | 0.0000 |
| Qwen-2.5-coder-0.5B-Instruct | 0.0764 | 0.0339 | 0.0167 | 0.0640 | 0.0612 | 0.2110 |
| Qwen-2.5-coder-1.5B-Instruct | 0.0594 | 0.0285 | 0.0151 | 0.0488 | 0.0469 | 0.1424 |
| Qwen-2.5-coder-3B-Instruct | 0.0372 | 0.0181 | 0.0095 | 0.0296 | 0.0286 | 0.0865 |
| Qwen-2.5-coder-7B-Instruct | 0.0408 | 0.0212 | 0.0123 | 0.0321 | 0.0314 | 0.0922 |
| Qwen-2.5-coder-14B-Instruct | 0.0349 | 0.0192 | 0.0117 | 0.0274 | 0.0268 | 0.0730 |
| Qwen-2.5-coder-32B-Instruct | 0.0387 | 0.0219 | 0.0137 | 0.0301 | 0.0297 | 0.0808 |

Table 13: Ablation study across similarity families on MBPP+.

| Model | MRI 1-gram | MRI 2-gram | MRI 3-gram | MRI Jaccard sim | MRI Minhash sim | MRI Cosine embed sim |
|---|---|---|---|---|---|---|
| Qwen-2.5 0.5B-Instruct | 0.3245 | 0.2302 | 0.1823 | 0.2548 | 0.2586 | 0.5986 |
| Qwen-2.5-1.5B-Instruct | 0.2705 | 0.1933 | 0.1521 | 0.2138 | 0.2187 | 0.5245 |
| Qwen-2.5-3B-Instruct | 0.2483 | 0.1827 | 0.1453 | 0.1955 | 0.1987 | 0.4353 |
| Qwen-2.5-7B-Instruct | 0.2401 | 0.1761 | 0.1395 | 0.1902 | 0.1932 | 0.4397 |
| Qwen-2.5-14B-Instruct | 0.1690 | 0.1229 | 0.0970 | 0.1345 | 0.1387 | 0.3316 |
| Qwen-2.5-32B-Instruct | 0.2053 | 0.1533 | 0.1226 | 0.1628 | 0.1650 | 0.3547 |
| Qwen-2.5-coder-0.5B-Instruct | 0.2825 | 0.2018 | 0.1597 | 0.2209 | 0.2222 | 0.5056 |
| Qwen-2.5-coder-1.5B-Instruct | 0.2813 | 0.2094 | 0.1693 | 0.2234 | 0.2257 | 0.4778 |
| Qwen-2.5-coder-3B-Instruct | 0.2641 | 0.1977 | 0.1583 | 0.2102 | 0.2109 | 0.4444 |
| Qwen-2.5-coder-7B-Instruct | 0.2451 | 0.1838 | 0.1476 | 0.1959 | 0.1979 | 0.4193 |
| Qwen-2.5-coder-14B-Instruct | 0.2099 | 0.1578 | 0.1264 | 0.1668 | 0.1685 | 0.3554 |
| Qwen-2.5-coder-32B-Instruct | 0.1994 | 0.1491 | 0.1188 | 0.1579 | 0.1590 | 0.3410 |

Table 14: Ablation study across similarity families on BigCodeBench.

# G  ANALYSIS BY LOGIC CHANGE TYPE

**Logic change types.**  To better understand *which* semantic edits drive memorization, we categorize each semantics-altering rewrite into one of the following nine logic change types:

- **Component Swap:** Swapping a function, class, or method while keeping the surrounding structure fixed (e.g., sum → max, np.std → np.var).
- **Data Transformation:**  Transforming the data *before* applying the main logic (e.g., np.log(data), data.T, data.pct_change(), tokenization or normalization steps).
- **Constraint Change:** Modifying rules, parameters, or logical conditions (e.g., loop stride in range(..., 2), bounds in inequalities, regex patterns, or SQL UNIQUE constraints).
- **Logic Reversal:**  Inverting the decision or ranking logic (e.g., max → min, sorted(...) → sorted(..., reverse=True), swapping the roles of x and y in comparisons).
- **Operator Change:** Swapping a single arithmetic or logical operator (e.g., ^ → +, == → !=).
- **Operation Order Swap:** Reordering two or more operations with the same components (e.g., applying blur then gray vs. gray then blur).
- **Constant Swap:** Changing hard-coded constants or literal values (e.g., switching between lowercase and UPPERCASE, or adjusting thresholds).
- **Workflow Modification:** Structural changes to the overall pipeline, such as inserting or removing processing stages (e.g., adding an extra zlib compression step or altering nested outputs).
- **Other:** Any semantics-altering edit that does not clearly fall into the above categories.

**Coverage across benchmarks.**  Table 15 reports the distribution of logic change types across BIGCODEBENCH and MBPP+. On BIGCODEBENCH, the changes are dominated by **Component Swap** (34.5%) and **Data Transformation** (33.8%), followed by **Constraint Change** (16.2%), with the remaining types each accounting for less than 5% of all instances. On MBPP+, the distribution is more balanced: **Constraint Change** (22.8%), **Logic Reversal** (18.0%), **Data Transformation** (15.3%), **Component Swap** (13.8%), and **Constant/Operator Swap** (9–10%) all contribute substantially. To avoid unstable estimates, all per-type statistics in this appendix exclude logic-change categories with fewer than 50 samples in a given dataset.

**Per-type RAD on MBPP+.**  Table 16 shows accuracy on original tasks $\mathrm{Acc}(\mathcal{T}_{\mathrm{ori}})$, rewritten tasks $\mathrm{Acc}(\mathcal{T}_{\mathrm{rew}})$, and the resulting accuracy drop RAD (all for semantics-altering rewrites, i.e., $\mathrm{RAD}_{\mathrm{rew}}$) for four Qwen-2.5 variants on MBPP+. Several patterns emerge:

- For both Instruct and Coder families, **Constraint Change** exhibits the largest RAD at both 0.5B and 14B, indicating that models are particularly brittle when loop bounds, thresholds, or other constraints are modified.
- **Data Transformation** also induces substantial drops (25–35% at 0.5B; 7–14% at 14B), showing the difficulty of tracking changes to data pre-processing or representation.
- **Component Swap** has comparatively *small* RAD and becomes almost benign at 14B, while **Logic Reversal** transitions from a sizeable drop at 0.5B to nearly no drop (or even slight improvements) at 14B.

These results show that, on the simpler MBPP+ benchmark, harmful memorization is not uniform across edit types: scaling primarily reduces vulnerability to Constraint Change and Data Transformation, while simple component/logic swaps become much easier for larger models to handle.

**Per-type RAD on BigCodeBench.**  Table 17 reports the same statistics for BIGCODEBENCH. Compared to MBPP+, all major logic-change types now induce much larger drops:

- At 0.5B, **Constraint Change**, **Data Transformation**, **Logic Reversal**, and **Workflow Modification** all yield extremely high RAD (often above 70% for Qwen-2.5-0.5B-Instruct), confirming that models struggle when rewrites modify global data flow or constraints on this more challenging benchmark.

- Scaling to 14B substantially improves Acc($\mathcal{T}_{\text{ori}}$), but RAD *remains high* for Data Transformation, Constraint Change, and Workflow Modification, with per-type drops still in the 30–45% range.

Thus, even at larger scales, BIGCODEBENCH exposes persistent brittleness when the rewrite changes how data are transformed, rather than merely swapping local components.

**MRI per logic-change type.** Table 18 and Table 19 aggregate these drops into MRI values stratified by logic-change type. On MBPP+ (Table 18), **Constraint Change** and **Data Transformation** consistently exhibit the highest per-type MRI across both Instruct and Coder series, whereas **Component Swap** has much smaller MRI and often approaches zero for larger models. On BIG-CODEBENCH (Table 19), all five major types (**Component Swap**, **Constraint Change**, **Data Transformation**, **Logic Reversal**, **Workflow Modification**) show substantial and relatively flat MRI across scales, with **Data Transformation**, **Constraint Change**, and **Workflow Modification** frequently attaining the highest per-type MRI.

| Logic Change Type | bigcodebench Count | bigcodebench % | mbppplus Count | mbppplus % |
|---|---|---|---|---|
| Total | 1140 | 100.00% | 378 | 100.00% |
| Component Swap | 393 | 34.47% | 52 | 13.76% |
| Data Transformation | 385 | 33.77% | 58 | 15.34% |
| Constraint Change | 185 | 16.23% | 86 | 22.75% |
| Logic Reversal | 50 | 4.39% | 68 | 17.99% |
| Workflow Modification | 50 | 4.39% | 10 | 2.65% |
| Constant Swap | 37 | 3.25% | 36 | 9.52% |
| Operation Order Swap | 20 | 1.75% | 12 | 3.17% |
| Operator Change | 18 | 1.58% | 36 | 9.52% |
| Other | 2 | 0.18% | 20 | 5.29% |

Table 15: Code Rewriting logic change type statistics for `bigcodebench` and `mbppplus`.

| Logic Change Type | qwen-2.5-coder-0.5b-instruct | | | | qwen-2.5-coder-14b-instruct | | | |
|---|---|---|---|---|---|---|---|---|
| | Total | Acc($\mathcal{T}_{\text{ori}}$) (↑) | Acc($\mathcal{T}_{\text{rew}}$) (↑) | RAD (↓) | Total | Acc($\mathcal{T}_{\text{ori}}$) (↑) | Acc($\mathcal{T}_{\text{rew}}$) (↑) | RAD (↓) |
| Component Swap | 52 | 48.1% | 44.2% | 8.0% | 52 | 75.0% | 71.2% | 5.1% |
| Constraint Change | 86 | 46.5% | 30.2% | 35.0% | 86 | 73.3% | 54.7% | 25.4% |
| Data Transformation | 58 | 55.2% | 41.4% | 25.0% | 58 | 72.4% | 62.1% | 14.3% |
| Logic Reversal | 68 | 42.6% | 35.3% | 17.2% | 68 | 72.1% | 76.5% | 0.0% |
| **MRI (all nine types)** | – | – | – | **0.0615** | – | – | – | **0.0313** |
| | qwen-2.5-0.5b-instruct | | | | qwen-2.5-14b-instruct | | | |
| Component Swap | 52 | 44.2% | 40.4% | 8.7% | 52 | 75.0% | 71.2% | 5.1% |
| Constraint Change | 86 | 39.5% | 24.4% | 38.2% | 86 | 64.0% | 48.8% | 23.6% |
| Data Transformation | 58 | 50.0% | 32.8% | 34.5% | 58 | 70.7% | 65.5% | 7.3% |
| Logic Reversal | 68 | 39.7% | 26.5% | 33.3% | 68 | 72.1% | 80.9% | 0.0% |
| **MRI (all nine types)** | – | – | – | **0.0722** | – | – | – | **0.0113** |

Table 16: Comparison of logic change type performance between four Qwen-2.5 variants on MBPP+. We omit logic change types with fewer than 50 samples to avoid statistical bias. RAD highlighted in red denotes the highest drop.

| Logic Change Type | qwen-2.5-coder-0.5b-instruct | | | | qwen-2.5-coder-14b-instruct | | | |
|---|---|---|---|---|---|---|---|---|
| | Total | Acc($\mathcal{T}_{ori}$) (↑) | Acc($\mathcal{T}_{rew}$) (↑) | RAD (↓) | Total | Acc($\mathcal{T}_{ori}$) (↑) | Acc($\mathcal{T}_{rew}$) (↑) | RAD (↓) |
| Component Swap | 393 | 13.5% | 6.4% | 52.8% | 393 | 46.8% | 29.3% | 37.5% |
| Constraint Change | 185 | 8.6% | 4.9% | 43.8% | 185 | 50.8% | 33.5% | 34.0% |
| Data Transformation | 385 | 10.6% | 3.4% | 68.3% | 385 | 46.8% | 26.8% | 42.8% |
| Logic Reversal | 50 | 8.0% | 6.0% | 25.0% | 50 | 46.0% | 34.0% | 26.1% |
| Workflow Modification | 50 | 10.0% | 6.0% | 40.0% | 50 | 42.0% | 24.0% | 42.9% |
| **MRI (all nine types)** | – | – | – | **0.1778** | – | – | – | **0.1237** |
| Logic Change Type | qwen-2.5-0.5b-instruct | | | | qwen-2.5-14b-instruct | | | |
| | Total | Acc($\mathcal{T}_{ori}$) (↑) | Acc($\mathcal{T}_{rew}$) (↑) | RAD (↓) | Total | Acc($\mathcal{T}_{ori}$) (↑) | Acc($\mathcal{T}_{rew}$) (↑) | RAD (↓) |
| Component Swap | 393 | 12.0% | 4.8% | 59.6% | 393 | 38.2% | 26.7% | 30.0% |
| Constraint Change | 185 | 11.9% | 2.7% | 77.3% | 185 | 40.5% | 27.0% | 33.3% |
| Data Transformation | 385 | 8.3% | 2.1% | 75.0% | 385 | 39.7% | 21.8% | 45.1% |
| Logic Reversal | 50 | 8.0% | 2.0% | 75.0% | 50 | 44.0% | 28.0% | 36.4% |
| Workflow Modification | 50 | 8.0% | 2.0% | 75.0% | 50 | 30.0% | 18.0% | 40.0% |
| **MRI (all nine types)** | – | – | – | **0.1740** | – | – | – | **0.0841** |

Table 17: Comparison of logic change type performance between four Qwen-2.5 variants on `BigCodeBench`. We omit logic change types with fewer than 50 samples to avoid statistical bias. RAD highlighted in red denotes the highest drop.

| Model | MRI (per logic change type) | | | | MRI (all nine types) |
|---|---|---|---|---|---|
| | Component Swap | Constraint Change | Data Transformation | Logic Reversal | All |
| Qwen-2.5-0.5B-Instruct | 0.0212 | 0.0929 | 0.0877 | 0.1119 | 0.0722 |
| Qwen-2.5-1.5B-Instruct | 0.0644 | 0.0584 | 0.0592 | 0.0553 | 0.0600 |
| Qwen-2.5-3B-Instruct | 0.0149 | 0.0380 | 0.0224 | 0.0000 | 0.0185 |
| Qwen-2.5-7B-Instruct | 0.0154 | 0.0543 | 0.0284 | 0.0072 | 0.0240 |
| Qwen-2.5-14B-Instruct | 0.0139 | 0.0573 | 0.0172 | 0.0000 | 0.0113 |
| Qwen-2.5-32B-Instruct | 0.0000 | 0.0000 | 0.0000 | 0.0073 | 0.0000 |
| *Mean ± Std* | *0.02 ± 0.02* | *0.05 ± 0.03* | *0.04 ± 0.03* | *0.03 ± 0.05* | *—* |
| Qwen-2.5-coder-0.5B-Instruct | 0.0195 | 0.0738 | 0.0480 | 0.0483 | 0.0615 |
| Qwen-2.5-coder-1.5B-Instruct | 0.0451 | 0.0465 | 0.0752 | 0.0284 | 0.0500 |
| Qwen-2.5-coder-3B-Instruct | 0.0000 | 0.0380 | 0.0406 | 0.0079 | 0.0328 |
| Qwen-2.5-coder-7B-Instruct | 0.0000 | 0.0382 | 0.0622 | 0.0189 | 0.0373 |
| Qwen-2.5-coder-14B-Instruct | 0.0189 | 0.0828 | 0.0372 | 0.0000 | 0.0313 |
| Qwen-2.5-coder-32B-Instruct | 0.0000 | 0.0786 | 0.0433 | 0.0356 | 0.0354 |
| *Mean ± Std* | *0.01 ± 0.02* | *0.06 ± 0.02* | *0.05 ± 0.01* | *0.02 ± 0.02* | *—* |

Table 18: MRI across logic change types for Qwen-2.5 Instruct and Coder variants on MBPP+. MRI highlighted in red denotes the highest MRI.

| Model | MRI (per logic change type) | | | | | MRI (all types) |
|---|---|---|---|---|---|---|
| | Component Swap | Constraint Change | Data Transformation | Logic Reversal | Workflow Modification | All |
| Qwen-2.5-0.5B-Instruct | 0.1635 | 0.1968 | 0.1951 | 0.2144 | 0.1900 | 0.1740 |
| Qwen-2.5-1.5B-Instruct | 0.1316 | 0.1551 | 0.1539 | 0.1698 | 0.0749 | 0.1444 |
| Qwen-2.5-3B-Instruct | 0.1334 | 0.1442 | 0.1399 | 0.1698 | 0.1558 | 0.1403 |
| Qwen-2.5-7B-Instruct | 0.1178 | 0.1355 | 0.1368 | 0.2118 | 0.1035 | 0.1296 |
| Qwen-2.5-14B-Instruct | 0.0685 | 0.0775 | 0.0998 | 0.0932 | 0.0850 | 0.0841 |
| Qwen-2.5-32B-Instruct | 0.0960 | 0.1130 | 0.1217 | 0.1268 | 0.1335 | 0.1143 |
| *Mean ± Std* | *0.12 ± 0.03* | *0.14 ± 0.04* | *0.14 ± 0.03* | *0.16 ± 0.05* | *0.12 ± 0.04* | *—* |
| Qwen-2.5-coder-0.5B-Instruct | 0.1628 | 0.1302 | 0.2060 | 0.0787 | 0.1203 | 0.1778 |
| Qwen-2.5-coder-1.5B-Instruct | 0.1302 | 0.2207 | 0.1650 | 0.2135 | 0.1742 | 0.1630 |
| Qwen-2.5-coder-3B-Instruct | 0.1385 | 0.1813 | 0.1848 | 0.2111 | 0.0841 | 0.1630 |
| Qwen-2.5-coder-7B-Instruct | 0.1321 | 0.1440 | 0.1581 | 0.1947 | 0.1590 | 0.1477 |
| Qwen-2.5-coder-14B-Instruct | 0.1192 | 0.1110 | 0.1336 | 0.0956 | 0.1353 | 0.1237 |
| Qwen-2.5-coder-32B-Instruct | 0.1029 | 0.1242 | 0.1276 | 0.1191 | 0.1279 | 0.1178 |
| *Mean ± Std* | *0.13 ± 0.02* | *0.15 ± 0.04* | *0.16 ± 0.03* | *0.15 ± 0.06* | *0.13 ± 0.03* | *—* |

Table 19: MRI across logic change types for Qwen-2.5 Instruct and coder variants on Big-CodeBench. MRI highlighted in red denotes the highest MRI.

## H EXAMPLES OF REGRESSED TASKS

We randomly selected 5 tasks from each of MBPP+ and BigCodeBench that PASSED in original but FAILED in code_rewriting from the evaluation results in Qwen2.5-Coder-32B-Instruct. For each task, we provide

- Original task prompt and its canonical solution
- Code_rewriting task prompt and the rewritten canonical solution
- Alignment and Difficulty analysis from GPT-5 to investigate (1) if the rewritten prompt aligns with its rewritten solution; (2) whether the difficulty of rewritten task align with its original version.

The following case studies confirms that such performance regression is not caused by the higher difficulty on rewritten tasks.

```
MBPP+, original, TASK 99
"""
Write a function to convert the given
decimal number to its binary equivalent,
represented as a string with no leading
zeros.
assert decimal_to_binary(8) == '1000'
"""

def decimal_to_binary(n):
    return bin(n).replace("0b","")
```

```
MBPP+, code_rewriting, TASK 99
"""
Write a function to convert the given
decimal number to its binary equivalent,
represented as a string in reverse bit
order (least significant bit first).
assert decimal_to_binary_rev(8) == '0001'
"""

def decimal_to_binary_rev(n):
    return bin(n)[2:][::-1]
```

```
Alignment & Difficulty Analysis from GPT-5
"""
Alignment: The prompt matches the code's behavior: it asks for the binary string with bits reversed (LSB first), and bin(n)
[2:][::-1] does exactly that. Input/output types are consistent (int → str). The expected example 8 → '0001' aligns with the
behavior. Minor optional clarifications (e.g., assume non-negative integers; reversed string may begin with '0') don't affect
alignment.

Difficulty: Both versions rely on bin() and simple string operations (slice/remove prefix vs. slice+reverse). They require
similar reasoning and have comparable complexity.
"""
```

Figure 4: Example of Task-99 from MBPP+ generated from Qwen2.5-Coder-32B-Instruct that PASSED in original but FAILED in code_rewriting.

```
MBPP+, original, TASK 224
"""
Write a python function to count the number
of set bits (binary digits with value 1) in
a given number.
assert count_Set_Bits(2) == 1
"""

def count_Set_Bits(n):
    return bin(n)[2:].count('1')
```

```
MBPP+, code_rewriting, TASK 224
"""
Write a python function to count the number
of unset bits (binary digits with value 0)
in a given number's binary representation
(without leading zeros).
assert count_Unset_Bits(2) == 1
"""

def count_Unset_Bits(n):
    return bin(n)[2:].count('0')
```

```
Alignment & Difficulty Analysis from GPT-5
"""
Alignment: The code counts '0's in bin(n)[2:], which matches "unset bits … without leading zeros" for non-negative integers
(e.g., 2 -> '10' ⇒ 1). The sample assertion aligns. Also, some contexts define "unset bits" relative to a fixed width (e.g.,
32-bit), but the prompt correctly narrows this by saying "without leading zeros.

Difficulty: Counting zeros instead of ones requires the same approach, complexity (O(k) over bit-length), and Python
knowledge (bin, count). It tests identical concepts with no added complexity.
"""
```

Figure 5: Example of Task-224 from MBPP+ generated from Qwen2.5-Coder-32B-Instruct that PASSED in original but FAILED in code_rewriting.

```
MBPP+, original, TASK 284
"""
Write a function that takes in a list and
element and checks whether all items in the
list are equal to the given element.
assert check_element(["green", "orange",
"black", "white"],'blue')==False
"""

def check_element(list1, element):
    return all(v == element for v in list1)
```

```
MBPP+, code_rewriting, TASK 284
"""
Write a function that takes in a list and
element and checks whether all items in the
list are equal to the given element when
compared case-insensitively (after
converting both to lowercase strings).
assert check_element_ci(["green", "orange",
"black", "white"],'blue')==False
"""

def check_element_ci(list1, element):
    target = str(element).lower()
    return all(str(v).lower() == target for v
in list1)
```

```
Alignment & Difficulty Analysis from GPT-5
"""
Alignment: The prompt explicitly states to compare "case-insensitively (after converting both to lowercase strings)." The
code converts both the target and each list item via str(...).lower() and uses all(...)—exactly matching the described
behavior. No ambiguity remains about non-string inputs because the prompt specifies conversion to strings.

Difficulty: Algorithmic structure is unchanged (a simple all check over a comprehension). The evolved version adds a minor
preprocessing step (str(...).lower()), which doesn't meaningfully change complexity or required concepts.
"""
```

Figure 6: Example of Task-284 from MBPP+ generated from Qwen2.5-Coder-32B-Instruct that PASSED in original but FAILED in code_rewriting.

```
MBPP+, original, TASK 767
"""
Write a python function to count the number
of pairs whose sum is equal to 'sum'. The
funtion gets as input a list of numbers and
the sum,
assert get_pairs_count([1,1,1,1],2) == 6
"""

def get_pairs_count(arr, sum_):
    cnt = 0
    for n in arr:
        cnt += arr.count(sum_ - n)
        if sum_ - n == n:
            cnt -= 1
    return cnt / 2
```

```
MBPP+, code_rewriting, TASK 767
"""
Write a python function to count the number
of ordered pairs (i, j) with i != j whose
sum is equal to 'sum'. The funtion gets as
input a list of numbers and the sum,
assert get_pairs_count_ordered([1,1,1,1],2)
== 12
"""

def get_pairs_count_ordered(arr, sum_):
    cnt = 0
    for n in arr:
        cnt += arr.count(sum_ - n)
        if sum_ - n == n:
            cnt -= 1
    return cnt
```

```
Alignment & Difficulty Analysis from GPT-5
"""
Alignment: The evolved prompt explicitly asks for ordered pairs (i, j) with i ≠ j summing to the target. The code counts, for
each element, the occurrences of its complement and subtracts one when the element equals its complement—precisely excluding
(i, i). It does not divide by 2, so each (i, j) and (j, i) are both counted. The example assert
get_pairs_count_ordered([1,1,1,1],2) == 12 matches this behavior.

Difficulty: Both tasks use the same O(n²) counting approach and require the same conceptual insight (complement counting and
handling self-pairs). The evolved task tests essentially the same skills and complexity.
"""
```

Figure 7: Example of Task-767 from MBPP+ generated from Qwen2.5-Coder-32B-Instruct that PASSED in original but FAILED in code_rewriting.

```
MBPP+, original, TASK 279
"""
Write a function to find the nth decagonal
number.
assert is_num_decagonal(3) == 27
"""

def is_num_decagonal(n):
    return 4 * n * n - 3 * n
```

```
MBPP+, code_rewriting, TASK 279
"""
Write a function to find the nth
dodecagonal number.
assert is_num_dodecagonal(3) == 33
"""

def is_num_dodecagonal(n):
    return 5 * n * n - 4 * n
```

```
Alignment & Difficulty Analysis from GPT-5
"""
Alignment: The prompt ("find the nth dodecagonal number") matches the code's formula 5n^2 - 4n and the test
is_num_dodecagonal(3) == 33.

Difficulty: Computing dodecagonal vs. decagonal numbers uses the same polygonal-number template P_k(n)=\frac{(k-2)n^2-(k-4)n}
{2}. The evolved task requires the same level of formula application and implementation effort as the original.
"""
```

Figure 8: Example of Task-279 from MBPP+ generated from Qwen2.5-Coder-32B-Instruct that PASSED in original but FAILED in code_rewriting.

```
BigCodeBench, original, TASK BigCodeBench/1134
"""
Computes the MD5 hash of each file's content in the specified `source_dir`,
prepends the hash along with a prefix to the original content, and writes the
modified content to new files in the `target_dir`. Existing files with the same
name in `target_dir` are overwritten.
The function should raise the exception for: FileNotFoundError if the source
directory does not exist.
The function should output with:
    list: A list of paths to the newly created files in the `target_dir`, each
with the hash prepended.
You should write self-contained code starting with:
```
import os
import glob
import hashlib
def task_func(source_dir, target_dir, prefix='#Hash: '):
```
"""

import os
import glob
import hashlib
def task_func(source_dir, target_dir, prefix='#Hash: '):
    if not os.path.exists(source_dir):
        raise FileNotFoundError(f"Source directory '{source_dir}' does not
exist.")
    if not os.path.exists(target_dir):
        os.makedirs(target_dir)
    new_files = []
    for file_path in glob.glob(os.path.join(source_dir, '*')):
        with open(file_path, 'r') as infile:
            content = infile.read()
        hash_object = hashlib.md5(content.encode())
        new_file_path = os.path.join(target_dir, os.path.basename(file_path))
        with open(new_file_path, 'w') as outfile:
            outfile.write(f"{prefix}{hash_object.hexdigest()}\n{content}")
        new_files.append(new_file_path)
    return new_files
```

```
BigCodeBench, code_rewriting, TASK BigCodeBench/1134
"""
Computes the MD5 hash of each file's content in the specified `source_dir`,
appends the hash along with a prefix to the original content, and writes the
modified content to new files in the `target_dir`. Existing files with the same
name in `target_dir` are overwritten.
The function should raise the exception for: FileNotFoundError if the source
directory does not exist.
The function should output with:
    list: A list of paths to the newly created files in the `target_dir`, each
with the hash appended.
You should write self-contained code starting with:
```
import os
import glob
import hashlib
def task_func(source_dir, target_dir, prefix='#Hash: '):
```
"""

import os
import glob
import hashlib
def task_func(source_dir, target_dir, prefix='#Hash: '):
    if not os.path.exists(source_dir):
        raise FileNotFoundError(f"Source directory '{source_dir}' does not
exist.")
    if not os.path.exists(target_dir):
        os.makedirs(target_dir)
    new_files = []
    for file_path in glob.glob(os.path.join(source_dir, '*')):
        with open(file_path, 'r') as infile:
            content = infile.read()
        hash_object = hashlib.md5(content.encode())
        new_file_path = os.path.join(target_dir, os.path.basename(file_path))
        with open(new_file_path, 'w') as outfile:
            outfile.write(f"{content}\n{prefix}{hash_object.hexdigest()}")
        new_files.append(new_file_path)
    return new_files
```

```
Alignment & Difficulty Analysis from GPT-5
"""
Alignment: The evolved prompt says to append the prefixed MD5 hash to each file's content and overwrite in target_dir; the code does exactly
that (outfile.write(f"{content}\n{prefix}{hash}")). It raises FileNotFoundError when source_dir is missing and returns a list of created paths
—both matching the prompt. Minor unstated behaviors (newline before hash, text-only reading/encoding, non-recursive glob, and not skipping
subdirectories explicitly) mirror the original and don't contradict the prompt.

Difficulty: The change from "prepend" to "append" is a one-line modification with identical algorithmic steps (iterate files, read, hash,
write, overwrite, return list). Complexity and concepts tested (filesystem ops, hashing, string formatting, error handling) remain the same.
"""
```

Figure 9: Example of Task-1134 from BigCodeBench generated from Qwen2.5-Coder-32B-Instruct that PASSED in original but FAILED in code_rewriting.

```
BigCodeBench, original, TASK BigCodeBench/16
"""
Backup all '.log' files in a specified directory to a tar.gz file and delete the
original files after backup. The backup file is named 'logs_backup.tar.gz' and
placed in the specified backup directory.
The function should raise the exception for: FileNotFoundError: If the specified
directory does not exist.
The function should output with:
    str: The path to the backup file if logs are found, otherwise returns a
message 'No logs found to backup'.
You should write self-contained code starting with:
```
import os
import glob
import subprocess
def task_func(directory, backup_dir='/path/to/backup'):
```
"""

import os
import glob
import subprocess
def task_func(directory, backup_dir='/path/to/backup'):
    if not os.path.exists(directory):
        raise FileNotFoundError(f"Directory '{directory}' not found.")
    log_files = glob.glob(os.path.join(directory, '*.log'))
    if not log_files:
        return "No logs found to backup"
    if not os.path.exists(backup_dir):
        os.makedirs(backup_dir)
    backup_file = os.path.join(backup_dir, 'logs_backup.tar.gz')
    subprocess.call(['tar', '-czvf', backup_file] + log_files)
    for file in log_files:
        os.remove(file)
    return backup_file
```

```
BigCodeBench, code_rewriting, TASK BigCodeBench/16
"""
Backup all '.log' files in a specified directory and its subdirectories
recursively to a tar.gz file and delete the original files after backup. The
backup file is named 'logs_backup.tar.gz' and placed in the specified backup
directory.
The function should raise the exception for: FileNotFoundError: If the specified
directory does not exist.
The function should output with:
    str: The path to the backup file if logs are found, otherwise returns a
message 'No logs found to backup'.
You should write self-contained code starting with:
```
import os
import glob
import subprocess
def task_func(directory, backup_dir='/path/to/backup'):
```
"""

import os
import glob
import subprocess
def task_func(directory, backup_dir='/path/to/backup'):
    if not os.path.exists(directory):
        raise FileNotFoundError(f"Directory '{directory}' not found.")
    log_files = glob.glob(os.path.join(directory, '**', '*.log'), recursive=True)
    if not log_files:
        return "No logs found to backup"
    if not os.path.exists(backup_dir):
        os.makedirs(backup_dir)
    backup_file = os.path.join(backup_dir, 'logs_backup.tar.gz')
    subprocess.call(['tar', '-czvf', backup_file] + log_files)
    for file in log_files:
        os.remove(file)
    return backup_file
```

```
Alignment & Difficulty Analysis from GPT-5
"""
Alignment: The evolved prompt correctly specifies recursive backup of ".log" files and matches the code's use of glob(..., recursive=True),
same I/O contract, same exception, same backup location/name, and deletion of originals.

Difficulty: The change from flat to recursive matching ('**/*.log' with recursive=True) is a minimal extension requiring the same concepts
(path handling, globbing, subprocess usage, file deletion). Algorithmic complexity and required skills are effectively unchanged.
"""
```

Figure 10: Example of Task-16 from BigCodeBench generated from Qwen2.5-Coder-32B-Instruct that PASSED in original but FAILED in code_rewriting.

```
BigCodeBench, original, TASK BigCodeBench/330
"""
Find the k largest numbers in a random-generated list
using heapq.
The function should output with:
    tuple: A tuple containing two lists:
    list[int]: The randomly generated list of integers
with the specified length.
    list[int]: The k largest numbers found using heapq.
You should write self-contained code starting with:
```
import heapq
import random
def task_func(list_length:5, k:int):
```
"""

import heapq
import random
def task_func(list_length:5, k:int):
    numbers = [random.randint(0, 100) for _ in
range(list_length)]
    heapq.heapify(numbers)
    largest_numbers = heapq.nlargest(k, numbers)
    return numbers, largest_numbers
```

```
BigCodeBench, code_rewriting, TASK BigCodeBench/330
"""
Find the k smallest numbers in a random-generated list
using heapq.
The function should output with:
    tuple: A tuple containing two lists:
    list[int]: The randomly generated list of integers
with the specified length.
    list[int]: The k smallest numbers found using
heapq.
You should write self-contained code starting with:
```
import heapq
import random
def task_func(list_length:5, k:int):
```
"""

import heapq
import random
def task_func(list_length:5, k:int):
    numbers = [random.randint(0, 100) for _ in
range(list_length)]
    heapq.heapify(numbers)
    largest_numbers = heapq.nsmallest(k, numbers)
    return numbers, largest_numbers
```

```
Alignment & Difficulty Analysis from GPT-5
"""
Alignment: The code correctly uses heapq.nsmallest to return the k smallest numbers and returns a tuple of two lists as specified. It starts
with the required boilerplate.

Difficulty: Finding k smallest vs. k largest with heapq is symmetric and requires the same concepts and complexity (both effectively O(n log
k)). The algorithmic thinking and library usage are equivalent.
"""
```

Figure 11: Example of Task-330 from BigCodeBench generated from Qwen2.5-Coder-32B-Instruct that PASSED in original but FAILED in code_rewriting.

```
BigCodeBench, original, TASK BigCodeBench/59
"""
Create a word cloud from the text of a Wikipedia page.
The function should output with:
    matplotlib.axes.Axes: The Axes object of the plotted data. Is None if there is
no wikipedia page with the title given as input.
You should write self-contained code starting with:
```
import wikipedia
from wordcloud import WordCloud
import matplotlib.pyplot as plt
def task_func(page_title):
```
"""

import wikipedia
from wordcloud import WordCloud
import matplotlib.pyplot as plt
def task_func(page_title):
    try:
        text = wikipedia.page(page_title).content
    except Exception as e:
        print(f"An error occured: {e}")
        return None
    wordcloud = WordCloud().generate(text)
    plt.figure(figsize=(10, 5))
    plt.imshow(wordcloud, interpolation='bilinear')
    plt.axis('off')
    ax = plt.gca()
    return ax
```

```
BigCodeBench, code_rewriting, TASK BigCodeBench/59
"""
Create a word cloud from the summary text of a Wikipedia page.
The function should output with:
    matplotlib.axes.Axes: The Axes object of the plotted data. Is None if there is
no wikipedia page with the title given as input.
You should write self-contained code starting with:
```
import wikipedia
from wordcloud import WordCloud
import matplotlib.pyplot as plt
def task_func(page_title):
```
"""

import wikipedia
from wordcloud import WordCloud
import matplotlib.pyplot as plt
def task_func(page_title):
    try:
        text = wikipedia.summary(page_title)
    except Exception as e:
        print(f"An error occured: {e}")
        return None
    wordcloud = WordCloud().generate(text)
    plt.figure(figsize=(10, 5))
    plt.imshow(wordcloud, interpolation='bilinear')
    plt.axis('off')
    ax = plt.gca()
    return ax
```

```
Alignment & Difficulty Analysis from GPT-5
"""
Alignment: The code matches the evolved prompt by using wikipedia.summary(page_title) and returns a matplotlib.axes.Axes after plotting the
word cloud. Input/output specs are followed, and the behavior is clear.

Difficulty: The change from full page content to summary does not alter the required concepts or complexity (same libraries, try/except,
generate/plot flow). It's essentially equivalent in effort and algorithmic thinking.
"""
```

Figure 12: Example of Task-59 from BigCodeBench generated from Qwen2.5-Coder-32B-Instruct that PASSED in original but FAILED in code_rewriting.

Figure 13: Example of Task-868 from BigCodeBench generated from Qwen2.5-Coder-32B-Instruct that PASSED in original but FAILED in code_rewriting.

## I  HARMLESS INTERNAL LOGIC MODIFICATIONS

We listed two representative cases from Llama3.1-8B-Instruct where a model has preserved all function and variable names yet altered only the internal implementation (e.g., swapping loops for comprehensions or changing search strategies), illustrating "harmless" syntactic similarity.

### I.1  TASK MBPP/18: LIST COMPREHENSION ¡-¿ EXPLICIT LOOP

**Model Answer in Original Set (list comprehension)**
```python
def remove_dirty_chars(string, second_string):
    second_string_set = set(second_string)
    filtered = [c for c in string if c not in second_string_set]
    return ''.join(filtered)
```

**Model Answer in Rewritten Set (explicit loop)**
```python
def remove_dirty_chars(string, second_string):
    second_string_set = set(second_string)
    result = []
    for c in reversed(string):
        if c not in second_string_set:
            result.append(c)
    return ''.join(reversed(result))
```

Here, the function name 'remove_dirty_chars', parameters, lookup-set variable, and character variable 'c' all remain identical; only the comprehension is replaced by a two-step loop.

### I.2  TASK MBPP/793: REVERSE-INDEX LOOKUP BINARY SEARCH

**Model Answer in Original Set (slice + index)**
```python
def last(arr, x):
    try:
        return len(arr) - 1 - arr[::-1].index(x)
    except ValueError:
        return -1
```

**Model Answer in Rewritten Set (binary search)**
```python
def last(arr, x):
    low, high = 0, len(arr) - 1
    result = -1
    while low <= high:
        mid = (low + high) // 2
        if arr[mid] == x:
            result = mid
            low = mid + 1
        elif arr[mid] < x:
            low = mid + 1
        else:
            high = mid - 1
    return result
```

Again, 'last', 'arr', and 'x' are preserved; the lookup logic is simply swapped from a reverse-slice search to an iterative binary-search routine.

## J  TRAINING DETAILS REGARDING SFT/RL

### J.1  FINE-TUNING DETAILS

**Framework and Compute.**  We adapted the Verl framework for supervised fine-tuning (SFT) and Proximal Policy Optimization (PPO), using its PyTorch Fully Sharded Data Parallel (FSDP) backend Experiments ran on a single machine (nnodes=1) with 4 GPUs (n_gpus_per_node=2). PPO rollouts used the VLLM backend; optimization used AdamW.

Table 20: Compute and framework configuration

| Item | Setting |
|------|---------|
| Framework | Verl (PyTorch FSDP backend) |
| Nodes / GPUs | nnodes=1, n_gpus_per_node=2 |
| PPO rollout backend | VLLM |
| Optimizer | AdamW |

**Dataset and Prompting.** Data followed Verl's standard format and was exported as a .parquet file with a 4:1 train/test split. Each problem description served as the prompt; the corresponding code solution was the target response.

Table 21: Dataset summary

| Aspect | Details |
|--------|---------|
| Format | .parquet (Verl standard) |
| Split | 4:1 train:test |
| Input (prompt) | Problem description |
| Target (response) | Code solution |
| Prompt template | See quoted block above |

The completed template we fed into the LLM was:

```
instruction_prefix = "Please provide a self-contained Python script that solves the
    following problem in a markdown code block:"

response_prefix = "Below is a Python script with a self-contained function that solves
    the problem and passes corresponding tests:"

prompt_chat = [
    {"role": "user", "content": f"""\
{instruction_prefix}
```
{problem.strip()}
```
"""},
    {"role": "assistant", "content": f"""\
{response_prefix}
```python
"""}
]
```

The **problem** is the description originally from the dataset, and we called the tokenizer.apply_chat_template to the **prompt_chat** to get the model response.

### J.1.1 SUPERVISED FINE-TUNING (SFT)

Default learning rate was $1 \times 10^{-5}$ for 20 epochs, with manual adjustments between $5 \times 10^{-6}$ and $1 \times 10^{-5}$ depending on model performance. We set max_prompt_length to 1024, batch size to 64, and micro_batch_size_per_gpu to 8. The selected checkpoint (named **model_name-SFT**) was the one immediately prior to observed overfitting, hence we can distinguish memorization from overfitting.

Moreover, we choose the checkpoint at epoch 20 (named **model_name-SFT-overfit**) as the fully overfitting epoch to measure the impact of overfitting to memorization.

### J.1.2 PROXIMAL POLICY OPTIMIZATION (PPO)

Actor, critic, and reference models used identical architectures over 20 epochs. The reward was binary: 1 if the generated response passed all test cases, else 0. We set max_prompt_length to 1024 and max_response_length to 512. Learning rates were $1 \times 10^{-5}$ for the critic and $1 \times 10^{-6}$ for the

Table 22: SFT hyperparameters

| Parameter | Value |
|---|---|
| Epochs | 20 |
| Learning rate | Default $1 \times 10^{-5}$; tuned $5 \times 10^{-6}$–$1 \times 10^{-5}$ |
| max_prompt_length | 1024 |
| Batch size | 64 |
| micro_batch_size_per_gpu | 8 |
| save_freq | after_each_epoch |
| Checkpoint selection | Epoch immediately prior to overfitting |

actor. We used `batch size` 64 with `micro_batch_size_per_gpu` 8, selecting the checkpoint with the highest test reward (named **model_name-PPO**) to get the best performance.

Table 23: PPO setup and hyperparameters

| Parameter | Value |
|---|---|
| Architectures | Actor/Critic/Reference identical |
| Epochs | 20 |
| Reward | Binary (1 if all tests pass; else 0) |
| max_prompt_length | 1024 |
| max_response_length | 512 |
| Learning rate (critic) | $1 \times 10^{-5}$ |
| Learning rate (actor) | $1 \times 10^{-6}$ |
| Batch size | 64 |
| micro_batch_size_per_gpu | 8 |
| save_freq | 5 |
| Checkpoint selection | Highest reward on validset |

## K  EVOLVED-TASK GENERATION (GPT-5)

- **API version**: gpt-5-2025-08-07.

- **Prompt template**: shown in Appendix B.

- **Parameters**: temperature: default; top-p: default; max-tokens 1080.

- **Post-processing**: regex clean-up.

- **Budget**: the estimated cost for generating one round of each evolution type (code rewriting, mutation and paraphrase) for both MBPP+ and BigCodeBench is approximately 450 USD.

