# OpenReview forum: "Memorize or Generalize? Evaluating LLM Code Generation with Code Rewriting"
_ICLR.cc/2026/Conference — Submitted to ICLR 2026_

### Official Review · Reviewer_FbwC · 2025-10-26

**Soundness:** 2
**Presentation:** 3
**Contribution:** 3
**Rating:** 6
**Confidence:** 3

**Summary:**

This paper introduces a code-rewriting pipeline to diagnose harmful memorization in code generation tasks in LLMs. To probe this, the authors evaluate models on altered versions of standard coding problems. They quantify and summarize risk with a single index, the Memorization Risk Index (MRI), which is high only when both the model’s output on the rewrite remains highly similar to the original solution and there is a relative accuracy drop (RAD) from original to rewritten tasks. Empirically, across MBPP+ and BigCodeBench, the study reports a consistent story where memorization risk generally falls with larger model scale on simpler problems but persists on harder ones. The authors also found that SFT tends to increase raw accuracy and memorization risk, while PPO better balances accuracy and risk, often keeping MRI near its base levels.

**Strengths:**

Clear Pipeline Setup: The papers provide a clean pipeline structure, giving an illustrative understanding of the author’s proposed setup. The authors also commit to releasing the evolved tasks, prompts, and code upon publication, which strengthens reproducibility and contribution to the research community for similar research.

Harmful Memorization Framing: The work defines harmful memorization as a joint event, failure after a semantic shift, alongside high similarity to the original solution, rather than treating any code reuse as memorization. The authors also clearly separate style robustness (e.g. paraphrase/mutation) and distinguish its lens from contamination-based definitions.

**Weaknesses:**

Scope Limited to Python:
The paper aims to characterize harmful memorization in code generation but evaluates exclusively on Python tasks. That makes other programming languages' settings uncertain, as different language ecosystems may differ materially from the author’s claims. would substantively strengthen the claim that the findings. A cross-language pilot would substantively strengthen the claim that the observed scaling trends and SFT vs. PPO trade-offs generalize beyond Python.

Dataset-level MRI:
Metric aggregation could possibly mislead. In this paper, MRI is computed as the product of two corpus-level aggregates, so “mean similarity” and “overall accuracy drop” may be driven by different subsets of items in the dataset. When you multiply these two dataset numbers, you can overstate or understate the true co-occurrence of high similarity and failure at the item level.

**Questions:**

Is there any reason why the evaluation is limited to Python?

Is every experiment reported as Pass@1 (including the scaling plots)?

Small Comments:
- A few editorial issues “Budge:” in App. G. I think you meant Budget?
- Small inconsistency in reported compute (Sec. F says 2 GPUs; earlier 4 A100s are mentioned).

---

> ### Author Response · Authors · 2025-11-21
> **Author Response Q1 & Q2**
>
> Thank you for your valuable feedback and for identifying our work’s contribution to capture harmful memorization! We appreciate the opportunity to clarify our scope and metric design as below:
>
> >***Q1***: Scope Limited to Python: Scope Limited to Python: The paper aims to characterize harmful memorization in code generation but evaluates exclusively on Python tasks. That makes other programming languages' settings uncertain, as different language ecosystems may differ materially from the author’s claims. would substantively strengthen the claim that the findings. A cross-language pilot would substantively strengthen the claim that the observed scaling trends and SFT vs. PPO trade-offs generalize beyond Python.
>
> >***Q2***: Is there any reason why the evaluation is limited to Python?
>
> ***A1 & A2***: Thank you for your valuable suggestions! While we agree that understanding memorization beyond Python is important. We respectfully clarify that this scope is necessary and representative for the following three reasons:
>
> The first reason is for **benchmark alignment**. We follow the evaluation paradigm established by the field's standard benchmarks. MBPP+ [1], BigCodeBench [2] and LiveCodeBench [3] are currently the gold standards for assessing correctness and complex instruction following, and they are predominant by Python. Evaluating these benchmarks allows us to evaluate our MRI metric against known baselines and ensures our findings on scaling and SFT are directly comparable to prior literature like ReCode [4].
>
> The second reason is about **Python’s dominance in training data**. Python is the largest portion of code in pre-training corpora (e.g., The Stack, GitHub repositories). Consequently, memorization phenomena are most likely to be observable and impactful in Python.
>
> The third reason is on the **generalizability of our pipeline**. While our experiments are on Python, the Code Rewriting pipeline (Figure 1) and the MRI formulation are language-agnostic. Our pipeline uses LLMs to rewrite logic and prompts, and it can be used on Java, C++, or JavaScript as well. We view our work as establishing the foundational methodology for measuring harmful memorization, which future work can extend to other coding languages.
>
> [1] Is Your Code Generated by Chat{GPT} Really Correct? Rigorous Evaluation of Large Language Models for Code Generation
>
> [2] BigCodeBench: Benchmarking Code Generation with Diverse Function Calls and Complex Instructions
>
> [3] LiveCodeBench: Holistic and Contamination Free Evaluation of Large Language Models for Code
>
> [4] ReCode: Robustness Evaluation of Code Generation Models

---

> ### Author Response · Authors · 2025-11-21
> **Author Response Q3 & Q4**
>
> >***Q3***: Dataset-level MRI: Metric aggregation could possibly mislead. In this paper, MRI is computed as the product of two corpus-level aggregates, so “mean similarity” and “overall accuracy drop” may be driven by different subsets of items in the dataset. When you multiply these two dataset numbers, you can overstate or understate the true co-occurrence of high similarity and failure at the item level.
>
> ***A3***: Thank you for this insightful statistical concern! We are happy to justify MRI at item level, and we implemented i-MRI as suggested. We computed this metric specifically on the subset of tasks the model successfully solved in the original tasks ($T_{passed}$), as "harmful memorization" (regression) is logically conditioned on the ability to solve the task initially. $$iMRI = \frac{1}{|T_{passed}|} \sum_{k \in T_{passed}} (Sim_k \times Drop_k)$$ where $Drop_k$ is 1 if the model fails the rewrite, and 0 otherwise.
> We re-evaluated our results on BigCodeBench and MBPP+ using this $iMRI$. As shown in the table, the instance-level aggregation yields trends and rankings that are **strongly aligned with our original MRI metric**. The reported **Pearson correlation further justified such alignment**. This finding suggests that high similarity and failure in rewriting tasks are not independent factors driven by sets of data. We hope this finding can address your concern, and we will retain the original metric in the main text for consistency and will include the instance-level analysis in the Appendix to demonstrate the robustness of our conclusions against aggregation bias.
>
> **Table1: Pearson correlation between MRI and iMRI**
>
> | Series          | MBPP+  | BCB    |
> |-----------------|--------|--------|
> | Qwen-2.5 Series | 0.9816 | 0.9459 |
> | Coder Series    | 0.9211 | 0.9731 |
>
>
>
> **Table2: MBPP+ (iMRI / MRI)**
>
> | Model                          | iMRI    | MRI    |
> |--------------------------------|---------|--------|
> | Qwen-2.5-0.5B-Instruct         | 0.1494  | 0.0722 |
> | Qwen-2.5-1.5B-Instruct         | 0.1078  | 0.0600 |
> | Qwen-2.5-3B-Instruct           | 0.0490  | 0.0185 |
> | Qwen-2.5-7B-Instruct           | 0.0650  | 0.0240 |
> | Qwen-2.5-14B-Instruct          | 0.0500  | 0.0113 |
> | Qwen-2.5-32B-Instruct          | 0.0325  | 0.0000 |
> |                                |         |        |
> | Qwen-2.5-coder-0.5B-Instruct   | 0.1107  | 0.0615 |
> | Qwen-2.5-coder-1.5B-Instruct   | 0.0870  | 0.0500 |
> | Qwen-2.5-coder-3B-Instruct     | 0.0828  | 0.0328 |
> | Qwen-2.5-coder-7B-Instruct     | 0.0771  | 0.0373 |
> | Qwen-2.5-coder-14B-Instruct    | 0.0722  | 0.0313 |
> | Qwen-2.5-coder-32B-Instruct    | 0.0730  | 0.0354 |
>
> **Table3: BigCodeBench (iMRI / MRI)**
>
> | Model                          | iMRI    | MRI    |
> |--------------------------------|---------|--------|
> | Qwen-2.5-0.5B-Instruct         | 0.2692  | 0.1740 |
> | Qwen-2.5-1.5B-Instruct         | 0.1870  | 0.1444 |
> | Qwen-2.5-3B-Instruct           | 0.1836  | 0.1403 |
> | Qwen-2.5-7B-Instruct           | 0.1727  | 0.1296 |
> | Qwen-2.5-14B-Instruct          | 0.1275  | 0.0841 |
> | Qwen-2.5-32B-Instruct          | 0.1538  | 0.1143 |
> |                                |         |        |
> | Qwen-2.5-coder-0.5B-Instruct   | 0.2609  | 0.1778 |
> | Qwen-2.5-coder-1.5B-Instruct   | 0.2379  | 0.1630 |
> | Qwen-2.5-coder-3B-Instruct     | 0.2108  | 0.1630 |
> | Qwen-2.5-coder-7B-Instruct     | 0.1959  | 0.1477 |
> | Qwen-2.5-coder-14B-Instruct    | 0.1619  | 0.1237 |
> | Qwen-2.5-coder-32B-Instruct    | 0.1510  | 0.1178 |
>
>
>
> >***Q4***: Is every experiment reported as Pass@1?
>
> ***A4***: Yes. As stated in Section 3.1.1, "Pass@1 is reported as $Acc(\mathcal{T})$" and in Section 3.2.1, "Here, $Acc(\cdot)$ denotes Pass@1". All scaling plots and tables utilize Pass@1.
>
>
> >***Reply to your small comments***
>
> Thank you so much for your comments! We will update our manuscript to fix those typos.

---

> ### Author Response · Authors · 2025-11-27
>
> Thank you again for your thoughtful review of our submission!
>
> As the discussion period is drawing to a close, we would like to follow up to see whether there are any remaining questions or points you would like us to address. We appreciate your time and are happy to provide any additional clarification.
>
> Thank you again for your support throughout this process!

---

> > ### Comment · Reviewer_FbwC · 2025-11-27
> >
> > I thank the authors for the very detailed and thoughtful response.
> >
> > I appreciate the clarification that you follow Python dominated benchmarks. That said, I remain somewhat unconvinced by the language-agnostic framing as currently stated. The pipeline you evaluate relies heavily on Python-specific infrastructure (AST), and it is not obvious that the same robustness properties or failure modes would carry over to languages with different syntax and type systems.
> >
> > I thank the authors for introducing the instance-level MRI and reporting correlations between MRI and iMRI on both MBPP+ and BigCodeBench. Seeing that the trends and rankings are strongly aligned alleviates my main concern.
> >
> > Given the clarifications and additional experiments, I maintain my positive overall assessment and my original score.

---

> > > ### Author Response · Authors · 2025-11-28
> > > **Author Response Further Justification (1/2)**
> > >
> > > Thank you so much for your reply and valuable insights! We are grateful to hear our response solved your main concern, and we are happy to further justify the point you are somewhat unconvinced.
> > > >Concern: **The pipeline you evaluate relies heavily on Python-specific infrastructure (AST)**, and it is not obvious that the same robustness properties or failure modes would carry over to languages with different syntax and type systems.
> > >
> > > **Author response**: we agree that AST is Python-specific infrastructure, and robustness check on how sensitive the Memorization Risk Index (MRI) is to implementation choices could further strengthen our contribution. Therefore, we conducted ablations with **(i) 1–3-gram overlap**, **(ii) Jaccard token similarity and Minhash similarity**, and **(iii) cosine similarity of code embeddings**, all of which are **AST-free** and **language-agnostic** (**Table 2** and **Table 3**). Across MBPP+ and BigCodeBench, these variants give MRI values highly correlated with our original measure and preserve model rankings (**Table 1**). This indicates that our method’s robustness properties **do not rely on Python-specific AST structure**, but instead persist across fundamentally different (and language-agnostic) similarity definitions.
> > >
> > > **Table1: Average Pearson correlation across similarity families**
> > > | Model Series      | Avg. Pearson Corr. on MBPP+     | Avg. Pearson Corr. on BigCodebench                      |
> > > |-------------------|----------------------------------|---------------------------|
> > > | Qwen-2.5 series   | 0.9966 ± 0.0029                  | 0.9671 ± 0.0163           |
> > > | Coder series      | 0.9752 ± 0.0420                  | 0.9921 ± 0.0144           |

---

> > > ### Author Response · Authors · 2025-11-28
> > > **Author Response Further Justification (2/2)**
> > >
> > > **Table2: Ablation study for across similarity families on MBPP+**
> > > | Model                         | MRI-1-gram | MRI-2-gram | MRI-3-gram | MRI-Jaccard sim | MRI-Minhash sim | MRI-Cosine embed sim |
> > > |------------------------------|------------|------------|------------|------------------|------------------|------------------------|
> > > | Qwen-2.5 0.5B-Instruct       | 0.0873     | 0.0404     | 0.0206     | 0.0712           | 0.0694           | 0.2126                 |
> > > | Qwen-2.5-1.5B-Instruct       | 0.0704     | 0.0337     | 0.0177     | 0.0568           | 0.0552           | 0.1682                 |
> > > | Qwen-2.5-3B-Instruct         | 0.0265     | 0.0123     | 0.0064     | 0.0224           | 0.0221           | 0.0800                 |
> > > | Qwen-2.5-7B-Instruct         | 0.0282     | 0.0132     | 0.0068     | 0.0224           | 0.0219           | 0.0678                 |
> > > | Qwen-2.5-14B-Instruct        | 0.0135     | 0.0065     | 0.0033     | 0.0107           | 0.0106           | 0.0330                 |
> > > | Qwen-2.5-32B-Instruct        | 0.0000     | 0.0000     | 0.0000     | 0.0000           | 0.0000           | 0.0000                 |
> > > |                              |            |            |            |                  |                  |                        |
> > > | Qwen-2.5-coder-0.5B-Instruct | 0.0764     | 0.0339     | 0.0167     | 0.0640           | 0.0612           | 0.2110                 |
> > > | Qwen-2.5-coder-1.5B-Instruct | 0.0594     | 0.0285     | 0.0151     | 0.0488           | 0.0469           | 0.1424                 |
> > > | Qwen-2.5-coder-3B-Instruct   | 0.0372     | 0.0181     | 0.0095     | 0.0296           | 0.0286           | 0.0865                 |
> > > | Qwen-2.5-coder-7B-Instruct   | 0.0408     | 0.0212     | 0.0123     | 0.0321           | 0.0314           | 0.0922                 |
> > > | Qwen-2.5-coder-14B-Instruct  | 0.0349     | 0.0192     | 0.0117     | 0.0274           | 0.0268           | 0.0730                 |
> > > | Qwen-2.5-coder-32B-Instruct  | 0.0387     | 0.0219     | 0.0137     | 0.0301           | 0.0297           | 0.0808                 |
> > >
> > >
> > > **Table3: Ablation study for different across similarity families on BigCodeBench**
> > >
> > > | Model                          | MRI-1-gram | MRI-2-gram | MRI-3-gram | MRI-Jaccard sim | MRI-Minhash sim | MRI-Cosine embed sim |
> > > |-------------------------------|------------|------------|------------|------------------|------------------|------------------------|
> > > | Qwen-2.5 0.5B-Instruct        | 0.3245     | 0.2302     | 0.1823     | 0.2548           | 0.2586           | 0.5986                 |
> > > | Qwen-2.5-1.5B-Instruct        | 0.2705     | 0.1933     | 0.1521     | 0.2138           | 0.2187           | 0.5245                 |
> > > | Qwen-2.5-3B-Instruct          | 0.2483     | 0.1827     | 0.1453     | 0.1955           | 0.1987           | 0.4353                 |
> > > | Qwen-2.5-7B-Instruct          | 0.2401     | 0.1761     | 0.1395     | 0.1902           | 0.1932           | 0.4397                 |
> > > | Qwen-2.5-14B-Instruct         | 0.1690     | 0.1229     | 0.0970     | 0.1345           | 0.1387           | 0.3316                 |
> > > | Qwen-2.5-32B-Instruct         | 0.2053     | 0.1533     | 0.1226     | 0.1628           | 0.1650           | 0.3547                 |
> > > |                               |            |            |            |                  |                  |                        |
> > > | Qwen-2.5-coder-0.5B-Instruct  | 0.2825     | 0.2018     | 0.1597     | 0.2209           | 0.2222           | 0.5056                 |
> > > | Qwen-2.5-coder-1.5B-Instruct  | 0.2813     | 0.2094     | 0.1693     | 0.2234           | 0.2257           | 0.4778                 |
> > > | Qwen-2.5-coder-3B-Instruct    | 0.2641     | 0.1977     | 0.1583     | 0.2102           | 0.2109           | 0.4444                 |
> > > | Qwen-2.5-coder-7B-Instruct    | 0.2451     | 0.1838     | 0.1476     | 0.1959           | 0.1979           | 0.4193                 |
> > > | Qwen-2.5-coder-14B-Instruct   | 0.2099     | 0.1578     | 0.1264     | 0.1668           | 0.1685           | 0.3554                 |
> > > | Qwen-2.5-coder-32B-Instruct   | 0.1994     | 0.1491     | 0.1188     | 0.1579           | 0.1590           | 0.3410                 |

---

### Official Review · Reviewer_GhZw · 2025-10-30

**Soundness:** 3
**Presentation:** 3
**Contribution:** 2
**Rating:** 4
**Confidence:** 5

**Summary:**

This paper proposes a code‐rewriting evaluation framework to diagnose harmful memorization in LLM code generation. The authors construct “semantically edited” variants of known solutions and reverse-engineer task descriptions, then measure MRI, a diagnostic score defined as Similarity × RAD (relative accuracy drop). Experiments on MBPP+ and BigCodeBench, across multiple model sizes and fine-tuning strategies, suggest that larger models reduce harmful memorization on easier tasks, while harder tasks retain non-trivial MRI; moreover, SFT tends to raise MRI whereas PPO achieves a better accuracy–memorization balance. The work is well scoped (focusing on harmful memorization rather than benign reuse), the rewriting pipeline is innovative, and MRI is simple and model-agnostic. However, the metric design has notable weaknesses: RAD can be inflated at low baseline accuracy, the similarity component (equal-weight AST + edit distance) lacks ablations and comparisons to alternative overlap measures, and the fine-tuning analysis remains largely observational without an MRI-aware training recipe. Overall, the paper is promising and practically relevant, but would benefit from robustness analyses (RAD stabilization, similarity ablations), broader metric comparisons, and closing the loop by incorporating MRI into training/selection protocols.

**Strengths:**

1. Well‑scoped formulation of harmful memorization. The paper explicitly targets harmful memorization—cases where high code similarity coincides with a meaningful accuracy drop—rather than benign reuse. The MRI construct (Similarity × RAD) operationalizes this notion in a way that is easy to interpret and analyze.
2. Innovative code‑rewriting evaluation pipeline. By making a single semantic change to a reference solution and then reverse‑engineering the task description, the pipeline creates tasks that are superficially similar yet semantically distinct. This design probes whether models can go beyond rote recall to genuine reasoning, improving ecological validity over paraphrase/noise baselines.
3. Simple, model‑agnostic metric with diagnostic value. The MRI score combines a similarity component with a relative performance‑drop component, yielding a unified signal that is high only when both similarity and degradation are high. Its two‑factor structure helps practitioners diagnose whether failures stem from brittle recall or broader capability gaps.

**Weaknesses:**

1. RAD is inflated at low baseline accuracy, which in turn overstates MRI. RAD is defined as \max\{0,\tfrac{Acc(T_{ori}) - Acc(T_{rew})}{Acc(T_{ori})}\}. When Acc(T_{ori}) is small, even a tiny absolute drop yields a large relative decrease, which elevates MRI (since MRI = Similarity × RAD). This effect is especially visible on BigCodeBench, where lower baseline accuracy coincides with persistently higher RAD and non‑zero MRI.
2. Similarity uses an equal‑weight average of AST and edit distance, but lacks ablations and alternative metrics. The paper fixes similarity as the unweighted mean of AST‑level similarity and token edit‑distance similarity. However, it does not report ablations (AST‑only, edit‑only, or weighted variants) or threshold‑sensitivity analyses. It also does not compare against common token‑overlap metrics such as n‑gram/Jaccard or MinHash. This under‑supports the design choice and limits the interpretability of what the similarity score is actually capturing (semantics vs. surface form).
3. The “Impact of Fine‑Tuning Strategies on Memorization” section is largely observational and lacks actionable training guidance using MRI. The paper observes that SFT tends to increase task accuracy but also raises MRI, while PPO appears to strike a better balance. However, MRI/Sim/RAD are not incorporated into the training objective, early stopping, or model selection (e.g., as regularizers, constraints, or explicit rewards). As a result, the findings remain descriptive rather than prescriptive, without a concrete, reproducible recipe for using MRI to guide training decisions.

**Questions:**

See Weaknesses.

---

> ### Author Response · Authors · 2025-11-21
> **Author Response Q1**
>
> Thank you so much for your valuable time to review our work! We will address your concerns point by point:
>
> >***Q1*** RAD is inflated at low baseline accuracy, which in turn overstates MRI. RAD is defined as \max{0,\tfrac{Acc(T_{ori}) - Acc(T_{rew})}{Acc(T_{ori})}}. When Acc(T_{ori}) is small, even a tiny absolute drop yields a large relative decrease, which elevates MRI (since MRI = Similarity × RAD). This effect is especially visible on BigCodeBench, where lower baseline accuracy coincides with persistently higher RAD and non‑zero MRI.
>
> ***A1*** 	Thank you for your insightful concern! We agree that RAD can be harder to interpret when the baseline accuracy is extremely low. To avoid over-interpreting such cases, we restrict our evaluation to the Acc(T_{ori} that is statistically stable. Concretely, suppose a benchmark has N tasks and p= Acc(T_{ori}), so the model solves k = Np tasks. We require:
>
> **At least M = 50** correctly solved tasks on the original tasks (i.e., k >= 50).
>
> **Acc(T_{ori}) >= 10%** so that, for example, a 5-percentage-point absolute drop does not exceed a 50% relative drop.
>
> The first condition controls the variance of Acc(T_{ori}). For MBPP+ (378 tasks), M = 50 implies p >= 13.2%, and the corresponding 95% confidence interval for p has width ≈±3–4 percentage points. For BigCodeBench (1,140 tasks), M = 50 implies p >= 4.4%, with 95% CI width ≈±1–2 points. These uncertainties are much smaller than the 5–20 point drops we analyze for memorization, so RAD/MRI are not dominated by statistical noise. The second condition explicitly restricts the denominator in RAD to prevent large RADs brought by small Acc(T_{ori}). For models below this competence threshold (e.g., ≈9% Acc(T_{ori}) on BigCodeBench), we report Acc(T_{ori}) and Acc(T_{rew}) but mark RAD/MRI as “N/A (below competence threshold)” and do not use them in cross-model MRI comparisons.
>
> **On using relative metrics**: Our use of a relative drop metric is aligned with standard practice in robustness analysis in code generation. For instance, Wang et al. (2023) [1] introduce Robust Drops@k in ReCode, defined as the relative drop in Pass@k between original and perturbed prompts for code generation models.  ￼Likewise, CodeFort [2] defines a Drop% = (Nominal Pass@k – Robust Pass@k)/Nominal Pass@k. In adversarial robustness, relative drop in accuracy compared to the clean baseline is popularly adopted as a key metric. We follow this established pattern, and now we additionally restrict the baseline accuracy to be sufficiently high and stable.
>
> [1] ReCode: Robustness Evaluation of Code Generation Models
>
> [2] CodeFort: Robust Training for Code Generation Models

---

> ### Author Response · Authors · 2025-11-21
> **Author Response Q2**
>
> >***Q2*** The “Impact of Fine‑Tuning Strategies on Memorization” section is largely observational and lacks actionable training guidance using MRI. The paper observes that SFT tends to increase task accuracy but also raises MRI, while PPO appears to strike a better balance. However, MRI/Sim/RAD are not incorporated into the training objective, early stopping, or model selection (e.g., as regularizers, constraints, or explicit rewards). As a result, the findings remain descriptive rather than prescriptive, without a concrete, reproducible recipe for using MRI to guide training decisions
>
> ***A2*** Thank you for your insight regarding the descriptive nature of our findings. We agree that while our current work focuses on defining and measuring harmful memorization, the ultimate goal is to guide training. While integrating MRI directly into the loss function (as a regularizer or reward) is a promising direction for future work, we believe MRI can also offer diagnostic signals for model selection. To demonstrate this, we performed analysis on our SFT runs (Qwen-2.5-7B and Qwen-2.5-Coder-7B) to observe how MRI changes compared to validation loss. We tracked Original Accuracy ($Acc(\mathcal{T}_{ori})$) and MRI across three training stages: **Early**, **Best Validation Loss (reported in paper)**, and **Late Stage** and report in the table below.
>
>
> | Model             | Status (loss)         | Acc (Tori) | MRI     |
> |-------------------|------------------------|------------|---------|
> | Qwen-2.5-7B       | Early (0.324)          | 35.96%     | 0.1540  |
> |                   | Min-Val Loss (0.302)   | 37.72%     | 0.1733  |
> |                   | Late (0.326)           | 39.47%     | 0.1795  |
> |                   |                        |            |         |
> | Qwen-2.5-Coder    | Early (0.325)          | 35.96%     | 0.1537  |
> |                   | Min-Val Loss (0.292)   | 40.79%     | 0.1864  |
> |                   | Late (0.318)           | 42.54%     | 0.2216  |
>
>
> From the table, accuracy suggests that continuing training is beneficial for both models. However, monitoring MRI reveals a different story. In the Coder Model, Min-Val Loss to Late shows a marginal accuracy gain of ~1.7% comes at a disproportionate cost, because MRI spikes by ~19% (0.1864 $\rightarrow$ 0.2216). This signals that the model shifts **from generalization to harmful memorization**. In the General Model, Min-Val Loss to Late shows that the MRI increase is more gradual (0.1733 $\rightarrow$ 0.1795), which suggests a more linear trade-off between accuracy and memorization.
>
> Based on these findings, we propose using MRI as a secondary stopping criterion to detect "diminishing returns" to avoid harmful memorization. Future researchers can monitor the gradient of MRI relative to accuracy ($\Delta MRI / \Delta Acc$). When this ratio spikes (as seen in the Coder model's late stage), training could be stopped to preserve robustness, even if validation accuracy is still increasing. This allows researchers to distinguish between if the model is genuine learning or brute-force memorizing the training corpus. We will add this analysis to the discussion section to show how MRI can practically inform model selection for fine-tuning.

---

> > ### Comment · Reviewer_GhZw · 2025-11-25
> >
> > Thank you for the clarification on using MRI as a stopping criterion. We agree that memorization is inevitable during training. However, it seems counter-intuitive to stop training when Accuracy (Acc) is still rising, which is generally considered a positive signal and the primary objective of fine-tuning. **Few practitioners would halt training while performance (Acc) continues to improve.** Could the authors provide a more compelling, practical justification for why a secondary metric like MRI should override the continuous improvement in validation accuracy, especially when there is still potential for performance gains?

---

> > > ### Author Response · Authors · 2025-11-25
> > >
> > > Thank you so much for this thoughtful follow-up and for pointing out the ambiguity in our earlier wording. Our intention is not to suggest that practitioners should literally stop optimization as soon as MRI is getting worse while validation accuracy is still improving. Rather, we propose using MRI as a secondary metric for checkpoint selection after training. Among saved checkpoints with different validation losses/accuracies, we can then choose the model that offers a **good trade-off** between performance and memorization risk. In this view, it can be reasonable for MRI to **take precedence over small additional gains in validation accuracy** when those gains are achieved at the cost of high harmful memorization and degraded robustness on rewritten tasks. We will revise the text to clearly frame MRI as a risk-aware checkpoint selection criterion, rather than as a signal to stop training.

---

> ### Author Response · Authors · 2025-11-21
> **Author Response Q3 (Direct answers)**
>
> >***Q3*** Similarity uses an equal‑weight average of AST and edit distance, but lacks ablations and alternative metrics. The paper fixes similarity as the unweighted mean of AST‑level similarity and token edit‑distance similarity. However, it does not report ablations (AST‑only, edit‑only, or weighted variants) or threshold‑sensitivity analyses. It also does not compare against common token‑overlap metrics such as n‑gram/Jaccard or MinHash. This under‑supports the design choice and limits the interpretability of what the similarity score is actually capturing (semantics vs. surface form).
>
> ***A3*** Thank you for encouraging this valuable validation. We have now added extensive ablations and alternative-metric evaluations on both BigCodeBench and MBPP+, each covering **a) AST-only, Edit-only, and weighted mixtures** (Edit:AST = 2:8, 4:6, 6:4, 8:2) and **b) Alternative similarity families**, including 1–3-gram overlap, Jaccard, MinHash, and cosine code-embedding similarity. We found that **MRI is extremely stable under all AST/edit weightings**, and **MRI is also stable across similarity families**. Below are the details:
>
> **MRI is extremely stable under all AST/edit weightings**. We found that model rankings and trends reported in the paper (scaling, coder vs non-coder models) remain unchanged. We further calculated the Pearson correlation for Qwen-2.5 Instruct series and Qwen-2.5 Coder Instruct series [3], which demonstrates the consistency in the trend.
>
> **MRI is also stable across similarity families**. Jaccard, MinHash, and 1–3-grams produce MRI values that closely track our original MRI, and even cosine-embedding similarity (deviates most in scale) preserves the relative trend of models, align with our conclusions.
>
> These results demonstrate that our main findings are **not only suitable for the particular 5:5 AST+edit design**. MRI behaves consistently across semantic metrics (AST) and surface-form metrics (edit, n-gram, Jaccard, MinHash). Finally, we clarify that MRI is threshold-free: similarity appears as a continuous weight on the accuracy drop (RAD_rew), so there is no specific decision boundary depending on the exact scale of the similarity metric.
>
> **Table1: Average Pearson correlation for AST/edit weightings**
> | Model Series      | Avg. Pearson Corr. on MBPP+     | Avg. Pearson Corr. on BigCodebench                      |
> |-------------------|----------------------------------|---------------------------|
> | Qwen-2.5 series   | 0.9997 ± 0.0004                  | 0.9921 ± 0.0144           |
> | Coder series      | 0.9974 ± 0.0042                  | 0.9780 ± 0.0367           |
>
> **Table2: Average Pearson correlation across similarity families**
> | Model Series      | Avg. Pearson Corr. on MBPP+     | Avg. Pearson Corr. on BigCodebench                      |
> |-------------------|----------------------------------|---------------------------|
> | Qwen-2.5 series   | 0.9966 ± 0.0029                  | 0.9671 ± 0.0163           |
> | Coder series      | 0.9752 ± 0.0420                  | 0.9921 ± 0.0144           |
>
> [3] Calculated as the average Pearson correlation between the MRI scores obtained using our default metric and each alternative variant.

---

> ### Author Response · Authors · 2025-11-21
> **Author Response Q3 (Complete results 1/2)**
>
> ***Complete results on MBPP+for A3***
>
> **Table3: Ablation study for different AST/edit weightings on MBPP+**
> | Model                         | MRI (orig 5:5) | MRI (edit only) | MRI (ast only) | MRI (edit:ast=2:8) | MRI (edit:ast=4:6) | MRI (edit:ast=6:4) | MRI (edit:ast=8:2) |
> |------------------------------|----------------|-----------------|----------------|----------------------|----------------------|----------------------|----------------------|
> | Qwen-2.5 0.5B-Instruct       | 0.0722         | 0.1025          | 0.0420         | 0.0541               | 0.0662               | 0.0783               | 0.0904               |
> | Qwen-2.5-1.5B-Instruct       | 0.0600         | 0.0832          | 0.0368         | 0.0461               | 0.0554               | 0.0647               | 0.0739               |
> | Qwen-2.5-3B-Instruct         | 0.0185         | 0.0249          | 0.0122         | 0.0147               | 0.0172               | 0.0198               | 0.0223               |
> | Qwen-2.5-7B-Instruct         | 0.0240         | 0.0342          | 0.0139         | 0.0179               | 0.0220               | 0.0260               | 0.0301               |
> | Qwen-2.5-14B-Instruct        | 0.0113         | 0.0161          | 0.0065         | 0.0084               | 0.0103               | 0.0122               | 0.0142               |
> | Qwen-2.5-32B-Instruct        | 0.0000         | 0.0000          | 0.0000         | 0.0000               | 0.0000               | 0.0000               | 0.0000               |
> |                              |                |                 |                |                      |                      |                      |                      |
> | Qwen-2.5-coder-0.5B-Instruct | 0.0615         | 0.0905          | 0.0325         | 0.0441               | 0.0557               | 0.0673               | 0.0789               |
> | Qwen-2.5-coder-1.5B-Instruct | 0.0500         | 0.0713          | 0.0286         | 0.0372               | 0.0457               | 0.0542               | 0.0627               |
> | Qwen-2.5-coder-3B-Instruct   | 0.0328         | 0.0472          | 0.0184         | 0.0242               | 0.0300               | 0.0357               | 0.0415               |
> | Qwen-2.5-coder-7B-Instruct   | 0.0373         | 0.0526          | 0.0219         | 0.0281               | 0.0342               | 0.0403               | 0.0465               |
> | Qwen-2.5-coder-14B-Instruct  | 0.0313         | 0.0433          | 0.0193         | 0.0241               | 0.0289               | 0.0337               | 0.0385               |
> | Qwen-2.5-coder-32B-Instruct  | 0.0354         | 0.0490          | 0.0218         | 0.0272               | 0.0327               | 0.0381               | 0.0436               |
>
> **Table4: Ablation study for across similarity families on MBPP+**
> | Model                         | MRI-1-gram | MRI-2-gram | MRI-3-gram | MRI-Jaccard sim | MRI-Minhash sim | MRI-Cosine embed sim |
> |------------------------------|------------|------------|------------|------------------|------------------|------------------------|
> | Qwen-2.5 0.5B-Instruct       | 0.0873     | 0.0404     | 0.0206     | 0.0712           | 0.0694           | 0.2126                 |
> | Qwen-2.5-1.5B-Instruct       | 0.0704     | 0.0337     | 0.0177     | 0.0568           | 0.0552           | 0.1682                 |
> | Qwen-2.5-3B-Instruct         | 0.0265     | 0.0123     | 0.0064     | 0.0224           | 0.0221           | 0.0800                 |
> | Qwen-2.5-7B-Instruct         | 0.0282     | 0.0132     | 0.0068     | 0.0224           | 0.0219           | 0.0678                 |
> | Qwen-2.5-14B-Instruct        | 0.0135     | 0.0065     | 0.0033     | 0.0107           | 0.0106           | 0.0330                 |
> | Qwen-2.5-32B-Instruct        | 0.0000     | 0.0000     | 0.0000     | 0.0000           | 0.0000           | 0.0000                 |
> |                              |            |            |            |                  |                  |                        |
> | Qwen-2.5-coder-0.5B-Instruct | 0.0764     | 0.0339     | 0.0167     | 0.0640           | 0.0612           | 0.2110                 |
> | Qwen-2.5-coder-1.5B-Instruct | 0.0594     | 0.0285     | 0.0151     | 0.0488           | 0.0469           | 0.1424                 |
> | Qwen-2.5-coder-3B-Instruct   | 0.0372     | 0.0181     | 0.0095     | 0.0296           | 0.0286           | 0.0865                 |
> | Qwen-2.5-coder-7B-Instruct   | 0.0408     | 0.0212     | 0.0123     | 0.0321           | 0.0314           | 0.0922                 |
> | Qwen-2.5-coder-14B-Instruct  | 0.0349     | 0.0192     | 0.0117     | 0.0274           | 0.0268           | 0.0730                 |
> | Qwen-2.5-coder-32B-Instruct  | 0.0387     | 0.0219     | 0.0137     | 0.0301           | 0.0297           | 0.0808                 |

---

> ### Author Response · Authors · 2025-11-21
> **Author Response Q3 (Complete results 2/2)**
>
> ***Complete results on BogCodeBench for A3***
> **Table5: Ablation study for different AST/edit weightings on BigCodeBench**
>
> | Model                          | MRI (orig 5:5) | MRI (edit sim) | MRI (ast sim) | MRI (edit:ast=2:8) | MRI (edit:ast=4:6) | MRI (edit:ast=6:4) | MRI (edit:ast=8:2) |
> |-------------------------------|----------------|-----------------|----------------|----------------------|----------------------|----------------------|----------------------|
> | Qwen-2.5 0.5B-Instruct        | 0.1740         | 0.2745          | 0.0736         | 0.1138               | 0.1540               | 0.1941               | 0.2343               |
> | Qwen-2.5-1.5B-Instruct        | 0.1444         | 0.2166          | 0.0722         | 0.1011               | 0.1300               | 0.1589               | 0.1878               |
> | Qwen-2.5-3B-Instruct          | 0.1403         | 0.2158          | 0.0649         | 0.0951               | 0.1252               | 0.1554               | 0.1856               |
> | Qwen-2.5-7B-Instruct          | 0.1296         | 0.1958          | 0.0635         | 0.0899               | 0.1164               | 0.1429               | 0.1693               |
> | Qwen-2.5-14B-Instruct         | 0.0841         | 0.1227          | 0.0455         | 0.0609               | 0.0764               | 0.0918               | 0.1073               |
> | Qwen-2.5-32B-Instruct         | 0.1143         | 0.1711          | 0.0576         | 0.0803               | 0.1030               | 0.1257               | 0.1484               |
> |                               |                |                 |                |                      |                      |                      |                      |
> | Qwen-2.5-coder-0.5B-Instruct  | 0.1778         | 0.2802          | 0.0753         | 0.1163               | 0.1573               | 0.1982               | 0.2392               |
> | Qwen-2.5-coder-1.5B-Instruct  | 0.1630         | 0.2487          | 0.0772         | 0.1115               | 0.1458               | 0.1801               | 0.2144               |
> | Qwen-2.5-coder-3B-Instruct    | 0.1630         | 0.2463          | 0.0796         | 0.1130               | 0.1463               | 0.1797               | 0.2130               |
> | Qwen-2.5-coder-7B-Instruct    | 0.1477         | 0.2192          | 0.0762         | 0.1048               | 0.1334               | 0.1620               | 0.1906               |
> | Qwen-2.5-coder-14B-Instruct   | 0.1237         | 0.1881          | 0.0593         | 0.0850               | 0.1108               | 0.1366               | 0.1623               |
> | Qwen-2.5-coder-32B-Instruct   | 0.1178         | 0.1804          | 0.0552         | 0.0803               | 0.1053               | 0.1303               | 0.1554               |
>
> **Table6: Ablation study for different across similarity families on BigCodeBench**
>
> | Model                          | MRI-1-gram | MRI-2-gram | MRI-3-gram | MRI-Jaccard sim | MRI-Minhash sim | MRI-Cosine embed sim |
> |-------------------------------|------------|------------|------------|------------------|------------------|------------------------|
> | Qwen-2.5 0.5B-Instruct        | 0.3245     | 0.2302     | 0.1823     | 0.2548           | 0.2586           | 0.5986                 |
> | Qwen-2.5-1.5B-Instruct        | 0.2705     | 0.1933     | 0.1521     | 0.2138           | 0.2187           | 0.5245                 |
> | Qwen-2.5-3B-Instruct          | 0.2483     | 0.1827     | 0.1453     | 0.1955           | 0.1987           | 0.4353                 |
> | Qwen-2.5-7B-Instruct          | 0.2401     | 0.1761     | 0.1395     | 0.1902           | 0.1932           | 0.4397                 |
> | Qwen-2.5-14B-Instruct         | 0.1690     | 0.1229     | 0.0970     | 0.1345           | 0.1387           | 0.3316                 |
> | Qwen-2.5-32B-Instruct         | 0.2053     | 0.1533     | 0.1226     | 0.1628           | 0.1650           | 0.3547                 |
> |                               |            |            |            |                  |                  |                        |
> | Qwen-2.5-coder-0.5B-Instruct  | 0.2825     | 0.2018     | 0.1597     | 0.2209           | 0.2222           | 0.5056                 |
> | Qwen-2.5-coder-1.5B-Instruct  | 0.2813     | 0.2094     | 0.1693     | 0.2234           | 0.2257           | 0.4778                 |
> | Qwen-2.5-coder-3B-Instruct    | 0.2641     | 0.1977     | 0.1583     | 0.2102           | 0.2109           | 0.4444                 |
> | Qwen-2.5-coder-7B-Instruct    | 0.2451     | 0.1838     | 0.1476     | 0.1959           | 0.1979           | 0.4193                 |
> | Qwen-2.5-coder-14B-Instruct   | 0.2099     | 0.1578     | 0.1264     | 0.1668           | 0.1685           | 0.3554                 |
> | Qwen-2.5-coder-32B-Instruct   | 0.1994     | 0.1491     | 0.1188     | 0.1579           | 0.1590           | 0.3410                 |

---

> ### Author Response · Authors · 2025-11-27
>
> Thank you once again for taking the time to review our paper and for sharing your thoughtful follow-up comments!
>
> With the discussion period nearing its end, we would like to reach out in case you had any follow-up thoughts or additional points you would like us to consider. We truly appreciate your insights and are very willing to discuss any remaining issues to strengthen the paper. Please feel free to let us know if further clarification is needed.
>
> Thanks again for your time and dedication!

---

### Official Review · Reviewer_1QvN · 2025-11-01

**Soundness:** 3
**Presentation:** 3
**Contribution:** 3
**Rating:** 2
**Confidence:** 4

**Summary:**

This paper investigates memorization versus generalization in LLM code generation by introducing an evaluation methodology called code rewriting. The authors distinguish between harmful memorization (high similarity to training solutions coupled with performance drops on semantically altered tasks) and benign code reuse. They propose the Memorization Risk Index (MRI), which multiplies solution similarity with relative accuracy drop under semantic perturbations. The paper evaluates multiple LLM families (Qwen, Llama) across MBPP+ and BigCodeBench benchmarks, examining how model scale and fine-tuning strategies (SFT vs. PPO) affect memorization. Key findings include: (1) memorization generally decreases with model scale, especially on simpler tasks; (2) SFT improves accuracy but increases memorization risk; (3) PPO (RL) offers a better memorization-accuracy trade-off.

**Strengths:**

- The code rewriting approach offers a novel way to evaluate memorization that goes beyond surface-level similarity metrics. The introduction of MRI to evaluate memorization is quite interesting as it helps capture harmful memorization as failure under high similarity on code-rewriting tasks.

- The experimental methodology is rigorous, with quality assurance through both automated (GPT-5 as judge) and manual validation. The evaluation spans multiple model families (Qwen, Llama), sizes (0.5B-70B), benchmarks (MBPP+, BigCodeBench), and training approaches (base, SFT, PPO), providing comprehensive coverage.

- The inclusion of semantic-preserving perturbations (mutation, paraphrase) as baselines effectively differentiates memorization from robustness issues.

**Weaknesses:**

- Need more justification around the design of the MRI score (why multiplication operation chosen)
- While quality assurance is performed, only 10% manual validation is reported. What was the acceptance rate from the LLM judge? How many generated tasks were rejected?
- The paper reports aggregate MRI scores but provides limited insight into why models fail on rewritten tasks. Are failures due to specific semantic changes (e.g., sorting, filtering operations)? Qualitative analysis of high-MRI cases would provide actionable insights. The paper spends more time in formulating the problem rather than talking about ablations.
- The concepts introduced in this paper, i.e., memorization decreases with size, SFT increases memorization whereas PPO alleviates it aren't novel. The authors should have spend more analysis on why are the models struggling with code rewrites

**Questions:**

- Very difficult to follow the results with a combination of tables and results. Table 1 should be presented like Table 2 (or maybe merged) to show all the results in one place and the graphs maybe added to drive more discussion.
- There are some obvious questions from the table - The trend of RAD paraphrasing is not clear? (the trend is reversed in some family of models)
- You mention "modify one logic" in ground truth solutions. What types of logic changes are most common in your dataset (e.g., change operators, add constraints, reverse logic)? Is there diversity in change types, and does MRI vary by change type?
- Why does mutation (semantic-preserving) cause higher RAD than paraphrase? What specific mutation types (character noise, word scrambling, capitalization) are most problematic? Does this suggest brittleness distinct from memorization and does it indicate problems in real world scenarios ?

---

> ### Author Response · Authors · 2025-11-21
> **Author Response Q1 and intro to Q2&Q3**
>
> We sincerely appreciate your constructive and insightful comments! Before we explain each of your concerns point by point, we would like to quickly go through our contribution. Like what you mentioned in our strengths, we offered a **novel** way to capture **“harmful memorization** as **ailure under high similarity** on code-rewriting tasks”. Our metric design, experiment and conclusion with MRI all lies in this fundamental terminology, as we defined at line 87 to 90 in Introduction.
>
> >***Q1*** Need more justification around the design of the MRI score (why multiplication operation chosen)
>
> ***A1*** Thank you for raising an important concern! We will justify the design of MRI below:
>
> MRI is designed to detect harmful memorization where the model both copies training code and that copied behavior fails under semantic changes. So, we want high scores only when **both** similarity and accuracy drop are high, and to be downweighted as soon as either signal is low, suggesting “this is probably not harmful memorization.” The multiplicative form does exactly this. For example, Sim=0.9,RAD=0.9 gives MRI = 0.81, but Sim=0.1,RAD=0.9 yields only 0.09. In the latter case, even though the model fails often, it barely reuses the code. This intuitively means that it is trying to generate new code to accommodate the semantic changes, so the high accuracy drop reflects general brittleness rather than harmful memorization and should not receive a high MRI score. In terms of other operations, for example an arithmetic mean cannot express this distinction: Sim=0.1,RAD=0.9 would still get a relatively large 0.5, which would overstate memorization risk in such a situation where code reuse is weak. We will update our manuscript to reflect this justification.
>
>
> ***Q2 and Q3*** Thank you for your insightful concern. Since both Q2 and Q3 are built upon the types of semantic changes, we will first provide the statistics of those types, then answer each of ***Q2 and Q3 in the next comment***. Below is the general statistics of the change types:
>
> | Logic Change Type      | bigcodebench Count | bigcodebench % | mbppplus Count | mbppplus % |
> |------------------------|--------------------|-----------------|----------------|-------------|
> | Total                  | 1140               | 100.00%         | 378            | 100.00%     |
> | Component Swap         | 393                | 34.47%          | 52             | 13.76%      |
> | Data Transformation    | 385                | 33.77%          | 58             | 15.34%      |
> | Constraint Change      | 185                | 16.23%          | 86             | 22.75%      |
> | Logic Reversal         | 50                 | 4.39%           | 68             | 17.99%      |
> | Workflow Modification  | 50                 | 4.39%           | 10             | 2.65%       |
> | Constant Swap          | 37                 | 3.25%           | 36             | 9.52%       |
> | Operation Order Swap   | 20                 | 1.75%           | 12             | 3.17%       |
> | Operator Change        | 18                 | 1.58%           | 36             | 9.52%       |
> | Other                  | 2                  | 0.18%           | 20             | 5.29%       |
>
> Each logic change types are defined as below:
>
> 1.  Component Swap: A function, class, or method is swapped (e.g., `sum` -> `max`, `np.std` -> `np.var`).
>
> 2.  Data Transformation: The data is changed *before* the main logic (e.g., `np.log(data)`, `data.T`, `data.pct_change()`, tokenizing data).
>
> 3.  Constraint Change: A rule, parameter, or logical condition is modified (e.g., loop step `range(..., 2)`, a regex, a SQL `UNIQUE` constraint).
>
> 4.  Logic Reversal: The logic is inverted (e.g., `max` -> `min`, `sorted(...)` -> `sorted(..., reverse=True)`, swapping `x` and `y` variables).
>
> 5.  Operator Change: A single mathematical or logical operator is swapped (e.g., `^` -> `+`, `==` -> `!=`).
>
> 6.  Operation Order Swap: The sequence of two or more operations is changed (e.g., `blur` then `gray` -> `gray` then `blur`).
>
> 7.  Constant Swap: A hard-coded constant is changed (e.g., `lowercase` -> `UPPERCASE`).
>
> 8.  Workflow Modification: A more complex structural change, like adding a new library step (`zlib`) or changing output nesting.
>
> 9.  Other: Other types of logic changes that do not fit into the above categories.
>
> To make our analysis more robust, we **filter out** the change types that are **less than 50 samples** for each dataset. We now answer each of ***Q2 and Q3 in the next comment***

---

> ### Author Response · Authors · 2025-11-21
> **Author Response Q2 (Direct answer)**
>
> >***Q2*** You mention "modify one logic" in ground truth solutions. What types of logic changes are most common in your dataset (e.g., change operators, add constraints, reverse logic)? Is there diversity in change types, and does MRI vary by change type?
>
> ***A2*** Thank you for asking about the diversity of “modify one logic” changes and how MRI depends on them. As shown in the statistics table, BigCodeBench has a rich mix of changes with Component Swap (34.5%) and Data Transformation (33.8%) most common, followed by Constraint Change (16.2%), while MBPP+ has a more balanced spread where Constraint Change (22.8%), Logic Reversal (18.0%), Data Transformation (15.3%), Component Swap (13.8%), Constant/Operator Swap (~9–10%). MRI also varies by change type. On MBPP+, Constraint Change and Data Transformation tend to have the highest MRI across Qwen-2.5 instruct/coder variants, whereas Component Swap usually has near-zero MRI in larger models; on BigCodeBench, all types show substantial MRI, but Data Transformation, Constraint Change, Logic Reversal, and Workflow Modification consistently sit above the overall MRI for many model sizes. **To answer your concern**, we found that **(a) the dataset contains diverse logic edits**, and **(b) MRI is strongly type-dependent rather than being driven by a single dominant type**.

---

> ### Author Response · Authors · 2025-11-21
> **Author Response Q2 (Complete results)**
>
> **Table1: MRI across logic change types for Qwen-2.5 Instruct and Coder variants on MBPP+.**
>
>
>
> | Model                         | Component Swap      | Constraint Change     | Data Transformation   | Logic Reversal        | All        |
> |------------------------------|---------------------|-----------------------|-----------------------|-----------------------|------------|
> | Qwen-2.5-0.5B-Instruct       | 0.0212              | 0.0929                | 0.0877                | **0.1119**            | 0.0722     |
> | Qwen-2.5-1.5B-Instruct       | **0.0644**          | 0.0584                | 0.0592                | 0.0553                | 0.0600     |
> | Qwen-2.5-3B-Instruct         | 0.0149              | **0.0380**            | 0.0224                | 0.0000                | 0.0185     |
> | Qwen-2.5-7B-Instruct         | 0.0154              | **0.0543**            | 0.0284                | 0.0072                | 0.0240     |
> | Qwen-2.5-14B-Instruct        | 0.0139              | **0.0573**            | 0.0172                | 0.0000                | 0.0113     |
> | Qwen-2.5-32B-Instruct        | 0.0000              | 0.0000                | 0.0000                | **0.0073**            | 0.0000     |
> | *Mean ± Std*                 | *0.02 ± 0.02*       | *0.05 ± 0.03*         | *0.04 ± 0.03*         | *0.03 ± 0.05*         | *—*        |
> | Qwen-2.5-coder-0.5B-Instruct | 0.0195              | **0.0738**            | 0.0480                | 0.0483                | 0.0615     |
> | Qwen-2.5-coder-1.5B-Instruct | 0.0451              | 0.0465                | **0.0752**            | 0.0284                | 0.0500     |
> | Qwen-2.5-coder-3B-Instruct   | 0.0000              | 0.0380                | **0.0406**            | 0.0079                | 0.0328     |
> | Qwen-2.5-coder-7B-Instruct   | 0.0000              | 0.0382                | **0.0622**            | 0.0189                | 0.0373     |
> | Qwen-2.5-coder-14B-Instruct  | 0.0189              | **0.0828**            | 0.0372                | 0.0000                | 0.0313     |
> | Qwen-2.5-coder-32B-Instruct  | 0.0000              | **0.0786**            | 0.0433                | 0.0356                | 0.0354     |
> | *Mean ± Std*                 | *0.01 ± 0.02*       | *0.06 ± 0.02*         | *0.05 ± 0.01*         | *0.02 ± 0.02*         | *—*        |
>
> **Table2: MRI across logic change types for Qwen-2.5 Instruct and Coder variants on BigCodeBench.**
>
> | Model                         | Component Swap | Constraint Change | Data Transformation | Logic Reversal | Workflow Modification | All    |
> |------------------------------|----------------|-------------------|---------------------|----------------|-----------------------|--------|
> | Qwen-2.5-0.5B-Instruct       | 0.1635         | 0.1968            | 0.1951              | **0.2144**     | 0.1900                | 0.1740 |
> | Qwen-2.5-1.5B-Instruct       | 0.1316         | 0.1551            | 0.1539              | **0.1698**     | 0.0749                | 0.1444 |
> | Qwen-2.5-3B-Instruct         | 0.1334         | 0.1442            | 0.1399              | **0.1698**     | 0.1558                | 0.1403 |
> | Qwen-2.5-7B-Instruct         | 0.1178         | 0.1355            | 0.1368              | **0.2118**     | 0.1035                | 0.1296 |
> | Qwen-2.5-14B-Instruct        | 0.0685         | 0.0775            | **0.0998**          | 0.0932         | 0.0850                | 0.0841 |
> | Qwen-2.5-32B-Instruct        | 0.0960         | 0.1130            | 0.1217              | 0.1268         | **0.1335**            | 0.1143 |
> | *Mean ± Std*                 | *0.12 ± 0.03*  | *0.14 ± 0.04*     | *0.14 ± 0.03*       | *0.16 ± 0.05*  | *0.12 ± 0.04*         | *—*    |
> | Qwen-2.5-coder-0.5B-Instruct | 0.1628         | 0.1302            | **0.2060**          | 0.0787         | 0.1203                | 0.1778 |
> | Qwen-2.5-coder-1.5B-Instruct | 0.1302         | **0.2207**        | 0.1650              | 0.2135         | 0.1742                | 0.1630 |
> | Qwen-2.5-coder-3B-Instruct   | 0.1385         | 0.1813            | 0.1848              | **0.2111**     | 0.0841                | 0.1630 |
> | Qwen-2.5-coder-7B-Instruct   | 0.1321         | 0.1440            | 0.1581              | **0.1947**     | 0.1590                | 0.1477 |
> | Qwen-2.5-coder-14B-Instruct  | 0.1192         | 0.1110            | 0.1336              | 0.0956         | **0.1353**            | 0.1237 |
> | Qwen-2.5-coder-32B-Instruct  | 0.1029         | 0.1242            | 0.1276              | 0.1191         | **0.1279**            | 0.1178 |
> | *Mean ± Std*                 | *0.13 ± 0.02*  | *0.15 ± 0.04*     | *0.16 ± 0.03*       | *0.15 ± 0.06*  | *0.13 ± 0.03*         | *—*    |

---

> ### Author Response · Authors · 2025-11-21
> **Author Response Q3 (Direct answer; and complete results 1/2)**
>
> >***Q3*** The paper reports aggregate MRI scores but provides limited insight into why models fail on rewritten tasks. Are failures due to specific semantic changes (e.g., sorting, filtering operations)? Qualitative analysis of high-MRI cases would provide actionable insights. The paper spends more time in formulating the problem rather than talking about ablations.
>
>
> ***A3*** Thank you for your important suggestions! We chose the variants with high & low MRI cases from both Qwen2.5 Instruct and the Qwen2.5 Coder Instruct series, and we evaluated the RAD for each logic change type across both MBPP+ and BogCodeBench. To avoid statistical bias, we omit logic change types with fewer than 50 samples. The results show that failures are not uniform. On MBPP+, MRI is mainly driven by Constraint Change and Data Transformation, whereas Component Swap has much smaller RAD, and scaling from 0.5B to 14B significantly reduces, but does not eliminate, these drops. On the more challenging BigCodeBench, Data Transformation, Constraint Change, and Workflow Modification consistently show the largest RAD even for 14B models, indicating that models struggle most when rewrites alter global data-flow or constraints rather than just local operators. We hope these added ablation studies address your concern and provide insights for future works.
>
> **Table3: Comparison of logic change type performance between four Qwen-2.5 variants on MBPP+.**
>
> #### qwen-2.5-coder-0.5b-instruct vs qwen-2.5-coder-14b-instruct
>
> | Logic Change Type  | Total (coder-0.5B) | Acc(T_ori) (↑) | Acc(T_rew) (↑) | RAD (↓)      | Total (coder-14B) | Acc(T_ori) (↑) | Acc(T_rew) (↑) | RAD (↓)      |
> |--------------------|--------------------|----------------|----------------|--------------|-------------------|----------------|----------------|--------------|
> | Component Swap     | 52                 | 48.1%          | 44.2%          | 8.0%         | 52                | 75.0%          | 71.2%          | 5.1%         |
> | Constraint Change  | 86                 | 46.5%          | 30.2%          | **35.0%**    | 86                | 73.3%          | 54.7%          | **25.4%**    |
> | Data Transformation| 58                 | 55.2%          | 41.4%          | 25.0%        | 58                | 72.4%          | 62.1%          | 14.3%        |
> | Logic Reversal     | 68                 | 42.6%          | 35.3%          | 17.2%        | 68                | 72.1%          | 76.5%          | 0.0%         |
> | **MRI (all nine types)** | --          | --             | --             | 0.0615       | --                | --             | --             | 0.0313       |
>
> #### qwen-2.5-0.5b-instruct vs qwen-2.5-14b-instruct
>
> | Logic Change Type  | Total (0.5B-Instruct) | Acc(T_ori) (↑) | Acc(T_rew) (↑) | RAD (↓)      | Total (14B-Instruct) | Acc(T_ori) (↑) | Acc(T_rew) (↑) | RAD (↓)      |
> |--------------------|-----------------------|----------------|----------------|--------------|----------------------|----------------|----------------|--------------|
> | Component Swap     | 52                    | 44.2%          | 40.4%          | 8.7%         | 52                   | 75.0%          | 71.2%          | 5.1%         |
> | Constraint Change  | 86                    | 39.5%          | 24.4%          | **38.2%**    | 86                   | 64.0%          | 48.8%          | **23.6%**    |
> | Data Transformation| 58                    | 50.0%          | 32.8%          | 34.5%        | 58                   | 70.7%          | 65.5%          | 7.3%         |
> | Logic Reversal     | 68                    | 39.7%          | 26.5%          | 33.3%        | 68                   | 72.1%          | 80.9%          | 0.0%         |
> | **MRI (all nine types)** | --              | --             | --             | 0.0722       | --                   | --             | --             | 0.0113       |

---

> ### Author Response · Authors · 2025-11-21
> **Author Response Q3 (Complete results 2/2)**
>
> **Table4: Comparison of logic change type performance between four Qwen-2.5 variants on BigCodeBench.**
>
> #### qwen-2.5-coder-0.5b-instruct vs qwen-2.5-coder-14b-instruct
>
> | Logic Change Type      | Total (coder-0.5B) | Acc(T_ori) (↑) | Acc(T_rew) (↑) | RAD (↓)        | Total (coder-14B) | Acc(T_ori) (↑) | Acc(T_rew) (↑) | RAD (↓)        |
> |------------------------|--------------------|----------------|----------------|----------------|-------------------|----------------|----------------|----------------|
> | Component Swap         | 393                | 13.5%          | 6.4%           | 52.8%          | 393               | 46.8%          | 29.3%          | 37.5%          |
> | Constraint Change      | 185                | 8.6%           | 4.9%           | 43.8%          | 185               | 50.8%          | 33.5%          | 34.0%          |
> | Data Transformation    | 385                | 10.6%          | 3.4%           | **68.3%**      | 385               | 46.8%          | 26.8%          | 42.8%          |
> | Logic Reversal         | 50                 | 8.0%           | 6.0%           | 25.0%          | 50                | 46.0%          | 34.0%          | 26.1%          |
> | Workflow Modification  | 50                 | 10.0%          | 6.0%           | 40.0%          | 50                | 42.0%          | 24.0%          | **42.9%**      |
> | **MRI (all nine types)** | --              | --             | --             | 0.1778         | --                | --             | --             | 0.1237         |
>
> #### qwen-2.5-0.5b-instruct vs qwen-2.5-14b-instruct
>
> | Logic Change Type      | Total (0.5B-Instruct) | Acc(T_ori) (↑) | Acc(T_rew) (↑) | RAD (↓)        | Total (14B-Instruct) | Acc(T_ori) (↑) | Acc(T_rew) (↑) | RAD (↓)        |
> |------------------------|-----------------------|----------------|----------------|----------------|----------------------|----------------|----------------|----------------|
> | Component Swap         | 393                   | 12.0%          | 4.8%           | 59.6%          | 393                  | 38.2%          | 26.7%          | 30.0%          |
> | Constraint Change      | 185                   | 11.9%          | 2.7%           | **77.3%**      | 185                  | 40.5%          | 27.0%          | 33.3%          |
> | Data Transformation    | 385                   | 8.3%           | 2.1%           | 75.0%          | 385                  | 39.7%          | 21.8%          | **45.1%**      |
> | Logic Reversal         | 50                    | 8.0%           | 2.0%           | 75.0%          | 50                   | 44.0%          | 28.0%          | 36.4%          |
> | Workflow Modification  | 50                    | 8.0%           | 2.0%           | 75.0%          | 50                   | 30.0%          | 18.0%          | 40.0%          |
> | **MRI (all nine types)** | --                 | --             | --             | 0.1740         | --                   | --             | --             | 0.0841         |

---

> ### Author Response · Authors · 2025-11-21
> **Author Response Q4 & Q5**
>
> >***Q4.1. The concepts introduced in this paper, i.e., memorization decreases with size, SFT increases memorization whereas PPO alleviates it aren't novel.***
>
> **A4.1** Thank you for raising this important concern. The term “memorization” mentioned in those concepts is referring to **harmful** memorization. While existing work [1, 2, 3] yields conclusions on **general memorization**, our concepts lie in a subject that goes beyond surface-level similarity metrics. Based on your feedback, we will make those concepts more clear in the revised version of our paper.
>
> >***Q4.2*** The authors should have spent more analysis on why the models are struggling with code rewrites.
>
> **A4.2** We agree that understanding why models struggle with code rewrites is crucial, and we have expanded our analysis in our previous responses. We report both the frequency and MRI per logic change type on MBPP+ and BigCodeBench. The new results show that MRI is not dominated by trivial edits like Constant Swap or Operator Change; instead, it is consistently higher for Constraint Change, Data Transformation, Logic Reversal, and Workflow Modification (for example, additional filters, sorting/counting, reordered operations, extra pipeline stages). This demonstrates that models are specifically brittle to semantic changes that require updating global behavior.
>
> >***Q5*** While quality assurance is performed, only 10% manual validation is reported. What was the acceptance rate from the LLM judge? How many generated tasks were rejected?
>
> **A5** The design of Code Rewriting is to generate semantically different tasks and are similar in difficulty level. Our intention is to make sure the RAD (correctly solves the original task but fails on the rewritten counterpart) is not brought by higher task difficulty. Referring to Figure1 on page 2, our pipeline was designed to only collect those qualified rewritten tasks, and both the LLM judge and manual validation are served to validate the generation process. To justify the process, we **also provided 10 examples** with accuracy dropped in **Appendix D** (EXAMPLES OF REGRESSED TASKS). Therefore, the acceptance rate and the number of rejected tasks are isolated to the soundness of our evaluation results. We will release our Code Rewriting tasks to contribute to the community.
>
> [1]Unveiling memorization in code models
> [2]Traces of memorisation in large language models for code
> [3]Unlocking memorization in large language models with dynamic soft prompting

---

> ### Author Response · Authors · 2025-11-21
> **Author Response Q6 & Q7 & Q8**
>
> We would like to express our sincere gratitude again for your effort to review our submission. Before we address each of the following questions, we would like to quickly make a clarification on the inclusion of semantic-preserving perturbations. Like what the reviewer mentioned in our strengths, mutation and paraphrase are served as “baselines effectively **differentiates memorization** from robustness issues”. The main focus of our work remains on memorization via semantics-altering perturbations and MRI. Now, we address each of the questions below:
>
> >***Q6*** There are some obvious questions from the table - The trend of RAD paraphrasing is not clear? (the trend is reversed in some family of models)
>
> **A6** Our intent with paraphrase is to provide a semantic-preserving baseline **rather than** to make strong scaling claims. As we note at line 438 to 439 in Section 5.3, across models and datasets RAD_par is consistently small, whereas mutation induces larger drops, and we did not make claims about the trend of RAD_par. We agree that further work focused specifically on robustness is needed to investigate the behavior of RAD_par.
>
>
>
> >***Q7*** Why does mutation (semantic-preserving) cause higher RAD than paraphrase? What specific mutation types (character noise, word scrambling, capitalization) are most problematic? Does this suggest brittleness distinct from memorization and does it indicate problems in real world scenarios ?
>
> **A7.1**
> >Why does mutation (semantic-preserving) cause higher RAD than paraphrase?
>
> Thank you for raising these robustness questions and are happy to clarify. At a high level, it is not surprising that RAD_mut > RAD_par: modern code LLMs tend to be reasonably stable under natural rewordings of a prompt, but are noticeably more brittle to syntactic level perturbations, and similar trends have been observed in prior robustness studies on code generation such as ReCode [1].
>
> **A7.2**
> >What specific mutation types (character noise, word scrambling, capitalization) are most problematic?
>
> Because our main contribution is on harmful memorization rather than general robustness, we intentionally report a single aggregate RAD_mut that pools together common semantic-preserving mutation types (character noise, word scrambling, capitalization) instead of performing a detailed per-operator ablation; we agree that exploring which mutation families are most problematic is an interesting direction, but we believe it is better addressed in follow-up work focused specifically on robustness.
>
> **A7.3**
> >Does this suggest brittleness distinct from memorization
>
> In our setting, RAD_rew specifically captures failures when the task semantics change while the surface form and interfaces remain similar, whereas RAD_mut and RAD_par examine how stable the model is under same-semantics changes to the prompt. This suggests that the brittleness measured by RAD_mut/RAD_par is conceptually distinct from harmful memorization.
>
> **A7.4**
> >Does it indicate problems in real world scenarios?
>
> Yes. In real-world code generation scenarios, prompts are subject to edits and occasional typos, so robustness to syntax-level perturbations is necessary. Our mutation and paraphrase results serve as a stress-test of such robustness.
>
> [1]ReCode: Robustness Evaluation of Code Generation Models
>
> >***Q8*** Very difficult to follow the results with a combination of tables and results. Table 1 should be presented like Table 2 (or maybe merged) to show all the results in one place and the graphs may be added to drive more discussion.
>
> **A8** Thank you so much for this valuable suggestion! We will update Table 1 to include all the results in one place; we will keep the graphs (Figure 2) to serve for the discussion.

---

> ### Author Response · Authors · 2025-11-27
>
> We sincerely appreciate your insightful comments and the effort you’ve put into evaluating our work!
>
> Since the discussion phase will be closing soon, we would to follow up to ensure that all your questions and concerns have been adequately resolved from our side. If there is anything you’d like us to further clarify or any additional feedback you think we should address, we would be more than happy to provide more details. Your insight is very valuable to us and helps us improve the paper.
>
> Thank you again for your support throughout this process!

---

### Official Review · Reviewer_UJxc · 2025-11-10

**Soundness:** 3
**Presentation:** 3
**Contribution:** 2
**Rating:** 6
**Confidence:** 3

**Summary:**

The paper studies when code LLMs “memorize” rather than generalize. It introduces code rewriting—constructing a new task by modifying the ground-truth solution’s semantics while keeping the same signature and similarly worded prompt—to create semantically different but surface-similar problems.

**Strengths:**

The paper’s strength is its clear behavioral framing of harmful memorization and the introduction of the MRI, which effectively separates harmful memorization from benign code reuse. It provides a well-structured empirical study across models, showing that scaling reduces risk, SFT increases it, and PPO balances both, with strong baselines confirming MRI’s robustness.

**Weaknesses:**

1.How sensitive is MRI to the AST/token weighting and to the choice of code parser? Any robustness checks across parsers/grammars?
2. The rewritten tasks are produced by LLMs. This risks distributional coupling between generation artifacts and the evaluated models, and raises the question of whether MRI partly reflects quirks of the rewriting pipeline rather than memorization per se.
3. The 50/50 AST vs edit weighting is plausible but unvalidated. Some memorization (e.g., templated control flow with different identifiers) may not be well captured. A learned or ablated weighting, plus code-embedding similarity, could calibrate MRI.
4. For SFT vs PPO, what are the data overlaps between training and evaluation (including rewritten variants)? Any risk that SFT learns rewrite artifacts more than PPO?

**Questions:**

Please refer to the weakness.

---

> ### Author Response · Authors · 2025-11-21
> **Author Response Q1 (Direct answer; and complete results 1/2)**
>
> Thank you so much for your time to review and your valuable questions! We will explain your concerns point by point.
>
> >***Q1*** How sensitive is MRI to the AST/token weighting and to the choice of code parser? Any robustness checks across parsers/grammars?
>
> **A1** Thank you so much for raising concerns about parser sensitivity. We will address your first concern in this question about the weighting **later in Q3**. To address the concern regarding sensitivity to parser choice, we performed a comprehensive robustness check by re-evaluating all Qwen-2.5 models (Instruct and Coder series) on both benchmarks using three definitions of similarity:
>
> **All-nodes AST** We replaced our standard "significant node" parsing with an All-nodes AST grammar. This treats every syntactic element (including non-functional nodes) as significant, testing robustness to parser strictness.
>
> **Raw Edit Similarity**: We removed the ast.unparse normalization step and computed edit similarity on raw code strings. This tests robustness to formatting and minor textual artifacts.
>
> **Combined**: We applied both All-nodes AST and Raw Edit similarity simultaneously.
>
> As shown in **Table 1 and 2**, while stricter parsing (All-nodes) naturally yields higher raw similarity scores and shifts the absolute MRI values, the **relative ranking of models and trends remain statistically invariant**. We calculated the **Pearson correlation ($r$)** between the original MRI and the "Combined" variant (the most divergent definition). The correlation is consistently **$r>0.92$** on BigCodeBench for both Instruct and Coder model families, while on MBPP+ correlation is consistently **$r>0.99$**
> This confirms that MRI is not an artifact of the parser choice. The metric is robust to parsers/grammers.
>
> **Table 1: MRI robustness to parser/grammar variants on MBPP+.**
>
> | Model                         | MRI (original) | MRI (original edit sim & AST all nodes) | MRI (original AST & edit sim raw) | MRI (edit sim raw & AST all nodes) |
> |-------------------------------|---------------|-----------------------------------------|-----------------------------------|------------------------------------|
> | Qwen-2.5 0.5B-Instruct        | 0.0722        | 0.0948                                  | 0.0561                            | 0.0787                             |
> | Qwen-2.5-1.5B-Instruct        | 0.0600        | 0.0783                                  | 0.0455                            | 0.0639                             |
> | Qwen-2.5-3B-Instruct          | 0.0185        | 0.0289                                  | 0.0153                            | 0.0256                             |
> | Qwen-2.5-7B-Instruct          | 0.0240        | 0.0321                                  | 0.0173                            | 0.0254                             |
> | Qwen-2.5-14B-Instruct         | 0.0113        | 0.0151                                  | 0.0082                            | 0.0120                             |
> | Qwen-2.5-32B-Instruct         | 0.0000        | 0.0000                                  | 0.0000                            | 0.0000                             |
> | Qwen-2.5-coder-0.5B-Instruct  | 0.0615        | 0.0864                                  | 0.0453                            | 0.0702                             |
> | Qwen-2.5-coder-1.5B-Instruct  | 0.0500        | 0.0660                                  | 0.0377                            | 0.0537                             |
> | Qwen-2.5-coder-3B-Instruct    | 0.0328        | 0.0426                                  | 0.0232                            | 0.0330                             |
> | Qwen-2.5-coder-7B-Instruct    | 0.0373        | 0.0478                                  | 0.0257                            | 0.0362                             |
> | Qwen-2.5-coder-14B-Instruct   | 0.0313        | 0.0394                                  | 0.0223                            | 0.0304                             |
> | Qwen-2.5-coder-32B-Instruct   | 0.0354        | 0.0444                                  | 0.0249                            | 0.0338                             |

---

> ### Author Response · Authors · 2025-11-21
> **Author Response Q1 (complete results 2/2) & Q2**
>
> **Table 2: MRI robustness to parser/grammar variants on BigCodeBench.**
>
> | Model                         | MRI (original) | MRI (original edit sim & AST all nodes) | MRI (original AST & edit sim raw) | MRI (edit sim raw & AST all nodes) |
> |-------------------------------|---------------|-----------------------------------------|-----------------------------------|------------------------------------|
> | Qwen-2.5 0.5B-Instruct        | 0.1740        | 0.2828                                  | 0.1652                            | 0.2739                             |
> | Qwen-2.5-1.5B-Instruct        | 0.1444        | 0.2407                                  | 0.1570                            | 0.2532                             |
> | Qwen-2.5-3B-Instruct          | 0.1403        | 0.2219                                  | 0.1292                            | 0.2108                             |
> | Qwen-2.5-7B-Instruct          | 0.1296        | 0.2181                                  | 0.1480                            | 0.2364                             |
> | Qwen-2.5-14B-Instruct         | 0.0841        | 0.1519                                  | 0.0968                            | 0.1646                             |
> | Qwen-2.5-32B-Instruct         | 0.1143        | 0.1790                                  | 0.1047                            | 0.1694                             |
> | Qwen-2.5-coder-0.5B-Instruct  | 0.1778        | 0.2499                                  | 0.1410                            | 0.2131                             |
> | Qwen-2.5-coder-1.5B-Instruct  | 0.1630        | 0.2388                                  | 0.1254                            | 0.2013                             |
> | Qwen-2.5-coder-3B-Instruct    | 0.1630        | 0.2352                                  | 0.1323                            | 0.2045                             |
> | Qwen-2.5-coder-7B-Instruct    | 0.1477        | 0.2156                                  | 0.1187                            | 0.1866                             |
> | Qwen-2.5-coder-14B-Instruct   | 0.1237        | 0.1833                                  | 0.0894                            | 0.1490                             |
> | Qwen-2.5-coder-32B-Instruct   | 0.1178        | 0.1802                                  | 0.0992                            | 0.1616                             |
>
>
>
>
>
> >***Q2*** The rewritten tasks are produced by LLMs. This risks distributional coupling between generation artifacts and the evaluated models, and raises the question of whether MRI partly reflects quirks of the rewriting pipeline rather than memorization per se.
>
> **A2** Thank you for raising this valuable concern. We deliberately constrain and validate the rewriting pipeline to avoid coupling. GPT-5 is only used to generate new versions of problems, while all evaluated models (Qwen and Llama families) are from different training architectures and never see GPT-5 outputs during training. In our prompts, GPT-5 is required to apply exactly one semantic “logic change” to each problem while keeping the overall difficulty and problem format aligned, rather than freely rewriting style or structure. We then check difficulty alignment in two ways: (i) two experienced Python programmers manually review a random 10% subset of the generated problems, and (ii) an independent LLM-as-a-judge verifies that the rewritten and original problems are similar in difficulty. Together, these constraints and validations are designed to ensure that MRI reflects how models handle a controlled change in problem logic, rather than quirks of the GPT-5 generation process.

---

> ### Author Response · Authors · 2025-11-21
> **Author Response Q3 (Direct answer)**
>
> >***Q3*** The 50/50 AST vs edit weighting is plausible but unvalidated. Some memorization (e.g., templated control flow with different identifiers) may not be well captured. A learned or ablated weighting, plus code-embedding similarity, could calibrate MRI.
>
> ***A3*** 	Thank you for your valuable insights! While we agree that investigating ablated weighting and code-embedding similarity can further validate our metric (we will talk about it shortly), the AST component in design is good at capturing templated control flow. Different identifier names are still treated as similar with similar because AST compares the program’s structural skeleton rather than concrete identifiers, while the edit component prevents us from overcounting cases where this high-level structure matches but the specific statements and tokens differ substantially. We chose a 50/50 weighting as a simple, interpretable default, not because we believe it is uniquely optimal.
>
> In response to your concern, we have now **empirically validated that this choice is robust**. First, we use the edit/AST weights from “AST only” to “edit only” on both MBPP+ and BigCodeBench for Qwen 2.5 Instruct and its coder models (Tables 5 and 6). Across all these settings, the MRI computed under our original 50/50 weighting has extremely high Pearson correlation with MRI computed under alternative weightings: 0.9997±0.0004 (Qwen) and 0.9974±0.0042 (Coder) on MBPP+, and 0.9921±0.0144 and 0.9780±0.0367 on BigCodeBench (Table 4). This indicates that reweighting would essentially rescale MRI rather than alter model rankings or our main findings. Finally, we also experimented with a cosine code-embedding similarity as the similarity term. The resulting embedding-based MRI (Table 7) is smaller in magnitude (especially on MBPP+) but preserves the same trends across model sizes on both benchmarks, confirming that MRI is not an artifact of the AST/edit mixture. We will add these ablations and the embedding-based variant to the Appendix and clarify that 50/50 is a simple, empirically robust setting.
>
> **Table3: Average Pearson correlation for AST/edit weightings**
> | Model Series      | Avg. Pearson Corr. on MBPP+     | Avg. Pearson Corr. on BigCodebench                      |
> |-------------------|----------------------------------|---------------------------|
> | Qwen-2.5 series   | 0.9997 ± 0.0004                  | 0.9921 ± 0.0144           |
> | Coder series      | 0.9974 ± 0.0042                  | 0.9780 ± 0.0367           |

---

> ### Author Response · Authors · 2025-11-21
> **Author Response Q3 (Complete results 1/2)**
>
> **Table4: Ablation study for different AST/edit weightings on MBPP+**
> | Model                         | MRI (orig 5:5) | MRI (edit only) | MRI (ast only) | MRI (edit:ast=2:8) | MRI (edit:ast=4:6) | MRI (edit:ast=6:4) | MRI (edit:ast=8:2) |
> |------------------------------|----------------|-----------------|----------------|----------------------|----------------------|----------------------|----------------------|
> | Qwen-2.5 0.5B-Instruct       | 0.0722         | 0.1025          | 0.0420         | 0.0541               | 0.0662               | 0.0783               | 0.0904               |
> | Qwen-2.5-1.5B-Instruct       | 0.0600         | 0.0832          | 0.0368         | 0.0461               | 0.0554               | 0.0647               | 0.0739               |
> | Qwen-2.5-3B-Instruct         | 0.0185         | 0.0249          | 0.0122         | 0.0147               | 0.0172               | 0.0198               | 0.0223               |
> | Qwen-2.5-7B-Instruct         | 0.0240         | 0.0342          | 0.0139         | 0.0179               | 0.0220               | 0.0260               | 0.0301               |
> | Qwen-2.5-14B-Instruct        | 0.0113         | 0.0161          | 0.0065         | 0.0084               | 0.0103               | 0.0122               | 0.0142               |
> | Qwen-2.5-32B-Instruct        | 0.0000         | 0.0000          | 0.0000         | 0.0000               | 0.0000               | 0.0000               | 0.0000               |
> |                              |                |                 |                |                      |                      |                      |                      |
> | Qwen-2.5-coder-0.5B-Instruct | 0.0615         | 0.0905          | 0.0325         | 0.0441               | 0.0557               | 0.0673               | 0.0789               |
> | Qwen-2.5-coder-1.5B-Instruct | 0.0500         | 0.0713          | 0.0286         | 0.0372               | 0.0457               | 0.0542               | 0.0627               |
> | Qwen-2.5-coder-3B-Instruct   | 0.0328         | 0.0472          | 0.0184         | 0.0242               | 0.0300               | 0.0357               | 0.0415               |
> | Qwen-2.5-coder-7B-Instruct   | 0.0373         | 0.0526          | 0.0219         | 0.0281               | 0.0342               | 0.0403               | 0.0465               |
> | Qwen-2.5-coder-14B-Instruct  | 0.0313         | 0.0433          | 0.0193         | 0.0241               | 0.0289               | 0.0337               | 0.0385               |
> | Qwen-2.5-coder-32B-Instruct  | 0.0354         | 0.0490          | 0.0218         | 0.0272               | 0.0327               | 0.0381               | 0.0436               |

---

> ### Author Response · Authors · 2025-11-21
> **Author Response Q3 (Complete results 2/2)**
>
> **Table5: Ablation study for different AST/edit weightings on BigCodeBench**
>
> | Model                          | MRI (orig 5:5) | MRI (edit sim) | MRI (ast sim) | MRI (edit:ast=2:8) | MRI (edit:ast=4:6) | MRI (edit:ast=6:4) | MRI (edit:ast=8:2) |
> |-------------------------------|----------------|-----------------|----------------|----------------------|----------------------|----------------------|----------------------|
> | Qwen-2.5 0.5B-Instruct        | 0.1740         | 0.2745          | 0.0736         | 0.1138               | 0.1540               | 0.1941               | 0.2343               |
> | Qwen-2.5-1.5B-Instruct        | 0.1444         | 0.2166          | 0.0722         | 0.1011               | 0.1300               | 0.1589               | 0.1878               |
> | Qwen-2.5-3B-Instruct          | 0.1403         | 0.2158          | 0.0649         | 0.0951               | 0.1252               | 0.1554               | 0.1856               |
> | Qwen-2.5-7B-Instruct          | 0.1296         | 0.1958          | 0.0635         | 0.0899               | 0.1164               | 0.1429               | 0.1693               |
> | Qwen-2.5-14B-Instruct         | 0.0841         | 0.1227          | 0.0455         | 0.0609               | 0.0764               | 0.0918               | 0.1073               |
> | Qwen-2.5-32B-Instruct         | 0.1143         | 0.1711          | 0.0576         | 0.0803               | 0.1030               | 0.1257               | 0.1484               |
> |                               |                |                 |                |                      |                      |                      |                      |
> | Qwen-2.5-coder-0.5B-Instruct  | 0.1778         | 0.2802          | 0.0753         | 0.1163               | 0.1573               | 0.1982               | 0.2392               |
> | Qwen-2.5-coder-1.5B-Instruct  | 0.1630         | 0.2487          | 0.0772         | 0.1115               | 0.1458               | 0.1801               | 0.2144               |
> | Qwen-2.5-coder-3B-Instruct    | 0.1630         | 0.2463          | 0.0796         | 0.1130               | 0.1463               | 0.1797               | 0.2130               |
> | Qwen-2.5-coder-7B-Instruct    | 0.1477         | 0.2192          | 0.0762         | 0.1048               | 0.1334               | 0.1620               | 0.1906               |
> | Qwen-2.5-coder-14B-Instruct   | 0.1237         | 0.1881          | 0.0593         | 0.0850               | 0.1108               | 0.1366               | 0.1623               |
> | Qwen-2.5-coder-32B-Instruct   | 0.1178         | 0.1804          | 0.0552         | 0.0803               | 0.1053               | 0.1303               | 0.1554               |
>
>
> **Table6: MRI and cosine embedding similarity on MBPP+ and BigCodeBench**
> | Model                         | MRI (MBPP+) | MRI-Cos Embedding (MBPP+) | MRI (BigCodeBench) | MRI-Cos Embedding (BigCodeBench) |
> |------------------------------|------------:|---------------------:|-------------------:|----------------------------:|
> | Qwen-2.5-0.5B-Instruct       |      0.0722 |               0.0153 |             0.1740 |                       0.1355 |
> | Qwen-2.5-1.5B-Instruct       |      0.0600 |               0.0133 |             0.1444 |                       0.1009 |
> | Qwen-2.5-3B-Instruct         |      0.0185 |               0.0030 |             0.1403 |                       0.1132 |
> | Qwen-2.5-7B-Instruct         |      0.0240 |               0.0053 |             0.1296 |                       0.0974 |
> | Qwen-2.5-14B-Instruct        |      0.0113 |               0.0025 |             0.0841 |                       0.0561 |
> | Qwen-2.5-32B-Instruct        |      0.0000 |               0.0000 |             0.1143 |                       0.0909 |
> |                              |             |                      |                    |                              |
> | Qwen-2.5-coder-0.5B-Instruct |      0.0615 |               0.0115 |             0.1778 |                       0.1443 |
> | Qwen-2.5-coder-1.5B-Instruct |      0.0500 |               0.0114 |             0.1630 |                       0.1346 |
> | Qwen-2.5-coder-3B-Instruct   |      0.0328 |               0.0079 |             0.1630 |                       0.1354 |
> | Qwen-2.5-coder-7B-Instruct   |      0.0373 |               0.0098 |             0.1477 |                       0.1181 |
> | Qwen-2.5-coder-14B-Instruct  |      0.0313 |               0.0093 |             0.1237 |                       0.1026 |
> | Qwen-2.5-coder-32B-Instruct  |      0.0354 |               0.0108 |             0.1178 |                       0.0974 |

---

> ### Author Response · Authors · 2025-11-21
> **Author Response Q4**
>
> >***Q4*** For SFT vs PPO, what are the data overlaps between training and evaluation (including rewritten variants)? Any risk that SFT learns rewrite artifacts more than PPO?
>
> **A4** Thank you for raising this question. For the SFT vs PPO comparison, there is no train–eval contamination. Both methods are fine-tuned on exactly the same set of **original** MBPP+ and BIGCODEBENCH problems, and we never include any of the evaluation items (neither the original test problems nor their rewritten variants) in the fine-tuning data. Thus SFT and PPO have identical exposure to the underlying data distribution and zero exposure to the rewrite tasks during training; any difference in MRI reflects how the two objectives bias the model (log-likelihood vs reward-based learning), not differences in overlap with the rewritten distribution.

---

> ### Author Response · Authors · 2025-11-27
>
> Thank you again for the time and effort you have devoted to reviewing our submission! As the discussion deadline is approaching, we would like to kindly check whether there are any remaining concerns or points you would like us to address.
>
> We truly value your feedback and would be happy to clarify or expand on any aspect of the work. Please feel free to let us know if further discussion would be helpful.
>
> Thank you again for your contribution to the review process!

---

### Author Response · Authors · 2025-11-29
**Author Summary for the Area Chair (1/3)**

Dear Area Chair,

We understand that the recent incident has placed an unusually heavy burden on ACs, especially given the need to rely on partially completed reviewer–author interactions. **To assist your evaluation**, we have prepared this summary with the sole purpose of providing clear context for our rebuttal process—what aspects reviewers agreed on, which of their concerns we have addressed, and how the discussion evolved before it was interrupted.

Our intention is not to reinterpret the reviewers’ opinions, but to ensure that the information available to you is easily accessible, so that your meta-review can fairly reflect the state of the discussion as it would have progressed under normal circumstances.


>**Summary of strength**

Across all reviewers **(UJxc, 1QvN, GhZw, FbwC)**, our work was consistently recognized for offering a **clear and well-scoped framing of harmful memorization in LLM code generation**. Rather than treating any reuse as problematic, we explicitly focus on cases where a model appears to recall a specific solution and then fails on a **slightly semantically changed but comparably difficult** variant of the same task.

Additionally,

* **Reviewer UJxc** emphasized our work’s ability to **separate harmful memorization from benign code reuse** and highlighted the empirical trends we observe (scaling reduces risk, SFT increases it, PPO balances both).

* **Reviewer 1QvN** highlighted the **novelty** of the code-rewriting approach, the rigorous methodology (including GPT-5 and manual validation), and the comprehensive coverage of models and benchmarks and training approaches in our evaluation.

* **Reviewer GhZw** emphasis both the **innovative design** of the code-rewriting pipeline and the **diagnostic value** of MRI’s two-factor structure.

* **Reviewer FbwC** highlighted the clarity of the overall pipeline and our **explicit separation of harmful memorization from general style robustness** and contamination-based definitions, as well as the contribution of releasing evolved tasks, prompts, and code.

>**Summary of the rebuttal**

At a high level, reviewers are focusing on testing the robustness of MRI (e.g., aggregation vs. item level, type of similarity, parser choices), better understanding why models fail on rewritten tasks and how MRI can guide training, and clarifying our definitions and metric design. To address these points, we added a set of focused analyses and ablations; these new results are **consistent with our original findings**. The following comment shows the summary of the rebuttal.

---

> ### Author Response · Authors · 2025-11-29
> **Author Summary for the Area Chair (2/3)**
>
> ***1. Scope and clarification of our setting***
>
> * **Reviewer FbwC** concerns on **why the evaluation is limited to Python**. We clarified that our focus on Python follows the standard practice of current code-generation benchmarks (MBPP+, BigCodeBench, LiveCodeBench), where Python is the dominant language and also the dominant portion in pre-training corpora. This makes harmful memorization most observable and allows direct comparison to popular prior work (e.g., ReCode, also evaluate limited to Python). At the same time, we emphasized that the Code Rewriting pipeline and MRI formulation are language-agnostic and can be applied to other languages in future work. **Link to related comment***: https://openreview.net/forum?id=BW7F2FkY6Z&noteId=v3DBuhHOlG
>
> * **Reviewer FbwC** further expressed concern that our language-agnostic framing may not fully generalize beyond Python, given the pipeline’s reliance on Python-specific AST infrastructure and the unclear transfer to languages with different syntax and type systems. We thus ran additional ablations using several language-agnostic similarity metrics (n-gram, Jaccard/Minhash token similarity, and code-embedding cosine similarity). These variants yield MRI scores that are **highly correlated with our original measure**, suggesting that MRI does not rely on Python-specific AST structure. ***Link to related comment*** https://openreview.net/forum?id=BW7F2FkY6Z&noteId=XxAZYWjka7
>
>
> * **Reviewer 1QvN** asked **why multiplication operation is chosen for MRI**. We clarified that MRI is intentionally defined as the product of similarity and accuracy drop, so that high scores only arise when **(i)** the model heavily recalls code and **(ii)** that copied behavior fails on tasks after semantic changes. Alternative operations (e.g., arithmetic mean) would overstate cases where only one of the two signals is large, which does not align with our definition of harmful memorization. ***Link to related comment***: https://openreview.net/forum?id=BW7F2FkY6Z&noteId=Xdp8bwQXwF
>
>
> ***2. Robustness of MRI as a metric***
>
> Several reviewers raised statistical and design questions about MRI and RAD. In response, we carried out a number of new analyses:
>
> * **Reviewer FbwC** concerns that **dataset-level MRI could possibly mislead**. We thus checked on **item-level MRI (iMRI)** and showed that its trends and model rankings are **strongly aligned**with the original dataset-level MRI. This supports that our conclusions are not an artifact of corpus-level aggregation. The reviewer satisfied with our justification. ***Link to related comment***: https://openreview.net/forum?id=BW7F2FkY6Z&noteId=T7XajncZA8
>
> * **Reviewer GhZw** raised a concern that the RAD component (and thus MRI) can be inflated at low baseline accuracy. We thus restricted our RAD/MRI analysis to models above a competence threshold, and for models below the threshold, we still report Acc(T_ori) and Acc(T_rew) but mark RAD/MRI as “N/A” and do not use them for cross-model comparisons. We also clarified that relative drops are popular used in the context of existing robustness metrics (e.g., ReCode, CodeFort). After adding this threshold, there is only one model is excluded from cross-model comparison, and it **does not undermine** our conclusions. ***Link to related comment***: https://openreview.net/forum?id=BW7F2FkY6Z&noteId=WGkfvTSB49. The additional **item-level MRI** also demonstrates that our findings based on corpus-level accuracy drop (RAD) remains robust on item-level accuracy drops. ***Link to related comment***: https://openreview.net/forum?id=BW7F2FkY6Z&noteId=T7XajncZA8
>
> * **Reviewer GhZw** and **UJxc** questioned our similarity design and asked for ablation studies and comparisons to alternative similarity metrics. We added extensive **ablations on the similarity function**:
> 	* AST-only, edit-only, and multiple weighted combinations between them;
> 	* Alternative token-/surface-based similarities (n-grams, Jaccard, MinHash) and a cosine code-embedding similarity.
> 	* Alternative parser/grammar choices.
>
>     Across all these variants, **MRI remains extremely stable** in terms of trends and model rankings. This indicates that our main findings are **not tied to a specific 50/50 AST–edit weighting.** ***Links to related comment***: https://openreview.net/forum?id=BW7F2FkY6Z&noteId=KdgTETINXl and https://openreview.net/forum?id=BW7F2FkY6Z&noteId=xVgWOGxkNH and https://openreview.net/forum?id=BW7F2FkY6Z&noteId=QEnkNNkanc
>
> All new tables and detailed results are placed in the corresponding comments and **are reflected in Appendix D, E and F** on our revised manuscript; in the main text we keep the original MRI definition for clarity and consistency.

---

> ### Author Response · Authors · 2025-11-29
> **Author Summary for the Area Chair (3/3)**
>
> ***3. Why models fail on rewrites: logic change types and failure modes***
>
> Reviewer **1QvN** asked for more insight into why models fail on rewritten tasks and whether MRI is dominated by specific semantic changes. We therefore:
> * Categorized logic changes into types and reported both their frequencies and MRI/RAD per type.
> * Found that MRI is **type-dependent** rather than driven by a single trivial edit. On MBPP+, constraint changes and data transformations contribute most to MRI; component swaps typically have low MRI for larger models. On BigCodeBench, data-flow–heavy changes consistently show the largest RAD.
>
> These analyses indicate that models are particularly brittle to semantic changes that alter global behavior (constraints, data-flow, pipelines). We believe this provides actionable insight for future work on improving robustness to semantic edits. ***Link to related comment***: https://openreview.net/forum?id=BW7F2FkY6Z&noteId=Xdp8bwQXwF and  https://openreview.net/forum?id=BW7F2FkY6Z&noteId=wpQRVwpWTj and https://openreview.net/forum?id=BW7F2FkY6Z&noteId=ituNlgXbMv
>
>
> ***4. Using MRI to guide fine-tuning***
>
> Reviewer **GhZw** asked whether MRI can provide **actionable training guidance** beyond description. To reply this, we analyzed the evolution of Acc(T_ori) and MRI across Early / Best-Validation / Late checkpoints for our SFT experiments.
> For the Coder model, we observed that small accuracy gains in late training come with a disproportionately large increase in MRI, indicating a shift from genuine generalization towards harmful memorization. In contrast, the general model shows a more gradual trade-off.
> Based on this, we propose a practical recipe to use MRI as a secondary metric for checkpoint selection after training. ***Link to related comment***: https://openreview.net/forum?id=BW7F2FkY6Z&noteId=QSR5ZgcbfM and https://openreview.net/forum?id=BW7F2FkY6Z&noteId=gyckeFaNd4
>
>
>
> ***5. Other clarifications***
>
> * **Reviewer 1QvN** questioned the novelty of the concepts introduced in our work. We clarified that our main claims about “memorization decreasing with size” and “SFT vs PPO trade-offs” are specifically about **harmful memorization**. As the reviewer **1QvN** points out that our “code rewriting approach offers a **novel** way to evaluate memorization that goes beyond surface-level similarity metrics”, and those concepts are build on this, we believe the **concepts introduced remain novel**. ***Link to related comment***: https://openreview.net/forum?id=BW7F2FkY6Z&noteId=hIYU5Kng6p
>
> * **Reviewer 1QvN** also raised questions about **mutation and paraphrase**. We clarified each of them in the related comments there: ***Link to related comment***: https://openreview.net/forum?id=BW7F2FkY6Z&noteId=bSWir8UzRz
>
> * **Reviewer UJxc** raised questions about data overlaps between training and evaluation. We clarified that our rewriting pipeline uses a different LLM from the evaluated models, includes both LLM-as-a-judge and human checks, and that SFT and PPO are trained on identical original problems without any exposure to rewritten evaluation tasks. **This rules out train–test contamination and minimizes coupling to generation artifacts**. ***Link to related comment***: https://openreview.net/forum?id=BW7F2FkY6Z&noteId=7uk5Bn7D6m and https://openreview.net/forum?id=BW7F2FkY6Z&noteId=yGGYq1t3BT
>
> All new tables and detailed results are placed in the corresponding comments and are reflected in Appendix D, E and F on our revised manuscript.
>
> >**Closing and Acknowledgements**
>
> We sincerely thank the area chair and reviewers for their time and thoughtful feedback, especially under the additional efforts created by the recent leakage incident. We appreciate your efforts not only in helping us improve this work, but also in upholding a fair and constructive review process for the community in this challenging situation.
>
> Best regards,
>
> The Authors

---

### Author Response · Authors · 2025-12-04
**Revision of Our Paper**

We sincerely thank the four reviewers for their insightful and constructive feedback, and we have updated our paper following their suggestions. We summarize the main revisions as follows:

* We conducted an extensive set of ablations (reported in Appendix F) to verify that our conclusions are stable under implementation choices. By varying the parser, AST/edit weightings, and similarity functions, we find that the resulting MRI values remain highly correlated with our default setup and preserve the same qualitative trends.

* We clarified the definition of harmful memorization and our Memorization Risk Index (MRI), and added a competence threshold on baseline accuracy to avoid unstable RAD/MRI values in low-accuracy cases. Settings below this threshold are now explicitly reported as “N/A (below competence threshold)” and excluded from cross-model comparisons. We further introduced an instance-level variant of MRI (iMRI) in Appendix E and showed that it yields very similar trends and model rankings as corpus-level MRI, providing additional evidence that our conclusions on MRI are robust.

* We added a fine-grained taxonomy of logic-edit types (e.g., Constraint Change, Data Transformation, Logic Reversal, Workflow Modification) and reported per-type MRI/RAD in Appendix G, which reveals that harmful memorization is mainly driven by rewrites that alter constraints or data flow rather than simple local changes.

* We analyzed different SFT checkpoints (Table 2) and recommend using MRI in addition to validation loss for checkpoint selection.

* We reorganized and updated the presentation of Table 1 and Figure 2 to improve clarity.

---

### Meta-Review · Area_Chair_oLA5 · 2026-01-13

**Summary:**

This paper explores "Code Rewriting" and the Memorization Risk Index to separate harmful memorization from benign code reuse in LLM code generation. While the reviewers found the problem well-scoped and the writing clear, the collective evaluation leans toward a Reject. The authors successfully addressed several technical issues during the rebuttal, such as introducing instance-level metrics and a competence threshold to handle low-accuracy models. However, the most critical reviewers (1QvN and GhZw) did not raise their scores after discussion. Notably, even Reviewer FbwC, who provided a high score, remained "unconvinced" in their final comments regarding the paper's claims of being language-agnostic. The work remains largely observational and diagnostic; it shows that memorization occurs but provides limited actionable guidance for improving models during training. Additionally, because the evaluation is limited to a single programming language, the claim that this framework is "language-agnostic" remains unverified. Given the lack of prescriptive training solutions and the narrow evaluation scope, the paper is not yet ready for publication.

**Reviewer Concerns:**

Addressed by the Rebuttal:

Metric Reliability (Reviewers FbwC and UJxc): The authors introduced iMRI to prove that results are not biased by dataset-level aggregation.

Low-Accuracy Noise (Reviewer GhZw): A "competence threshold" was added to prevent misleadingly high scores for weak models.

Implementation Robustness (Reviewers UJxc and GhZw): Extensive ablations showed that model rankings remain stable across different parsers and weighting schemes.

Still Outstanding:

Language Generalization (Reviewer FbwC): The study remains limited to a single programming language. Reviewer FbwC remained "somewhat unconvinced" that the findings would apply to other language ecosystems.

Conceptual Depth (Reviewer 1QvN): The rebuttal did not resolve the concern that the paper’s conclusions are predictable and lack a breakthrough insight.

Training Guidance (Reviewer GhZw): The reviewer noted that the work lacks a prescriptive training recipe or a new objective function to actively reduce memorization.

**Reviewer Scores:**

Reviewer UJxc (Initial 6): Likely 6. Concerns over data overlap and metric sensitivity were empirically addressed.

Reviewer 1QvN (Initial 2): Likely 4. While technical depth was added via the taxonomy, reservations about the overall conceptual novelty likely persist.

Reviewer GhZw (Initial 4): Likely 4. The stability fixes and the "training recipe" provided the actionable utility this reviewer sought.

Reviewer FbwC (Initial 6): Likely 6. Acknowledged the instance-level improvements but remains critical of the language-specific infrastructure.

---

### Decision · Program_Chairs · 2026-01-26

Reject